## Resource

**Aging, Senescence and Plasticity**

# SenSet defines cell-type specific senescence signatures in the aged human lung

Euxhen Hasanaj [ID] [1,15], Delphine Beaulieu [ID] [2,15], Cankun Wang [ID] [3], Qianjiang Hu [ID] [2], Lorena Rosas [ID] [4], Marta Bueno [ID] [2], John C Sembrat[2], Ricardo H Pineda[2], Maria Camila Melo-Narvaez[5,6], Nayra Cardenes [ID] [2], Zhao Yanwu[2], Zhang Yingze[2], Robert Lafyatis [ID] [7], Alison Morris[2], Ana Mora [ID] [8], Mauricio Rojas [ID] [8], Dongmei Li[9], Irfan Rahman [ID] [10], Gloria S Pryhuber [ID] [11], Mareike Lehmann[5,6,12], Jonathan Alder [ID] [2], Aditi Gurkar [ID] [13], Toren Finkel [ID] [13], Qin Ma [ID] [3], Jose Lugo-Martinez [ID] [14], Barnabás Póczos[1], Ziv Bar-Joseph [ID] [1,14 ✉], Oliver Eickelberg [ID] [2 ✉] & Melanie Königshoff [ID] [2 ✉]

## Abstract

Cellular senescence is defined as an irreversible growth arrest observed when cells are exposed to a variety of stressors, including DNA damage, oxidative stress, or nutrient deprivation. Although senescence is a well-established driver of aging and age-related diseases, it is a highly heterogeneous process with significant variations across organisms, tissues, and cell types. The relatively low abundance of senescent cells in healthy aged tissues poses a major challenge to the longitudinal study of senescence in specific organs, including the human lung. To overcome this limitation, we developed a positive-unlabeled learning framework to generate a comprehensive list of senescence marker genes in human lungs (termed SenSet) using the largest publicly available single-cell lung dataset, the Human Lung Cell Atlas (HLCA). We validated SenSet in a highly complex ex vivo human 3D lung tissue culture model subjected to the senescence inducers bleomycin, doxorubicin, or irradiation, and established its sensitivity and accuracy in characterizing senescence. Using SenSet, we identified and validated cell-type-specific senescence signatures in distinct lung cell populations upon aging and environmental exposure. Our study provides a comprehensive analysis of senescent cells in the healthy aging lung, presenting fundamental implications for our understanding of major lung diseases, including cancer, fibrosis, chronic obstructive pulmonary disease, or asthma.

**Keywords** Senescence; Markers; Lung; Positive-unlabeled Learning; Transcriptomics
**Subject Categories** Molecular Biology of Disease; Respiratory System

See also: Aging, Senescence and Plasticity

## Introduction

Cellular senescence refers to a permanent arrest of cell division, triggered by a variety of stressors or insults. DNA damage is a common inducer of senescence, which can occur over time or in response to oxidative stress, nutrient deprivation, and oncogenic signaling, among others (Hayflick and Moorhead, 1961; Finkel and Holbrook, 2000; Campisi and d'Adda di Fagagna, 2007; van Deursen, 2014; Huang et al, 2022; SenNet Consortium, 2022). The absence of cell division can detrimentally impact tissue regeneration and repair, thereby contributing to numerous age-related diseases, including but not limited to cardiopulmonary and neurodegenerative diseases or metabolic syndromes (Mehdizadeh et al, 2022; Martínez-Cué and Rueda, 2020; Murakami et al, 2022). Senescent cells (SnCs) are resistant to apoptosis, which at least partially accounts for the accumulation of SnCs in advanced age. Moreover, SnCs have been known to undergo the senescence-associated secretory phenotype (SASP), further exacerbating tissue

[1]Machine Learning Department, School of Computer Science, Carnegie Mellon University, Pittsburgh, PA, USA. [2]Center of Lung Aging and Regeneration, Division of Pulmonary, Allergy, Critical Care, and Sleep Medicine, University of Pittsburgh School of Medicine, Pittsburgh, PA, USA. [3]Department of Biomedical Informatics, College of Medicine, Ohio State University, Columbus, OH, USA. [4]Dorothy M. Davis Heart and Lung Research Institute, Division of Pulmonary, Critical Care and Sleep Medicine, Department of Medicine, Ohio State University, Columbus, OH, USA. [5]Comprehensive Pneumology Center (CPC) with the CPC-M bioArchive / Institute of Lung Health and Immunity (LHI), Helmholtz Zentrum München; Member of the German Center for Lung Research (DZL), Munich, Germany. [6]Institute for Lung Research, Philipps-University Marburg, German Center for Lung Research (DZL), Marburg, Germany. [7]Division of Rheumatology, Department of Medicine, University of Pittsburgh Medical Center, Pittsburgh, PA, USA. [8]Dorothy M. Davis Heart and Lung Research Institute, Division of Pulmonary, Critical Care, and Sleep Medicine, Department of Internal Medicine, Ohio State University, Columbus, OH, USA. [9]Department of Clinical and Translational Research, University of Rochester Medical Center, Rochester, NY, USA. [10]Department of Environmental Medicine, University of Rochester Medical Center, Rochester, NY, USA. [11]Department of Pediatrics, University of Rochester Medical Center, Rochester, NY, USA. [12]Institute for Lung Health (ILH), German Center for Lung Research (DZL), Giessen, Germany. [13]Aging Institute, University of Pittsburgh School of Medicine, Pittsburgh, PA, USA. [14]Computational Biology Department, School of Computer Science, Carnegie Mellon University, Pittsburgh, PA, USA. [15]These authors contributed equally: Euxhen Hasanaj, Delphine Beaulieu. ✉E-mail: zivbj@cs.cmu.edu; oliver.eickelberg@pitt.edu; koenigm@pitt.edu

dysfunction in aged individuals by selective secretion of soluble mediators driving senescence in adjacent cells (Coppé et al, 2010). Both SASP and senescence markers are highly cell type-dependent (Tripathi et al, 2021). As a result, precisely characterizing the identity of SnCs, especially in healthy tissues across age, is imperative for understanding aging.

The discovery of genetic markers for senescence is a crucial first step in understanding the mechanisms involved. Not only will such markers facilitate the identification of SnCs, but they will also allow us to track the trajectory of SnC development, their spatial location, and the cell-cell interactions that drive senescence. As such, senescence markers could help identify potential targets for therapeutic interventions, particularly when removal of SnCs could be beneficial.

A number of prior studies and methods have sought to compile gene sets associated with cellular senescence using data from gene knockout models, functional assays, and curated literature reviews. For example, the SenMayo list includes 125 genes reported to be upregulated in human or mouse SnCs (Saul et al, 2022). Similarly, Fridman and Tainsky cataloged genes involved in senescence-related pathways (Fridman and Tainsky, 2008), while CellAge is a curated database of 279 genes implicated in the regulation of senescence (Avelar et al, 2020). A limitation of using these sets is their limited overlap and the lack of validation in an independent, large cohort of aging individuals.

In this work, we aimed to identify a refined set of marker genes with high sensitivity for detecting senescence in human lungs by leveraging these prior gene sets. Although the genes in these prior sets may not be universally consistent, they can be interpreted as partial or noisy information for labeling senescence. To make use of this incomplete supervision, we adopted a *weakly supervised learning* approach (Zhou, 2018), which can effectively leverage incomplete information to classify samples (in our case, to determine if a cell is senescent or not). This approach is well-suited to handling the inherent variability and complexity of biological systems, where a gene may have multiple functions, including roles beyond the promotion of senescence. Specifically, we used a class of learners known as positive-unlabeled (PU) learning algorithms (Bekker and Davis, 2020) to identify SnCs in a large cohort of single-cell lung transcriptomes spanning the human lifespan. While it is currently impossible to definitively identify which cells in this cohort are senescent, PU learning allows us to isolate a subset of cells in older individuals that differ significantly from healthy young cells, based on prior knowledge of senescence-associated genes. To mitigate confounding effects arising from age-related changes unrelated to senescence (e.g., inflammation or epigenetic drift), we used a variant known as the PUc estimator (Sakai and Shimizu, 2019), which explicitly accounts for covariate shift between age groups.

Using our PU learning framework, we derived a refined set of senescence-associated genes by integrating four prior marker lists: GO:0090398, Fridman, SenMayo, and CellAge (Ashburner et al, 2000; Fridman and Tainsky, 2008; Saul et al, 2022; Avelar et al, 2020). The analysis was performed on the Human Lung Cell Atlas (HLCA) (Sikkema et al, 2023), a large single-cell dataset comprising 106 healthy donors aged 10 to 76. By applying PU learning across different age groups, we identified a subset of genes showing robust senescence-related activity, which we termed SenSet. We benchmarked SenSet using four independent public datasets spanning replicative senescence, telomere dysfunction, and fibrotic disease models. We then validated SenSet in human ex vivo lung tissue, using precision-cut lung slices (PCLS) subjected to senescence induction by bleomycin, doxorubicin, or irradiation (Aoshiba et al, 2003; Yang et al, 2012; Borrego-Soto et al, 2015). This approach confirmed the effectiveness of SenSet in identifying SnCs and established its value as a robust senescence gene signature.

## Results

We developed a computational method that uses the largest publicly available single-cell lung dataset—the Human Lung Cell Atlas (HLCA) (Sikkema et al, 2023)—to identify senescent cell populations in the lung across different ages. Our approach is based on positive-unlabeled learning under covariate shift (PUc) (Sakai and Shimizu, 2019) that enables the derivation of a list of senescence markers by direct differential expression (DE) analysis of healthy (i.e., non-senescent) and senescent cells in aged samples (Fig. 1). To achieve this, we trained and tested this PUc learning approach by treating different age groups in the HLCA as (un)labeled data ("Methods").

### Demographics and characterization of the HLCA

The 106 tissue samples in the HLCA were derived from individuals aged 10–76 years, including 51 never-smokers, 19 former smokers, and 28 active smokers (Fig. 2A). Overall, within the HLCA, tissue samples were derived from 69 males and 38 females. Among the 51 never-smokers, 33 were male, with most of them belonging to the older age group (Fig. 2B). Thus, this dataset enables analysis of senescence signatures based on cigarette smoke exposure. Most of the samples originated from lung parenchyma from donor lungs that have been deemed not suitable for transplantation (Fig. 2C). A total of 50 cell types were present in the atlas at the finest annotation level, with respiratory basal cells and alveolar macrophages being the two most prevalent cell types among the never-smoker (NS) group (Fig. 2D–H). Among active smokers, type II pneumocytes and basal cells were the most common (Fig. EV1). The total number of cells was 584,944 with 301,791 cells from never-smokers (Fig. EV2). Average total gene counts increased with age among smokers with a Pearson correlation of 0.33 ($P = 0.01$), while a slight but non-significant decrease was observed for never-smokers (Fig. 2I). We first analyzed the expression of *CDKN1A* and *CDKN2A*, which encode the senescence markers p21 and p16, respectively. Notably, *CDKN1A* was upregulated in smokers for the older two age groups (Fig. 2J). The most significant upregulation was observed for the oldest smoker group compared with non-smokers (two-tailed *t* Test, $P = 0.009$). No significant differences in *CDKN2A* expression were observed across age or by comparing smokers with never-smokers (Fig. 2K).

### Generation of SenSet from the HLCA

Several senescence gene sets have been published to date (GO: 0090398, Fridman, SenMayo, and CellAge (Ashburner et al, 2000; Fridman and Tainsky, 2008; Saul et al, 2022; Avelar et al, 2020)). We first examined the extent of overlap between these gene sets. We observed that the pairwise overlap between them is relatively

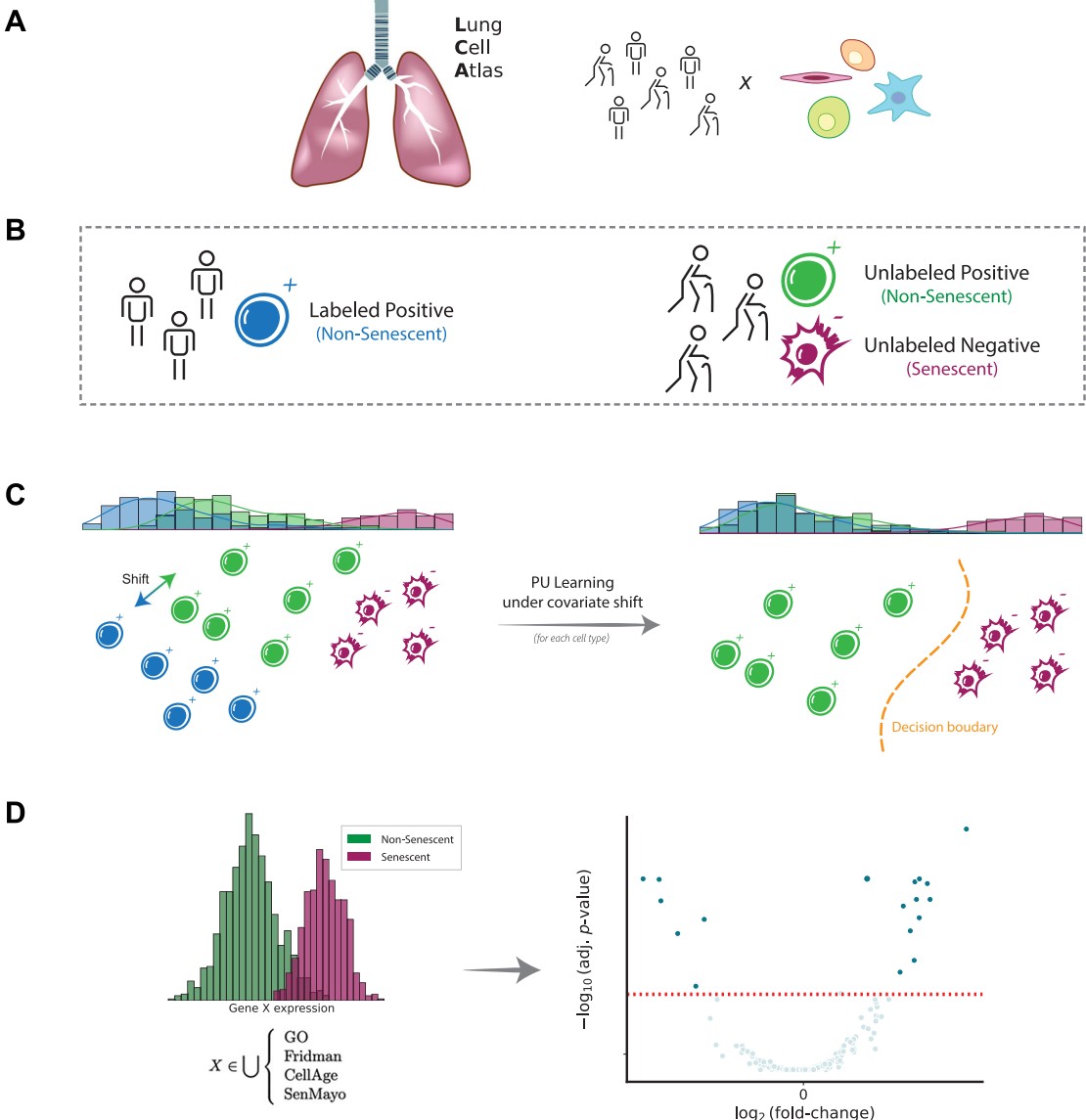

**Figure 1.  PUc pipeline for senescent-cell detection.**

Given a large single-cell lung cohort from young and old donors (**A**), we designate cells from young donors as labeled positives (non-senescent), whereas all cells from old donors are initially unlabeled (**B**). A covariate-shift-aware positive-unlabeled classifier (PUc) assigns senescence probabilities to the unlabeled cells and flags those with high scores as SnCs (**C**). Predicted SnCs are aggregated within each type to derive refined, cell-type-specific senescence-marker profiles (**D**).

small compared with the size of each individual gene set, with the highest overlap of 34 genes shared between the GO and CellAge sets. The union $\mathbf{U}$ of all sets contains 501 unique marker genes, of which 434 were detected in the HLCA.

We sought to identify a subset of $\mathbf{U}$ that demonstrates greater sensitivity for senescence. The PUc estimator (Sakai and Shimizu, 2019) constructs a model of healthy cells based on data from (non-smoker) young and middle-aged individuals (groups $\mathcal{Y}$, $\mathcal{M}$, respectively) and applies this model to identify cells that are senescent in the aged group (group $\mathcal{A}$, Figs. 1 and 2A). PUc accounts for potential covariate shifts in $\mathcal{A}$, which may arise due to other aging processes and hallmarks, including inflammation or epigenetic alterations (Li et al, 2023a; Wang et al, 2022a). The advantage of this approach is that it allows for a direct comparison

of SnCs against non-SnCs within the same group, the oldest age group $\mathcal{A}$. This addresses the challenge of age-related confounding factors that may arise when comparing older with younger individuals. While PUc identifies cells that generally deviate from the healthy (young) profile, we hypothesize that a significant proportion of these non-healthy cells identified by PUc are indeed senescent. We denote these cells with a $(-)$ superscript to signify that they belong to the negative (senescent) class. Similarly, a $(+)$ superscript will denote non-senescent cells for that group.

We applied PUc to 31 cell types in the HLCA with a sufficient number of cells per age group (at least 50), using data exclusively from non-smokers to study senescence genes without the impact of cigarette smoke exposure. PUc identified at least 10 cells in the negative class—claimed here to be senescent—within 22 of these

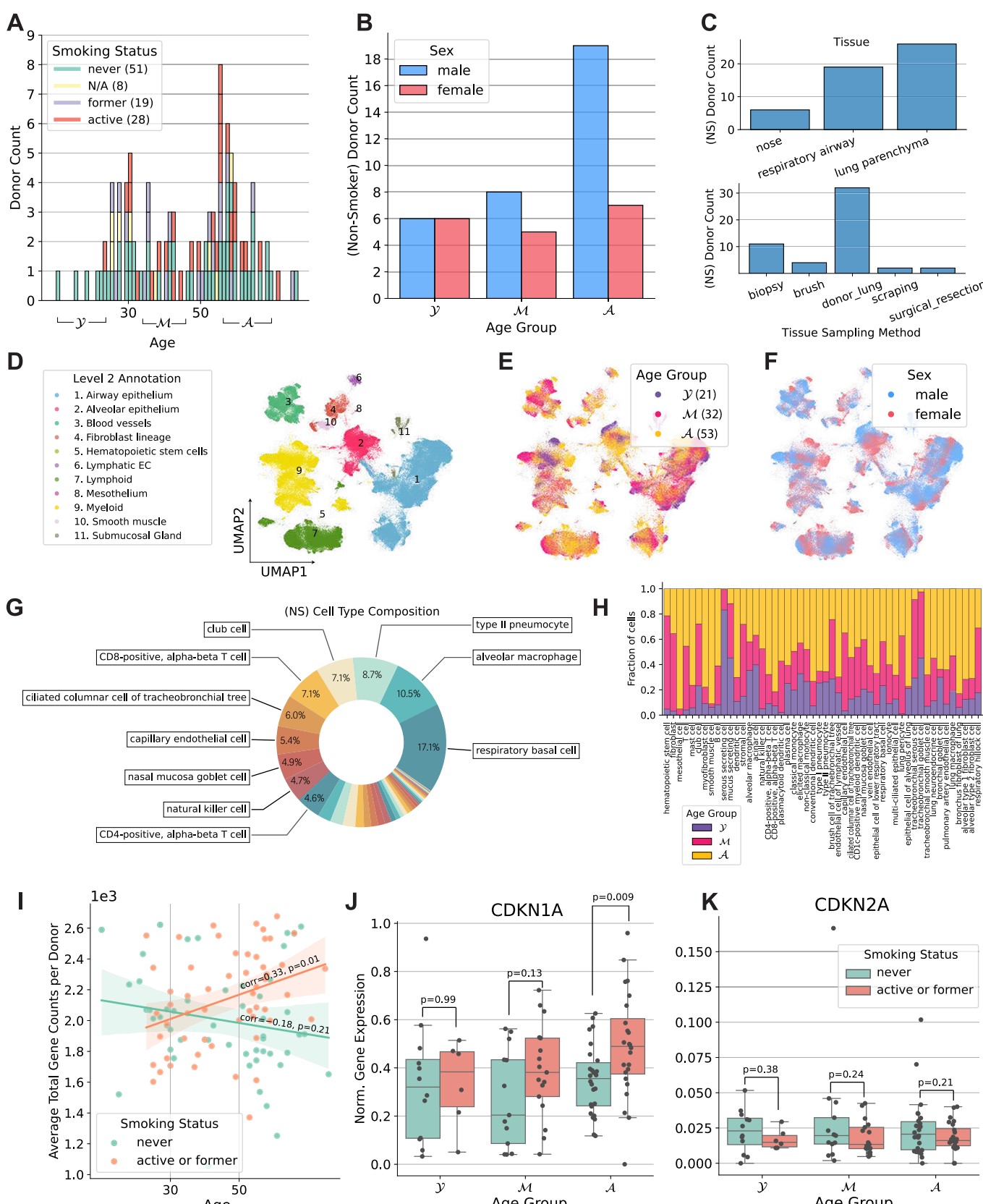

 **Figure 2.   Overview of the Human Lung Cell Atlas dataset.**

(A) Donor age distribution stratified by smoking status. (B) Donor counts by age group and sex among non-smokers (NS). (C) Donor numbers per tissue and sampling method (NS). (D–F) UMAP embedding colored by (D) level-2 cell types, (E) age group, and (F) sex (NS). (G) Cell-type composition for non-smokers. (H) Age-group representation within each cell type (NS). (I) Mean normalized cell counts per donor ("corr" denotes Pearson correlation, with *P* values obtained via a two-sided *t* test). (J, K) Normalized expression of CDKN1A and CDKN2A, respectively. *P* values obtained via a two-sided *t* test. For each panel, values of *n*, following the bar order, are: 12, 6, 13, 17, 26, 24. Boxplots show median, interquartile range, and whiskers at 1.5× IQR.

cell types (Figs. 3A, EV3, and EV4). The proportion of cells assigned to this class ranged from 0 to 76%. Major cell types with high enrichment for senescence markers included alveolar type 1 fibroblasts, respiratory basal cells, and tracheobronchial smooth muscle cells (44%, 39%, and 53%, respectively). Of note, for some of the cell types with a high percentage of SnCs, such as tracheobronchial serous cells (76%), few differentially expressed (DE) genes in these cell types were consistent with other types, suggesting mislabeling or that PUc assumptions for senescence might not hold true for this example.

## Differential expression analysis and enrichment of SenSet genes

To identify markers, we performed a DE test between senescent $\mathcal{A}^{(-)}$ and non-senescent $\mathcal{A}^{(+)}$ cells for each cell type within the old age group and identified genes in the overlap senescence signature that were significantly enriched in at least six cell types (adj. $P = 0.05$). This number was chosen to obtain a set approximately the same size as the base sets. We term our list SenSet (Table 1, 106 genes). SenSet showed the highest overlap with CellAge (52 genes), followed by Fridman (32) and SenMayo (26) (Fig. 3B).

The senescence hallmark gene *CDKN1A* was enriched in 7 cell types; thus, it is included in SenSet, while *CDKN2A*, which was enriched in only alveolar macrophages, is not included (Fig. EV5). SenSet also contains SASP protein members, such as C-X-C motif chemokine ligand 8 (*CXCL8*), interleukin 18 (*IL18*), and insulin growth factor binding protein 7 (*IGFBP7*) (Ortiz-Montero et al, 2017; Zhang et al, 2019; Dinarello, 2006; Siraj et al, 2024). Additional genes upregulated in most cell types$^{(-)}$ include ZFP36 ring finger protein (*ZFP36*, 16 cell types), Jun proto-oncogene (*JUN*, 13), and thioredoxin interacting protein (*TXNIP*), early growth response 1 (*EGR1*), Fos proto-oncogene (*FOS*) (11 each), all of which encode proteins involved in signaling pathways that regulate the transcriptional response to hypoxia and cellular stress (Hettiarachchi et al, 2019; Chinenov and Kerppola, 2001) (Fig. 3C). In contrast, nucleoside diphosphate kinase 2 (*NME2*), a suppressor of apoptosis, was downregulated in 9 cell types, followed by nucleophosmin 1 (*NPM1*) and glyceraldehyde-3-phosphate dehydrogenase (*GAPDH*) (7 each), involved in DNA replication and cell cycle (Liu et al, 2015; Box et al, 2016; Mansur et al, 1993).

Gene set enrichment analysis (GSEA) (Subramanian et al, 2005) using the MSigDB gene set (Liberzon et al, 2015) revealed that SenSet is significantly enriched for genes involved in TNF-alpha signaling via NF-*κ*B (27 genes, adj. $P = 1e-29$), apoptosis (15 genes, adj. $P = 1e-13$), and hypoxia (15 genes, adj. $P = 1e-12$) (Fig. 3D). In addition, SenSet is enriched for genes associated with arthritis (12 genes, adj. $P = 1e-8$) and lung disease (10 genes, adj. $P = 1e-8$) based on Jensen's disease set (Grissa et al, 2022). Notably, gene ontology

(GO) (Ashburner et al, 2000) analysis highlighted enrichment for the process "regulation of smooth muscle cell proliferation" (adj. $P = 1e-6$). A full list of significant GO categories is presented in Fig. EV6.

Finally, we performed a similar DE analysis for the middle-aged group $\mathcal{M}$ that resulted in a list of 139 marker genes. Of these, 90 overlap with our original SenSet (106 genes), demonstrating strong agreement between the two signatures. A breakdown of the overlap per cell type is shown in Fig. EV7.

## Cell-type activation of SenSet genes in the atlas

Cell types showed considerable heterogeneity in the expression of SenSet markers (Fig. 3E,F). From all SenSet genes, 87 were upregulated in alveolar macrophages$^{(-)}$ and 84 in tracheobronchial smooth muscle cells (TSM)$^{(-)}$, representing the highest numbers among the 22 cell types considered. The finding for TSM cells aligns with the GO analysis performed earlier. Conversely, basal$^{(-)}$ cells and type 1 fibroblasts$^{(-)}$ showed a downregulation of 43 and 34 genes, respectively.

Type II pneumocytes and fibroblasts are crucial structural cell types in the lung that have been implicated in senescence (Yao et al, 2021; Hayflick and Moorhead, 1961; Lin and Xu, 2020). In fibroblasts$^{(-)}$, 19 SenSet genes were upregulated. Type II pneumocytes$^{(-)}$ also show an upregulation of 19 SenSet genes (different set), and 1 downregulated gene, *CTNNB1*. We found substantial overlap in upregulated genes between basal$^{(-)}$ cells and type II$^{(-)}$ pneumocytes, fibroblasts$^{(-)}$, respectively, with *TNFRSF1A*, *CITED2*, and *ZFP36* in common across all three. For instance, 9 genes were upregulated in both fibroblasts$^{(-)}$ and basal$^{(-)}$ cells, and 9 genes were also upregulated in both type II pneumocytes$^{(-)}$ and basal$^{(-)}$ cells (Fig. 3G).

Basal cells represent bona fide stem cells of the lung, and stem cell exhaustion, recognized as a hallmark of aging, has been associated with senescence (Ruzankina and Brown, 2007). Among 34 downregulated SenSet genes in fibroblasts$^{(-)}$, 19 of these were also downregulated in basal$^{(-)}$ cells (Fig. 3H).

Several genes were found to be upregulated in one cell type and downregulated in the other. For instance, *LMNA*, encoding for lamin A protein, known to be downregulated in more immature/undifferentiated cells (Sehgal et al, 2013), was downregulated in basal$^{(-)}$ cells, but upregulated in both fibroblasts$^{(-)}$ and type II pneumocytes$^{(-)}$.

## SenSet is consistently enriched across replicative and fibrotic senescence models

To assess whether SenSet reliably captures senescence across diverse biological settings, we analyzed three independent single-cell and one bulk RNA-seq dataset.

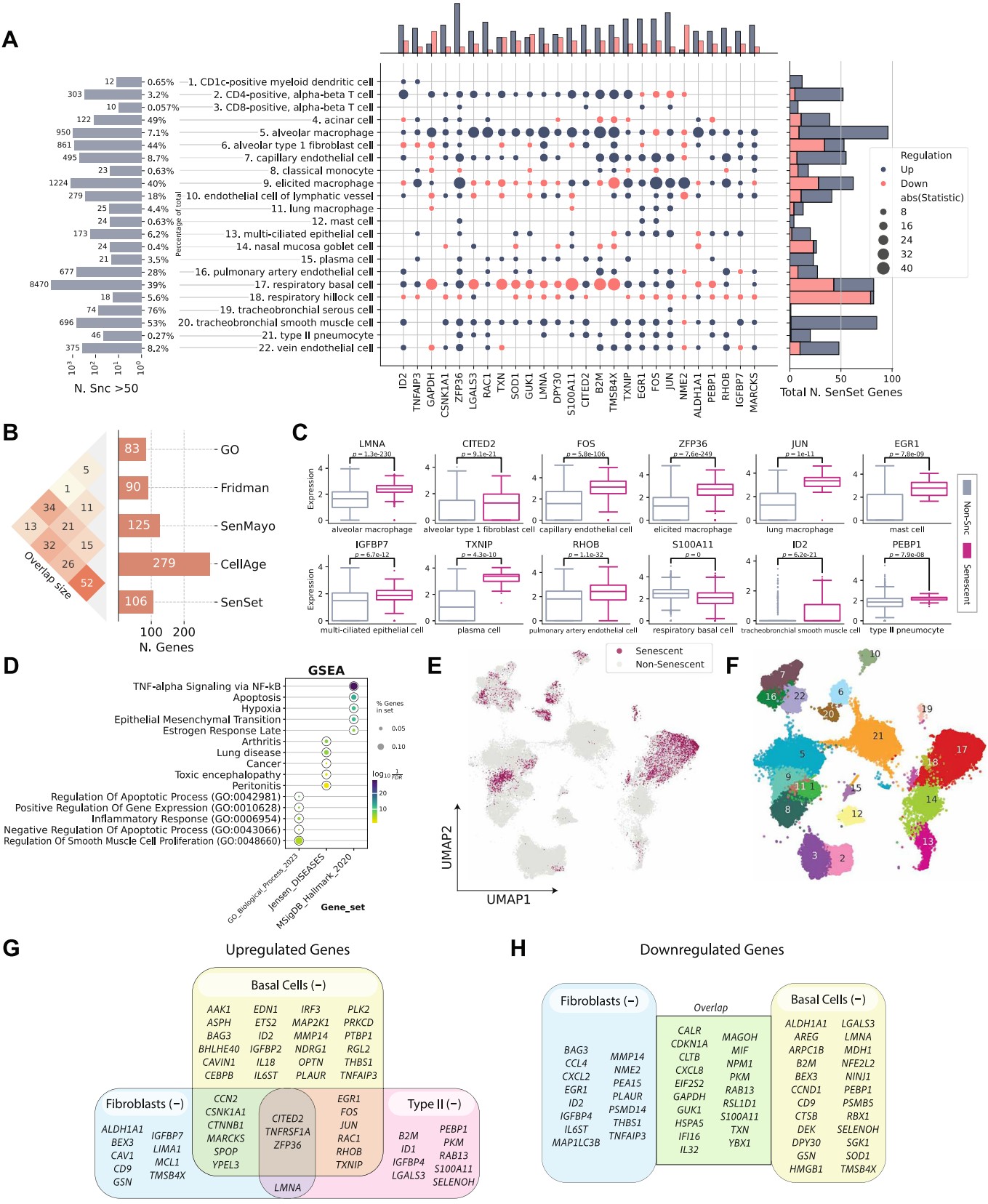

◄  **Figure 3.  PUc identified a novel SenSet senescence signature.**

(A) HLCA and SenSet summary: left, absolute and relative counts of predicted SnCs; center, genes most frequently enriched across cell types; right, number of SenSet markers assigned to each cell type (also Figs. EV3 and 4). (B) Pairwise overlap between SenSet and previously published senescence gene lists. (C) Expression of representative SenSet markers in selected lineages, comparing healthy single-cells and SnCs. (P values obtained via a Mann–Whitney U test; values of n shown in Figs. EV2 and 3. Boxplots show median, interquartile range, and whiskers at 1.5× IQR. (D) Top Gene Ontology, Jensen, and MSigDB terms enriched among SenSet genes. (E) UMAP highlighting predicted SnCs. (F) UMAP colored by cell types that contain at least 10 SnCs; cluster numbers match (A). (G, H) Cell-type-specific enrichment of SenSet markers in basal$^{(-)}$ cells, fibroblasts$^{(-)}$, and type II pneumocytes$^{(-)}$. Source data are available online for this figure.

### Replicative senescence in human mesenchymal stem cells (MSCs)

Using a scRNA-seq dataset from Taherian Fard et al, (2024), we compare late (day 86, T2) with mid-passage (day 49, T1) cells for two biological replicates (#2 and #3). Preranked GSEA using GSEApy (Fang et al, 2023) applied to the five senescence lists showed that both SenSet and the Fridman list achieved high normalized enrichment scores (NES) in each replicate (Fig. 4A,B). SenSet was top-ranked in replicate #3 (FDR $q = 0$), whereas Fridman was the best for replicate #2 (FDR $q = 0$). This concordance with a bona fide replicative senescence model indicates that SenSet reflects a true growth-arrest program rather than an acute stress response. Notably, both *CDKN1A* and *CDKN2A* were upregulated in T2, with *CDKN1A* showing the largest fold change (Fig. 4C–F).

### Telomere-dysfunction model in human alveolar epithelium (A549)

We next analyzed the bulk RNA-seq dataset of Sullivan et al (2021), consisting of an alveolar epithelial-like A549 line in which a dominant-negative TRF2 (T2DN) is conditionally expressed to uncouple telomeres and trigger senescence. Preranked GSEA of T2DN versus control cells yielded significant enrichment for three signatures: Fridman, SenMayo, and SenSet (FDR $q = 0$), with SenSet having the largest proportion of lead genes (39%), followed by SenMayo (31%) (Fig. 4G,H). These results support the notion that SenSet sensitively captures the telomere-dysfunction senescence program in epithelial cells.

### Idiopathic pulmonary fibrosis (IPF) lung atlas

In the IPF atlas of Adams et al (2020), we contrasted IPF and control samples within each annotated cell type. Aggregating preranked GSEA results across cell types revealed that SenSet achieved the lowest overall FDR values, with nine cell types reaching significance of FDR $q = 0$ (Fig. EV8A). We further examined normalized enrichment scores (NES) for both SenSet and SenMayo and observed strong agreement in the directionality of enrichment across cell types (Fig. EV8B). A notable outlier, goblet cells, displayed positive NES for SenSet but a negative NES for SenMayo. A closer inspection showed that 25 SenSet-specific lead genes, absent from SenMayo, drove the positive score. These findings underscore the utility of SenSet for detecting senescence in a fibrotic lung environment.

### Liver cirrhosis

Finally, we analyzed a scRNA-seq cirrhotic-liver dataset (Rama-chandran et al, 2019) consisting of five healthy and five cirrhotic livers. Preranked GSEA assigned SenSet an FDR $q = 0$, with 56 of 101 overlapping genes contributing as lead genes (Fig. EV8C). Of the other four lists, only Fridman was also significantly associated with the differential genes in this dataset, though with a lower significance (FDR $q = 0.001$). Thus, SenSet generalizes beyond

pulmonary tissue and sensitively reports senescence in another fibrotic organ.

Collectively, these analyses demonstrate that SenSet is consistently and selectively enriched across models of replicative, pulmonary-fibrotic, and hepatic-fibrotic senescence, validating its broad utility as a context-independent senescence signature.

## Validation of senescence markers in a human lung tissue model

To further refine and validate SenSet, we utilized a highly complex ex vivo human 3D tissue culture model based on precision-cut lung slices (PCLS). PCLS recapitulates the complexity of the lung environment in situ, enabling the study of various lung cell types and lineages present in the parenchymal region, along with the extracellular matrix (ECM) within the lungs' native 3D architecture at high temporal and spatial resolution. We have previously shown that this model recapitulates the onset and progression of lung injury and disease and enables the testing of candidate therapeutics in viable ex vivo human tissue affected by disease (Uhl et al, 2015; Alsafadi et al, 2017, 2020).

PCLS were generated from the lower left lung lobe (Fig. 5A) of healthy donors aged 20–78 (Table 2). To induce cellular senescence, PCLS were treated with either bleomycin (15 μg/mL) or doxorubicin (0.1 μM) for up to 6 days. Lung structure remained intact after 6 days in culture, as shown by hematoxylin and eosin (H&E) staining (Figs. 5B and EV9).

Senescence induction was validated by several readouts, with an increase in β-galactosidase staining (Fig. 5C), an increase in p21-positive cells, and a decrease in Ki67-positive cells in PCLS (Fig. 5D–F). The increase in p21-positive cells relative to control conditions was significant after bleomycin (3.02-fold ± 0.89) and doxorubicin treatment (2.01-fold ± 0.94, Fig. 5E). The decrease in Ki67-positive cells was significant after bleomycin (0.13-fold ± 0.12) and not significant after doxorubicin (0.22-fold ± 0.13). This was further confirmed by an increase in p21 protein level measured by western blot after bleomycin (6.05-fold ± 4.19) and doxorubicin treatment (1.94-fold ± 1.49) (Fig. 5G). Moreover, *GDF-15*, a known SASP protein, was significantly increased after bleomycin (4.63-fold ± 2.68) and doxorubicin treatment (4.06-fold ± 3.11) (Fig. 5H). Additional expression of fibrotic markers and p16/p21 levels is shown in Fig. EV10.

To validate that our SenSet list, which was derived from the HCLA using lung tissue across the ages, is indeed a senescence signature, we subjected our human senescence induction model to single-nucleus RNA sequencing (snRNA-seq) and further analyzed cell-type-specific gene expression. Senescence was induced in PCLS by bleomycin or doxorubicin as described above, and we further included PCLS subjected to irradiation, as previously reported (Melo-Narvaez et al, 2024). The samples used for irradiation

**Table 1.** All 106 SenSet genes with full names.

| Symbol | Full name | Symbol | Full name |
|---|---|---|---|
| AAK1 | AP2 Associated Kinase 1 | IRF3 | Interferon Regulatory Factor 3 |
| AKR1B1 | Aldo-Keto Reductase Family 1 Member B | ISG15 | ISG15 Ubiquitin-Like Modifier |
| ALDH1A1 | Aldehyde Dehydrogenase 1 Family Member A1 | JUN | Jun Proto-Oncogene, AP-1 Transcription Factor Subunit |
| AREG | Amphiregulin | LGALS3 | Galectin 3 |
| ARPC1B | Actin-Related Protein 2/3 Complex Subunit 1B | LIMA1 | LIM Domain And Actin Binding 1 |
| ASPH | Aspartate Beta-Hydroxylase | LMNA | Lamin A/C |
| B2M | Beta-2-Microglobulin | MAGOH | Mago Homolog, Exon Junction Complex Subunit |
| BAG3 | BAG Cochaperone 3 | MAP1LC3B | Microtubule Associated Protein 1 Light Chain 3 Beta |
| BEX3 | Brain Expressed X-Linked 3 | MAP2K1 | Mitogen-Activated Protein Kinase Kinase 1 |
| BHLHE40 | Basic Helix-Loop-Helix Family Member E40 | MAP2K3 | Mitogen-Activated Protein Kinase Kinase 3 |
| CALR | Calreticulin | MARCKS | Myristoylated Alanine Rich Protein Kinase C Substrate |
| CAV1 | Caveolin 1 | MCL1 | MCL1 Apoptosis Regulator, BCL2 Family Member |
| CAVIN1 | Caveolae Associated Protein 1 | MDH1 | Malate Dehydrogenase 1 |
| CCL3 | C-C Motif Chemokine Ligand 3 | MIF | Macrophage Migration Inhibitory Factor |
| CCL3L1 | C-C Motif Chemokine Ligand 3 Like 1 | MMP14 | Matrix Metallopeptidase 14 |
| CCL4 | C-C Motif Chemokine Ligand 4 | NDRG1 | N-Myc Downstream Regulated 1 |
| CCN2 | Cellular Communication Network Factor 2 | NFE2L2 | NFE2 Like BZIP Transcription Factor 2 |
| CCND1 | Cyclin D1 | NINJ1 | Ninjurin 1 |
| CD44 | CD44 Molecule (IN Blood Group) | NME2 | NME/NM23 Nucleoside Diphosphate Kinase 2 |
| CD9 | CD9 Molecule | NPM1 | Nucleophosmin 1 |
| CDKN1A | Cyclin Dependent Kinase Inhibitor 1A | OPTN | Optineurin |
| CEBPB | CCAAT Enhancer Binding Protein Beta | PEA15 | Proliferation And Apoptosis Adapter Protein 15 |
| CITED2 | Cbp/P300 Interacting Transactivator With Glu/Asp Rich Carboxy-Terminal Domain 2 | PEBP1 | Phosphatidylethanolamine Binding Protein 1 |
| CLTB | Clathrin Light Chain B | PKM | Pyruvate Kinase M1/2 |
| CSNK1A1 | Casein Kinase 1 Alpha 1 | PLAUR | Plasminogen Activator, Urokinase Receptor |
| CTNNB1 | Catenin Beta 1 | PLK2 | Polo Like Kinase 2 |
| CTSB | Cathepsin B | PRKCD | Protein Kinase C Delta |
| CXCL2 | C-X-C Motif Chemokine Ligand 2 | PSMB5 | Proteasome 20S Subunit Beta 5 |
| CXCL8 | C-X-C Motif Chemokine Ligand 8 | PSMD14 | Proteasome 26S Subunit, Non-ATPase 14 |
| DEK | DEK Proto-Oncogene | PTBP1 | Polypyrimidine Tract Binding Protein 1 |
| DPY30 | Dpy-30 Histone Methyltransferase Complex Regulatory Subunit | RAB13 | RAB13, Member RAS Oncogene Family |
| EDN1 | Endothelin 1 | RAC1 | Rac Family Small GTPase 1 |
| EGR1 | Early Growth Response 1 | RBX1 | Ring-Box 1 |
| EIF2S2 | Eukaryotic Translation Initiation Factor 2 Subunit Beta | RGL2 | Ral Guanine Nucleotide Dissociation Stimulator Like 2 |
| ETS2 | ETS Proto-Oncogene 2, Transcription Factor | RHOB | Ras Homolog Family Member B |
| EWSR1 | EWS RNA Binding Protein 1 | RSL1D1 | Ribosomal L1 Domain Containing 1 |
| FOS | Fos Proto-Oncogene, AP-1 Transcription Factor Subunit | S100A11 | S100 Calcium Binding Protein A11 |
| GAPDH | Glyceraldehyde-3-Phosphate Dehydrogenase | SELENOH | Selenoprotein H |
| GMFG | Glia Maturation Factor Gamma | SGK1 | Serum/Glucocorticoid Regulated Kinase 1 |
| GSN | Gelsolin | SMARCB1 | SWI/SNF Related, Matrix Associated, Actin Dependent Regulator Of Chromatin, Subfamily B, Member 1 |
| GUK1 | Guanylate Kinase 1 | SOD1 | Superoxide Dismutase 1 |
| HDAC1 | Histone Deacetylase 1 | SPOP | Speckle Type BTB/POZ Protein |

**Table 1.** (continued)

| Symbol | Full name | Symbol | Full name |
|--------|-----------|--------|-----------|
| HMGB1 | High Mobility Group Box 1 | THBS1 | Thrombospondin 1 |
| HSPA5 | Heat Shock Protein Family A (Hsp70) Member 5 | TMSB4X | Thymosin Beta 4 X-Linked |
| ID1 | Inhibitor Of DNA Binding 1 | TNFAIP3 | TNF Alpha Induced Protein 3 |
| ID2 | Inhibitor Of DNA Binding 2 | TNFRSF1A | TNF Receptor Superfamily Member 1A |
| IFI16 | Interferon Gamma Inducible Protein 16 | TPR | Translocated Promoter Region, Nuclear Basket Protein |
| IGFBP2 | Insulin Like Growth Factor Binding Protein 2 | TXN | Thioredoxin |
| IGFBP4 | Insulin Like Growth Factor Binding Protein 4 | TXNIP | Thioredoxin Interacting Protein |
| IGFBP7 | Insulin Like Growth Factor Binding Protein 7 | VIM | Vimentin |
| IL18 | Interleukin 18 | YBX1 | Y-Box Binding Protein 1 |
| IL32 | Interleukin 32 | YPEL3 | Yippee Like 3 |
| IL6ST | Interleukin 6 Cytokine Family Signal Transducer | ZFP36 | ZFP36 Ring Finger Protein |

originated from peritumor tissue. We observed an increase in the average expression of several markers in treated samples, including *GDF-15*, *MMP2*, and *FGF2* (Fig. 5I). Some minor individual differences in total-counts-per-cell were found across conditions (Fig. 6A).

For each gene in SenSet as well as each of the prior gene sets, we performed a Mann–Whitney *U* test (Mann and Whitney, 1947) to determine if the gene is up or downregulated in our PCLS senescence models. Notably, SenSet achieves the highest proportion of significantly regulated genes in all samples when compared to all prior lung senescence lists (adj. *P* = 0.05, Fig. 6B) (an expanded figure with results that include SenePy (Sanborn et al, 2025) is shown in Fig. EV11). Several SenSet genes that were upregulated in $\mathcal{A}^{(-)}$ (HCLA) were also upregulated after senescence induction in human PCLS ex vivo, including *JUN*, and *IGFBP7*. The transcription factor *TXNIP*, which is known to be suppressed by p21 overexpression under disturbed shear stress in endothelial cells (Obikane et al, 2010), was downregulated in response to all three senescence inducers as well as the cell cycle regulators *NME2* and *NPM1*, which were also the top downregulated genes in $\mathcal{A}^{(-)}$ (Fig. 6C). *CDKN1A* was increased after all three treatments, with the highest induction after bleomycin treatment. No such increase was observed for *CDKN2A* (Fig. EV12).

After we compared the expression of senescence markers across conditions using the entire sample, we next turned our attention to cell-type-specific DE analysis of the marker genes. We identified all major cell lineages in our ex vivo human tissue model and identified four major epithelial cell types (Fig. 6D) based on lung canonical markers (Travaglini et al, 2020) (Figs. EV13 and 14), without discernible batch condition or cell cycle effects on data following integration using scVI (Lopez et al, 2018) (Fig. 6E). A UMAP plot (McInnes et al, 2018) of the normalized expression values of *CDKN1A* and *CDKN2A* across conditions reveals a stronger signal for *CDKN1A* in bleomycin and doxorubicin samples (Fig. 6F). For each cell type, we performed a similar Mann–Whitney *U* test between treatment and control samples, and combined the *P* values of these tests using Pearson's method (Pearson, 1933) (Fig. 7A). This analysis revealed that SenSet markers showed significant enrichment across ten cell types (*P* ≤0.05). In comparison, markers from Fridman and SenMayo lists showed significant enrichment in only four cell types. CellAge, on

the other hand, was not significantly enriched in any cell type, likely due to the high number of non-DE markers in that list.

A close inspection of the markers across cell types confirmed that in the prior lists, many genes were not DE in any cell type (Figs. 7B, EV15, EV16, EV17, and EV18). In contrast, nearly all SenSet genes (all but seven) were DE in at least one cell type (Fig. 7C). SenSet markers were predominantly upregulated in AT1$^{(-)}$ and AT2$^{(-)}$ cells, while mostly downregulated in fibroblasts$^{(-)}$ and macrophages$^{(-)}$. Notably, 15 genes downregulated in fibroblasts$^{(-)}$ in the HLCA dataset were also downregulated in fibroblasts$^{(-)}$ in the treated PCLS data: *CALR*, *GAPDH*, *GUK1*, *IL32*, *MIF*, *NME2*, *NPM1*, *PKM*, *PLAUR*, *RAB13*, *S100A11*, *THBS1*, *TNFAIP3*, *TXN*, *YBX1*. A similar correspondence was observed for (elicited) macrophages and AT2 cells. For AT2$^{(-)}$ cells, 13 SenSet markers were upregulated in both the HLCA (19 total) and the PCLS (73 total), including *TNFRSF1A*, *PKM*, *PEBP1*, *ID1*, *ZFP36*, *LGALS3*, *RAC1*, *LMNA*, *S100A11*, *CITED2*, *B2M*, *JUN*, *SELENOH*.

## Analysis of smokers in the HLCA

Air pollutants and cigarette smoke exposure are a major risk factor for the development and exacerbation of age-related lung diseases, including chronic obstructive pulmonary disease (COPD). COPD incidence and prevalence increase with age (Tsuji et al, 2010, 2006; Meiners et al, 2015; Easter et al, 2020). The disease is characterized by impaired lung repair and progressive distal lung tissue destruction (emphysema) and airways remodeling and inflammation (chronic bronchitis) (Christenson et al, 2022). First, we computed the Wasserstein distance (Villani, 2009) between pairs of smokers and non-smokers from different age groups *across all genes*. The Wasserstein distance is a measure of the difference between two probability distributions and provides a notion of how "close" the gene expression of two populations for a given cell type is. We found that for 12 out of 18 cell types, the expression space of young smokers (<30) was closer in distribution to that of old non-smokers (≥ 50) than that of young non-smokers (Fig. 8A).

Given that our SenSet signature was based on non-smokers, we next aimed to investigate whether SenSet enrichment is altered upon smoking. We performed DE testing on a few cell types of interest to evaluate if senescence markers were enriched in smokers for the youngest $\mathcal{Y}$ and oldest age groups $\mathcal{A}$. The analysis showed

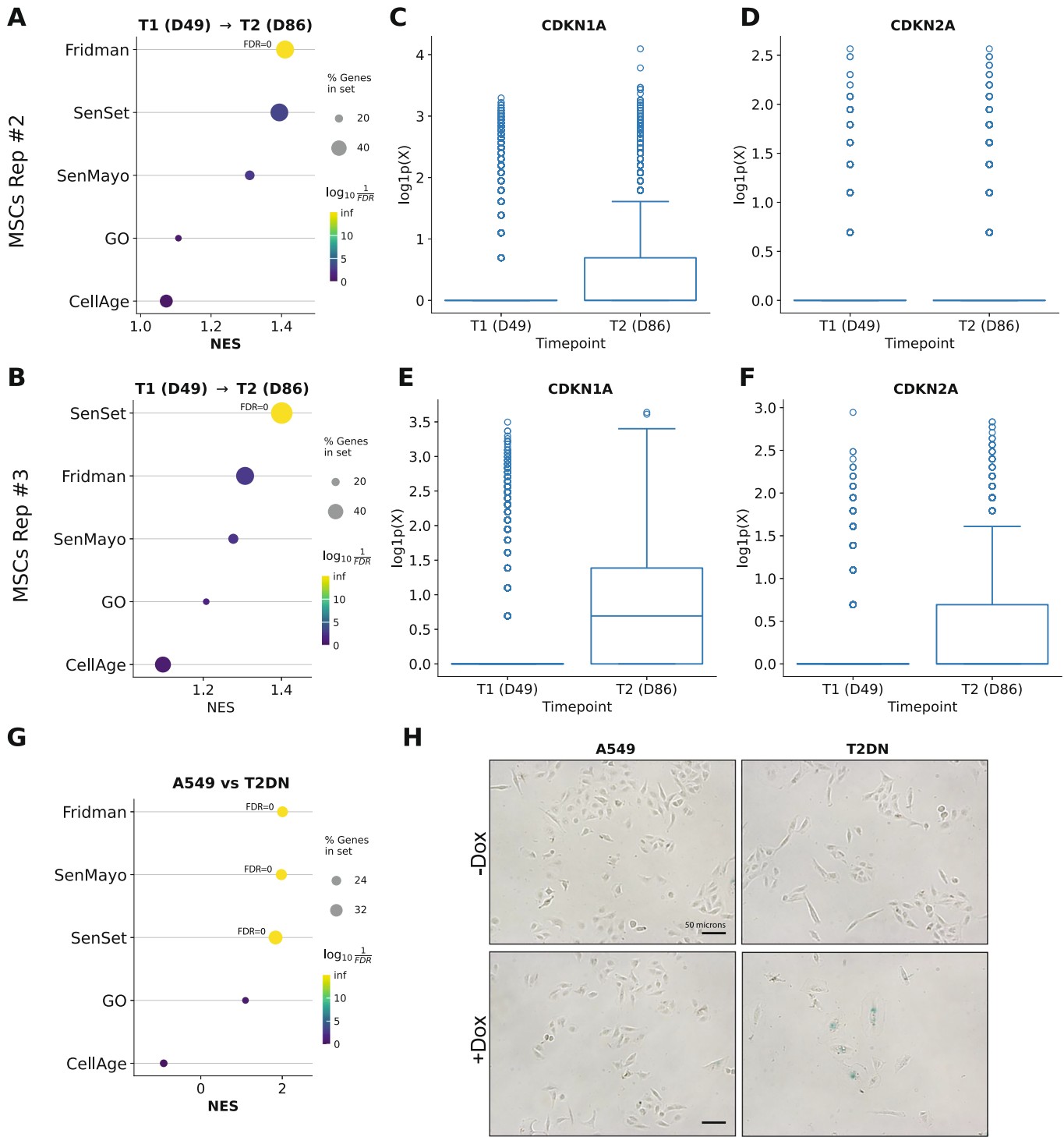

**Figure 4.  Senescence enrichment and marker expression across replicative and telomere-dysfunction models.**

(A, B) Preranked GSEA dotplots for human MSCs undergoing replicative senescence: replicate #2 (A) and replicate #3 (B). (C–F) Log1p-transformed counts for CDKN1A and CDKN2A for replicate #2 (C, D) and replicate #3 (E, F). Boxplots show median, interquartile range, and whiskers at 1.5× IQR. (G) Preranked GSEA dot plot comparing A549 cells expressing dominant-negative TRF2 (T2DN). (H) Representative immunofluorescence images of A549 and T2DN cells cultured ± doxycycline (Dox) to induce TRF2-DN. Scale bar, 50 microns. Source data are available online for this figure.

**Table 2.  Donor information.**

| Donor | Sex | Age | Smoking status | IHF p21 | WB p21 | Multiplex assay | scRNA-seq |
|-------|-----|-----|----------------|---------|--------|-----------------|-----------|
| LTC 113 | F | 56 | Never | ✓ | | | ✓ |
| LTC 117 | M | 73 | Never | ✓ | | ✓ | ✓ |
| LTC 118 | F | 23 | Never | ✓ | | ✓ (D) | |
| LTC 119 | F | 38 | Never | ✓ (D) | | ✓ (D) | |
| LTC 120 | M | 21 | Never | ✓ (B) | | ✓ (B) | ✓ (B) |
| LTC 121 | M | 62 | Former | ✓ (B) | ✓ (B) | ✓ (B) | |
| LTC 124 | M | 36 | Never | ✓ | ✓ | ✓ | ✓ |
| LTC 127 | M | 41 | Never | ✓ | ✓ | ✓ | |
| LTC 137 | F | 78 | Former | ✓ | ✓ | | |
| LTC 164 | M | 57 | Former | | ✓ | | |
| LTC 176 | M | 25 | Never | | ✓ | | |
| LTC 200 | M | 20 | Never | | ✓ | | |
| E170[a] | M | 75 | Former | | | | ✓ (I) |
| E185[b] | F | 81 | Former | | | | ✓ (I) |
| E187 | M | 64 | Former | | | | ✓ (I) |
| E196 | F | 70 | Never | | | | ✓ (I) |

*IHF* immunohistofluorescence, *WB* western blot.
[a]Peritumor tissue.
[b]COPD Gold II; B: Bleomycin; D: Doxorubicin; I: Irradiation.
If parentheses are missing, it is assumed to be (B, D).

differences in senescence enrichment between smokers of varying ages (Fig. 8B). Specifically, we found that most markers were upregulated in basal cells from smokers (around 80% of the genes across all lists except for SenMayo at 50%). For other cell types, including CD4/CD8-positive, alpha-beta T cells and AT2 cells, we found that aged smokers showed an upregulation of a higher number of senescence markers than young smokers, across all marker lists, with SenSet containing most such genes. Lastly, for AT2 cells, around 35% of all genes were downregulated and 40% upregulated in the young, while around 60% were upregulated in the older age group.

## Discussion

Aging remains the strongest risk factor for chronic lung diseases, with SnCs accumulating in tissues and contributing to pathology through mechanisms such as extracellular matrix remodeling and pro-inflammatory signaling. Cellular senescence is induced by diverse stimuli, including oncogene activation associated with tumor suppressor inactivation (Serrano et al, 1997), oxidizing agents inducing DNA damage (Duan et al, 2005), or chemotherapeutic agents, such as bleomycin and doxorubicin (Aoshiba et al, 2003; Fitsiou et al, 2022), with pathways varying by cell type, inducer, and time course. SnCs are defined by an altered DNA damage response ($\gamma$H2AX activation), expression of cyclin-dependent kinase inhibitors (p16, p21), enhanced SASP secretion (mTOR, cGAS–STING, NF-$\kappa$B), and apoptosis resistance (BCL-2) (Hernandez-Segura et al, 2018), yet these features are not unique to senescence and often overlap with other cellular states, including mitochondrial dysfunction or apoptosis.

This study presents a machine learning-based framework to identify a novel gene set specific to SnCs, originally generated using lung samples from the Human Lung Cell Atlas (HLCA). By leveraging single-cell transcriptomic data, our PU learning approach helped us distinguish SnCs across different age groups. Differential expression tests between SnCs and non-SnCs led to the creation of SenSet, which we validated in senescence-induced human lung ex vivo tissue. Senescence was induced in human PCLS by bleomycin, doxorubicin, or irradiation. We subjected our human senescence induction model to snRNA-seq to confirm the enrichment of SenSet at the cell-type level. SenSet is a subset of the union of four existing senescence marker sets, which enabled our weak supervision approach to identify senescence characteristics of cells by incorporating prior knowledge. Unlike prior lists, SenSet was entirely computationally derived, which provides a data-driven identification of genes that capture senescence-specific features across cell types.

Our findings revealed that fibroblasts and basal cells exhibited a high proportion of SnCs in aged lungs, accounting for 44% and 39% of all cells of that type in the oldest age group $\mathcal{A}$. Fibroblasts contribute to alveolar maturation and regeneration by producing extracellular matrix components and are also central to the pathogenesis of age-related diseases, such as idiopathic pulmonary fibrosis (IPF), a disease which shows accumulation of senescent fibroblasts and increased SASP secretion (Álvarez et al, 2017). Similarly, a recent work also demonstrated that fibroblasts are the main cell type undergoing senescence changes under homeostatic conditions and contribute to epithelial regeneration in a novel mouse model tracking p16[INK4a+] cells (Reyes et al, 2022). Previous studies have shown that senolytic treatments targeting SnCs can alleviate fibrosis in mouse models (Schafer et al, 2017). Next to

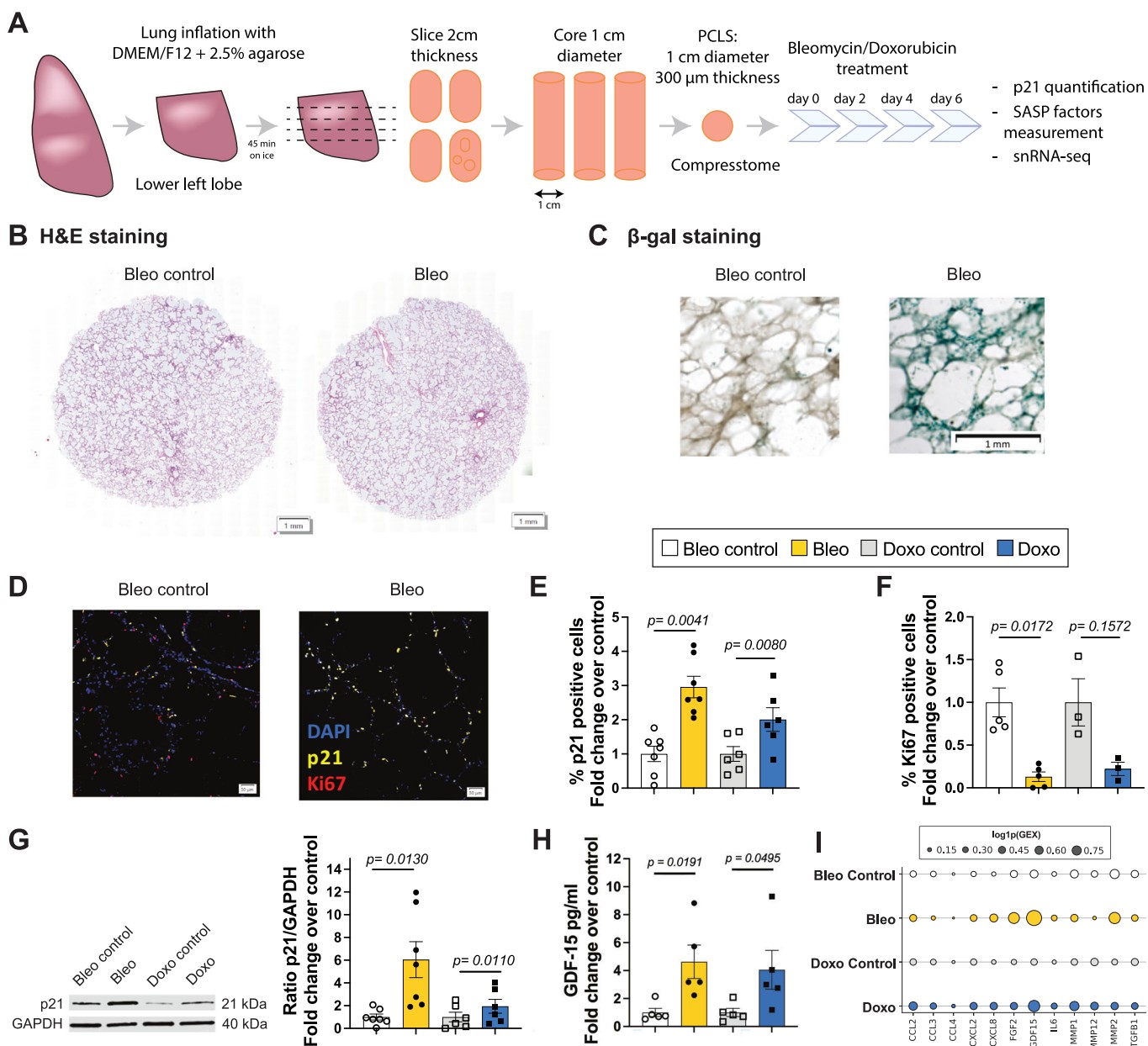

**Figure 5. Senescence induction in human PCLS by DNA damage.**

(A) PCLS were generated from healthy donors' lower left lobe lung with an age range of 20 to 78 years old. Senescence was induced by treatment with bleomycin (Bleo) at 15 µg/mL, or doxorubicin (Doxo) at 0.1 µM for 6 days, and PCLS and supernatants were collected. (B) Hematoxylin eosin (H&E) staining on 4-µm sliced formalin-fixed paraffin-embedded human PCLS at day 6. (C) β-galactosidase staining on whole PCLS at day 6. (D) p21 (yellow) and Ki67 (red) immunohistofluorescence (IHF) on 4-µm sliced formalin-fixed paraffin-embedded human PCLS at day 6. (E) Percentage of p21 and (F) Ki67-positive cells, fold change over control, based on p21 and Ki67 IHF staining after bleomycin (E: $n = 7$; F: $n = 5$) or doxorubicin (E: $n = 6$; F: $n = 3$) treatment. (G) Quantification of p21 protein level by western blot (WB) after bleomycin ($n = 7$) or doxorubicin ($n = 6$) treatment and representative blot. (H) Mean log-expression of SASP factors after snRNA sequencing. (I) SASP factor, GDF-15, measured by Luminex™ assay on human PCLS supernatants after bleomycin ($n = 4$) or doxorubicin treatment ($n = 3$). All error bars represent the standard error of the mean (SEM). P values obtained via a paired t test after verification of normality. Source data are available online for this figure.

fibroblasts, basal cells showed significant SnC-associated gene regulation. Basal cells, which are the main stem cells in the proximal airways, can self-renew and differentiate into different airway epithelial cell types, such as secretory or ciliated cells (Parekh et al, 2020). These cells are critical for maintaining epithelial integrity and repair, particularly after injury. The number of basal cells gradually decreases in the proximal-distal axis in the

airway epithelium. Notably, stem cell exhaustion is a common hallmark of aging (Oh et al, 2014).

Alveolar type 2 (AT2) cells, essential for surfactant production and alveolar repair, exhibited notable senescence-associated changes. AT2 cells have been demonstrated to harbor senescence features in the diseased lung, such as in pulmonary fibrosis (Yao et al, 2021). Elimination of senescent epithelial cells using senolytics

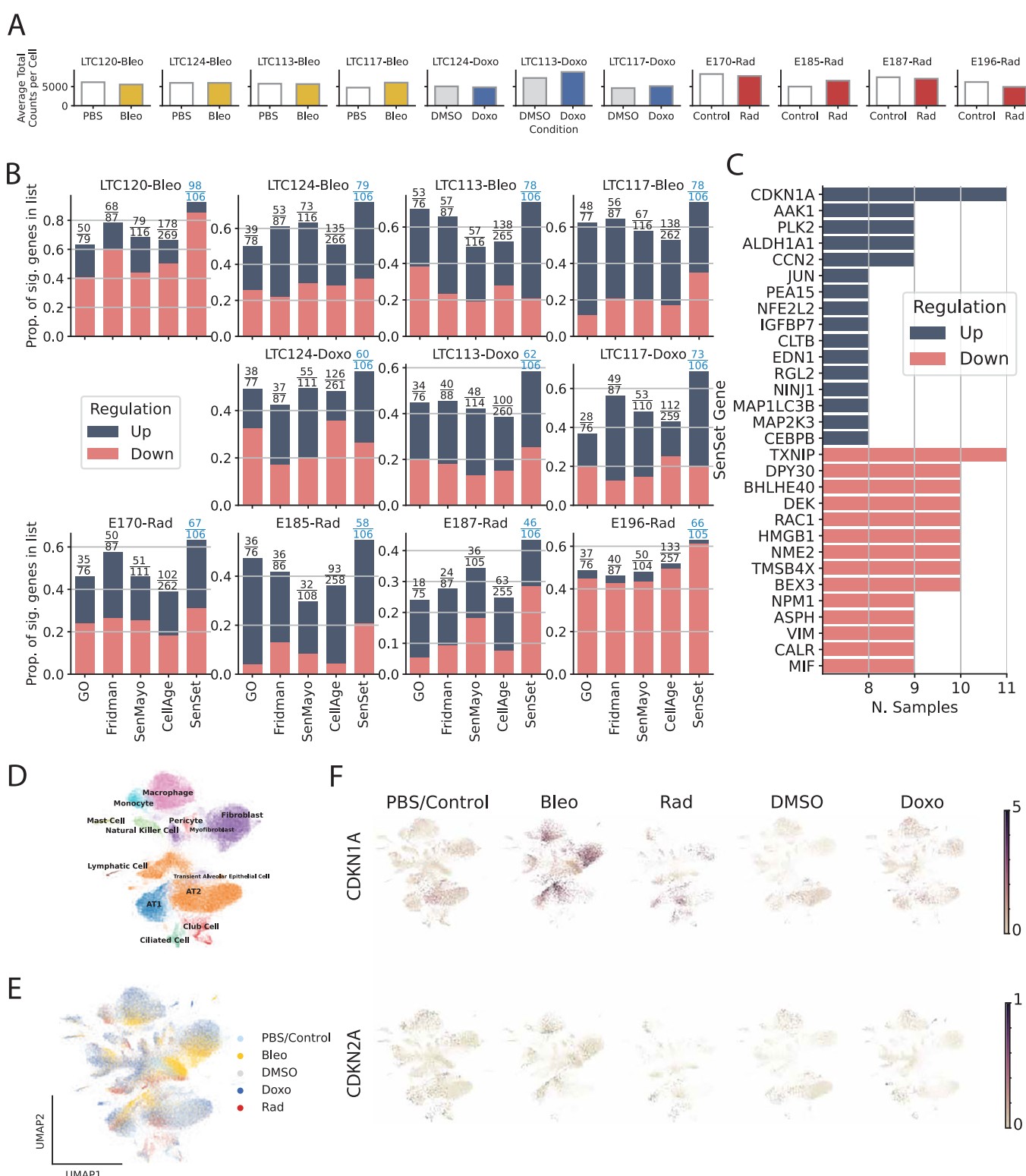

**Figure 6. Validation of SenSet.**

(A) Mean UMI counts per cell across samples and experimental conditions. (B) For each subject, the proportion of genes in each senescence list that are significantly up- or downregulated after treatment. (C) SenSet genes consistently up- (blue) or downregulated (red) in most samples. (D) Leiden clusters annotated with canonical lung markers after scVI harmonization. (E) Same embedding colored by treatment condition. (F) Normalized expression of CDKN1A and CDKN2A across conditions (max value clipped for visibility). Source data are available online for this figure.

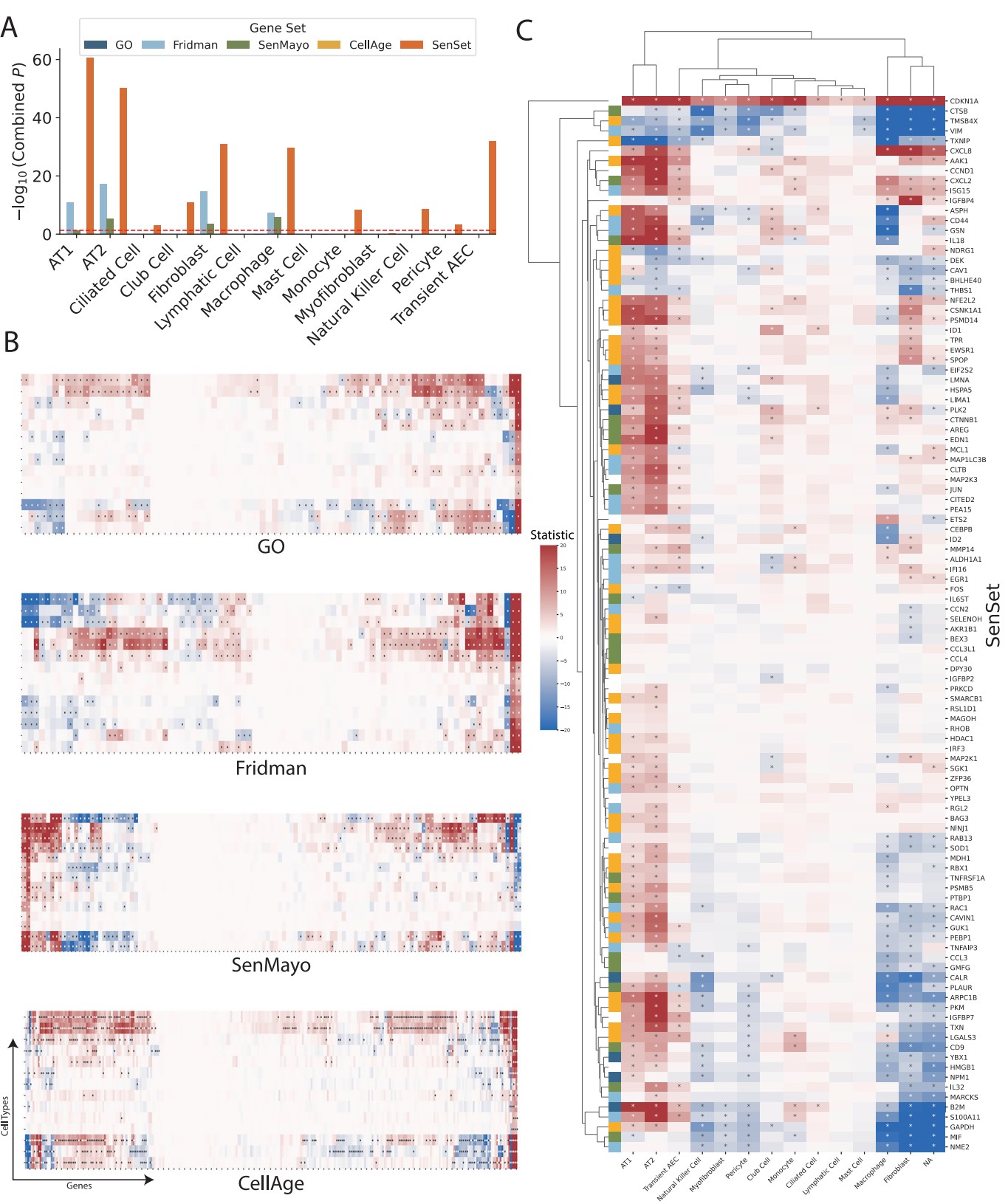

Figure 7.   Cell-type-resolved senescence signatures.

(A) For each cell type, DE tests were run between treatment and control; marker-gene significance across donors was aggregated with Pearson's combined *P* value. (B, C) Wilcoxon rank-sum statistics for every marker gene-cell-type pair in PCLS; asterisks indicate FDR-adjusted *q* < 0.05. Complete heatmaps are shown in Figs. EV15–18. Source data are available online for this figure.

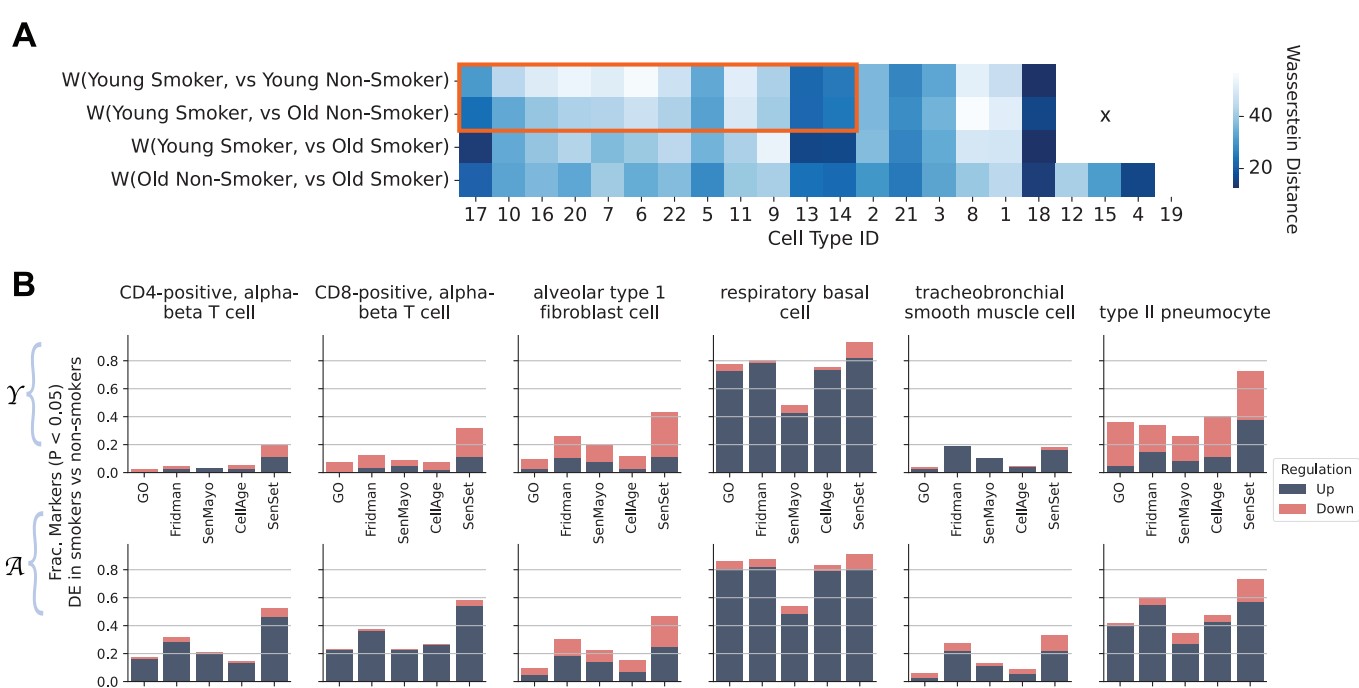

Figure 8.   Smoking-related shifts in senescence signatures within the HLCA.

(A) Wasserstein distance between smoker and non-smoker expression profiles, across different age groups. Cell types outlined in red show a smaller distance for young-smoker vs. old-non-smoker than for young-smoker vs. young-non-smoker, indicating smoking-associated changes that mimic aging. Cell IDs match numbers in Fig. 3A. (B) For selected types, proportion of SenSet genes significantly enriched in smokers versus non-smokers among young (𝒴) and old (𝒜) donors. Source data are available online for this figure.

has a beneficial effect in reducing lung fibrosis in mice (Lehmann et al, 2017). Although only 0.27% of AT2 cells in healthy HLCA samples were identified as senescent (Fig. 3), their transcriptional changes after senescence induction were among the most pronounced, with most SenSet genes upregulated after senescence induction in human PCLS. These results support the notion that AT2 cells are particularly sensitive to exposures and heavily involved in the injury response. In contrast, fibroblasts were the main cell type with the most downregulated SenSet genes after senescence induction in PCLS (Fig. 7).

Several genes were consistently upregulated across fibroblasts[(−)], basal[(−)] cells, and AT2[(−)] cells, emphasizing their shared senescence-associated pathways. Among the 19 upregulated genes in fibroblasts[(−)], 8 genes are also upregulated in basal[(−)] cells and/or in AT2[(−)] cells such as *TNFRSF1A*, *YPEL3*, *SPOP*, *ZFP36*, *CITED2*, and *MARCKS* (Fig. 3G). Of these, *TNFRSF1A* encodes a receptor mediating inflammatory cytokine production, while *YPEL3*, a p53 downstream gene, is known to induce senescence (Kelley et al, 2010). *SPOP*, a tumor suppressor frequently mutated in cancers, was also upregulated and has been implicated in senescence induction and myofibroblast activation (Zhu et al, 2015; Yang et al, 2023). *ZFP36*, a gene regulating inflammatory cytokine production

in psoriatic skin and metabolic pathways by growth factor induction (Angiolilli et al, 2022; Cicchetto et al, 2023), showed consistent expression patterns across SnCs (Fig. 3A), reflecting its broad involvement in cellular stress responses. *ZFP36* is known to be induced in cellular senescence in fibroblasts and in different human tissues (Xu et al, 2022). Similarly, *CITED2*, which modulates TGF-β signaling (Chou et al, 2012), was associated with cellular proliferation and senescence, with its reduced expression affecting aging and senescence in tendon-derived stem cells (Hu et al, 2017). *MARCKS*, an actin-binding protein involved in cell motility and secretion, was also highly expressed in these cells.

A total of 19 genes were consistently downregulated in both fibroblasts[(−)] and basal[(−)] cells. For instance, *IL32*, a pro-inflammatory cytokine, was reduced in both, aligning with its role in regulating survival, proliferation, and mitochondrial metabolism in myeloma cell lines (Aass et al, 2022). *YBX1*, which regulates some SASP factors and senescence markers in human primary keratinocytes (Kwon et al, 2018), displayed similar downregulation in lung fibroblasts and basal cells.

In contrast, some genes were downregulated in fibroblasts[(−)] while being upregulated in other cell types, highlighting cell-type-specific senescence responses. For instance, *PLAUR*, involved in

fibroblast-to-myofibroblast differentiation (Bernstein et al, 2007), was downregulated in fibroblasts[(−)] but upregulated in basal[(−)] cells. Interestingly, *PLAUR* was identified as an upregulated gene in datasets from murine and human SnCs. *PLAUR* encodes for urokinase plasminogen activator receptor (uPAR), and treatment with uPAR-directed CAR T cells was a good senolytic strategy to decrease SnCs in vivo and in vitro (Amor et al, 2020). Other genes included *ID2*, which is known to antagonize the growth-suppressive activities of p16 and p21 (Lasorella et al, 1996), and *TNFAIP3*, which encodes for TNF-$\alpha$-induced protein 3 (A20). In mice, fibroblast A20 deletion recapitulates major pathological features of systemic sclerosis (Wang et al, 2022b). Altered expression of A20 in hematopoietic stem and progenitor cells leads to an aging-like phenotype, potentially impairing their functional capacity (Smith et al, 2020). *IGFBP4* encodes for insulin-like growth factor binding protein 4 and is downregulated in fibroblasts[(−)] and upregulated in AT2[(−)] cells. IGFBP-4 was induced with irradiation in mice and humans and showed a pro-aging effect (Alessio et al, 2020).

*TXNIP*, associated with oxidative stress responses, was upregulated in both basal[(−)] cells and AT2[(−)] cells. *TXNIP* was shown to have a role in cellular senescence; its expression increases with age in $\beta$-cells and serum samples from humans, and it aggravates age-related and obesity-induced structural failure associated with an induction of cell cycle arrest and oxidative stress (Li et al, 2023b). Furthermore, *TXNIP* deletion induces premature aging in hematopoietic stem cells, inhibiting p38 (Jung et al, 2016). Other genes included *FOS* and *JUN*, which are proto-oncogenes widely recognized for their involvement in the cellular senescence process (Seshadri and Campisi 1990; Weitzman et al, 2000).

Among secreted proteins measured in the supernatants, GDF-15 exhibited the largest increase (Fig. 5H), aligning with its established role as an age-associated and senescence marker (Conte et al, 2022; Basisty et al, 2020; Schafer et al, 2020). Validation of the SenSet gene list in an ex vivo model confirmed the induction of hallmark senescence markers, including p21. While *CDKN1A* (p21) showed robust increases after senescence induction, an increase in *CDKN2A* (p16) mRNA remained largely undetectable in our transcriptomic data across both the HLCA and the ex vivo model (Figs. EV5 and EV12). Previous studies have similarly reported increased *CDKN1A* but not *CDKN2A* transcript levels in SnCs from complex tissues (De Man et al, 2026). These findings highlight the difficulty of detecting *CDKN2A* mRNA in tissue-level single-cell datasets, despite its established role in cell culture models of senescence. Supporting this, we observed an increase in p16 protein levels in PCLS after bleomycin treatment (Fig. EV10C), suggesting that *CDKN2A* may be regulated post-transcriptionally or induced in ways not captured at the RNA level.

Several SenSet genes were validated in our model, including *AAK1*, which induces several SASP factors (Ferrand et al, 2015); *ALDH1A1*, linked to SASP regulation and senescence in ovarian cancer stem cells (Muralikrishnan et al, 2022); and *PLK2*, a kinase implicated in senescence pathways with reduced expression in glioblastoma (Ding et al, 2022). *CNN2*, known to promote fibroblast senescence (Jun and Lau, 2017), further supported the relevance of these markers in defining SnCs. Interestingly, *TXNIP*, which was upregulated in 11 cell types[(−)] in HLCA, was downregulated in the ex vivo model, indicating its inducer-specific role in oxidative stress-mediated senescence. Similarly, genes such as *NME2* and *NPM1*, which play a role in cell cycle and tumor

suppression (Liu et al, 2015; Box et al, 2016), were consistently downregulated in both datasets. This finding aligns with a prior study that *NPM1* upregulation inhibits p53-mediated senescence (Wong et al, 2013).

Our findings underscore how environmental factors interact with aging processes to drive senescence in specific lung cell types. Smoking increases the production of reactive oxygen species (ROS), which induces cellular damage and oxidative stress, triggering inflammation and cellular senescence. These processes are believed to contribute to abnormal lung injury responses, impaired tissue repair, and increased susceptibility to chronic lung diseases such as COPD and pulmonary fibrosis (Wang et al, 2025; Eckhardt and Wu, 2021). A comparative analysis of smoker and non-smoker groups within the HLCA revealed that smoking may accelerate aging in the lung. Notably, we shed light on major cell types that are likely mechanistically involved in these processes. Basal cells showed an upregulation of ~80% of all SenSet genes, positioning them as primary responders to senescence in smokers.

Despite these results, our study has some limitations that need to be considered. The assumption that cells from individuals younger than 30 are universally healthy may not hold true in all cases. However, since our setup is a probabilistic framework, we do not need to assume that there are no senescent cells in younger lungs; only that the probability to encounter such cells is much lower. It is well-established that senescence increases in multiple organs with age (Yousefzadeh et al, 2021; Mebratu et al, 2023; Calhoun et al, 2016). Furthermore, our probabilistic labeling tolerates some "contamination" by true senescent cells, which is supported by the fact that the PUc classifier identifies some SnCs for several cell types in $\mathcal{Y}$ (Fig. EV3). In addition, the use of arbitrary age thresholds (30 and 50) could influence the PUc classifier, although the framework remains valid as long as healthy cells in patients younger than 50 are similar. Recent studies have shown that significant developmental dysregulation occurs at the ages of 44 and 60 (Shen et al, 2024), leading us to believe that most healthy cells in this age range (30–50) are similar to those in the youngest group. The upper threshold was set to 50 in order to expand the set of individuals for this age group. Furthermore, the identification of marker genes is performed using one group only (ages 50 + ), and not by comparing age groups against each other, so this likely does not constitute a major limitation of the subsequent analysis.

Some cell types, such as tracheobronchial serous cells exhibited an unusually large fraction of SnCs (76%) as assigned by the PUc learner (Fig. 3A). Serous cells were the least represented population in old tissue ($n = 97$, Fig. EV2), followed by acinar cells ($n = 250$, 49% SnCs), leading us to suspect that for cell types with very few cells in this age group, the method likely overestimates senescence proportions. We thus recommend using the SenSet set for cell types with a larger number of cells. Furthermore, only one marker was assigned to serous cells, limiting its contribution to SenSet.

To generate SenSet, we selected genes that were differentially expressed across the majority of cell types. The rationale for this choice is that, even if certain cell types contained a relatively high proportion of false-positive senescent cells, their influence on the final gene set would be limited. This approach comes with a tradeoff as prioritizing consistency across cell types may reduce sensitivity to cell-type-specific senescence markers. However, we found that for 14 of the 22 analyzed cell types, ~80%+ of the cell-

type-derived markers were retained in SenSet, indicating somewhat similar inclusion rates across cell types (Fig. EV19). While the remaining excluded genes may still carry biologically relevant information, our strategy favors robustness and stronger statistical guarantees over maximal cell-type specificity. Finally, only 18 of the 31 cell types analyzed using the PUc framework contained no SnCs in $\mathcal{Y}$, suggesting that the method is robust to a small number of SnCs in the young.

Finally, while we identify cell-type-specific senescence-associated gene signatures, we do not experimentally dissect the underlying mechanisms driving these differences. Future work should focus on mechanistic studies to validate and functionally characterize these cell-type-specific pathways.

In conclusion, this study advances our understanding of SnCs in the healthy human lung by identifying cell type-specific senescence markers, applying a robust machine learning framework to a large aging cohort. The validation of these findings in ex vivo models with perturbed senescence strengthens their relevance, offering a foundation for future research into the role of SnCs in normal aging and accelerated aging in the context of chronic lung diseases, such as COPD. By linking cellular senescence to environmental stressors like smoking, this work highlights potential targets for therapeutic interventions aimed at mitigating the detrimental effects of senescence on lung health.

# Methods

### Reagents and tools table

| Reagent/ resource | Reference or source | Identifier or catalog number |
|---|---|---|
| DMEM/F-12 with HEPES | Thermo Fisher Scientific | Cat 12400024 |
| Penicillin–Str-eptomycin | Merck MilliporeSigma, Sigma-Aldrich | Cat P00781 |
| Amphotericin B solution | Merck MilliporeSigma, Sigma-Aldrich | Cat A2942 |
| DMSO | Merck MilliporeSigma, Sigma-Aldrich | Cat D2438 |
| **Experimental models** | | |
| Precision-cut Lung Slices (PCLS) | The study was approved by the local ethics committee of the Ludwig-606 Maximilians University of Munich, Germany (Ethic vote #19-630) and University of Pittsburgh 607 (IRB PRO14010265). Written informed consent was obtained for all study participants. Other human lung tissue for hPCLS samples were collected by The Ohio State University (OSU) Comprehensive Transplant Center (CTC) Human Tissue Biorepository, which adheres to ISBER Best Practices for Biorepositories. Informed consent was obtained from IPF patients under the Total Transplant Care Protocol (IRB protocol # 2017H0309), and donor research authorization was secured by our local organ procurement agency, Lifeline of Ohio. Samples were sent to our laboratory by the CTC Biorepository through an Honest Broker process (IRB protocol # 20170310), under a secondary research IRB protocol # 2021H0180. | |

| Reagent/ resource | Reference or source | Identifier or catalog number |
|---|---|---|
| **Recombinant DNA** | | |
| **Antibodies** | | |
| Anti-p21 antibody [EPR362] | Abcam | Cat ab109520 |
| Anti-p21 Waf1/Cip1 antibody (12D1) | Cell Signaling | Cat 2947 |
| Anti-ARPC5/ p16 ARC antibody [EP1551Y] | Abcam | Cat ab51243 |
| Anti-Fibronectin antibody | Abcam | Cat ab2413 |
| Anti-alpha smooth muscle Actin antibody | Abcam | Cat ab21027 |
| Anti-alpha smooth muscle Actin antibody | Santa Cruz Biotechnology | Cat sc-32251 |
| Anti-Ki67 antibody (8D5) | Cell Signaling | Cat 9449 |
| Anti-GAPDH antibody | Abcam | Cat ab9485 |
| Anti-β-Actin—Peroxidase antibody | Sigma-Aldrich | Cat A3854 |
| Anti-rabbit IgG 647 Secondary antibody | Biotium | Cat 20047 |
| Anti-mouse IgG 750 Secondary antibody | Biotum | Cat 20463 |
| Anti-rabbit red (IRDye® 680RD Donkey anti-Rabbit IgG Secondary antibody) | Li-COR Bio | Cat 926-68073 |
| Anti-rabbit green (IRDye® 800CW Goat anti-Rabbit IgG Secondary antibody) | Li-COR Bio | Cat 926-32211 |
| Anti-rabbit IgG, HRP-linked Antibody | Cell Signaling | Cat 7074 |

| Reagent/ resource | Reference or source | Identifier or catalog number |
|---|---|---|
| Anti-mouse IgG, HRP-linked Antibody | Cell Signaling | Cat 7076 |
| Donkey anti-Rabbit IgG (H + L) Highly Cross-Adsorbed Secondary Antibody, Alexa Fluor™ 647 | Invitrogen | Cat A-31573 |
| Donkey anti-Mouse IgG (H + L) Highly Cross-Adsorbed Secondary Antibody, Alexa Fluor™ 488 | Invitrogen | Cat A-21202 |
| **Oligonucleotides and other sequence-based reagents** | | |
| PCR primers | IDT | See Table below |
| **Chemicals, enzymes, and other reagents** | | |
| Bleomycin | Fresenius Kabi | Cat 10361 |
| Doxorubicin | Merck MilliporeSigma, Sigma-Aldrich | Cat D1515-10MG |
| DAPI | Thermo Fisher Scientific | Cat D1306 |
| iTaq Universal Probes One-Step Kit | Bio-Rad | Cat 1725141 |
| **Software** | | |
| OLYMPUS OlyVIA 3.1 software | Olympus Soft Imaging Solutions GmbH | |
| FIJI version 2.9.0 | | |
| Image Studio version 6.0 | | |
| **Other** | | |

**Table. List of PrimeTime® primer assays (Integrated DNA Technologies).**

| Gene | Assay IDT |
|---|---|
| CDKN2A | Hs.PT.58.14776964.g |
| CDKN1A | Hs.PT.58.40874346.g |
| IL-6 | Hs.PT.56a.40226675 |
| GDF15 | Hs.PT.58.40089589 |
| FGF2 | Hs.PT.56a.24613308 |
| TGFB1 | Hs.PT.56a.39813975 |
| COL1A1 | Hs.PT.56a.15517795 |

| Gene | Assay IDT |
|---|---|
| COL3A1 | Hs.PT.58.4249241 |
| FN1 | Hs.PT.58.40005963 |
| ACTA2 | Hs.PT.56a.2542642 |

## The Human Lung Cell Atlas

The (core) Human Lung Cell Atlas (HLCA) (Sikkema et al, 2023) was downloaded from https://data.humancellatlas.org/hca-bio-networks/lung/atlases/lung-v1-0. Counts were already normalized. The HLCA harmonizes scRNA-seq data from 14 datasets, encompassing 106 individuals aged between 10 and 76 years. We removed one individual for whom the age was not available. While five levels of annotation are available in the data, we used the finest level, assigning one of 50 cell types to over 500,000 cells for the analysis in this study. The dataset also contains individuals with a smoking history, including 19 former and 28 active smokers. Smoking status is not available for 8 individuals, and these were not included in the analysis.

## PCLS culture and senescence induction (bleomycin and doxorubicin)

The protocols for preparing human PCLS and inducing senescence in PCLS through treatment with bleomycin and doxorubicin have been published on protocols.io: https://www.protocols.io/view/human-derived-precision-cut-lung-slices-hpcls-agar-36wgq3xr5lk5/v1 https://www.protocols.io/view/senescence-induction-by-dna-damage-in-pcls-e6nvwdwpwlmk/v1. Lungs from healthy donors (ages 20–78) were collected, and the lower left lobes were inflated with 2.5% agarose in DMEM/F-12 with HEPES (Thermo Fisher Scientific Cat #12400024). After 45 min on ice, the lobes were sliced, and 1-cm diameter cores were extracted. Precision-cut lung slices (PCLS) were prepared using a Compresstome® to obtain 300-µm thick tissue slices with a diameter of 1 cm. The PCLS were then placed in a 24-well plate containing 1 mL of DMEM/F-12 medium supplemented with 1% FBS, 1% Penicillin–Streptomycin (Merck MilliporeSigma, Sigma-Aldrich Cat #P00781), and 0.3 µg/mL Amphotericin B solution (Merck MilliporeSigma, Sigma-Aldrich Cat #A2942) and incubated for 24 h at 37 °C with 5% $CO_2$ (day −1).

At 24 h (day 0), 72 h (day 2), and 120 h (day 4), the medium was replaced with fresh medium containing treatments diluted in DMEM/F-12 with 0.1% FBS, 1% Penicillin–Streptomycin, and 0.3 µg/mL Amphotericin B. The PCLS were treated under the following conditions: in PBS as the control, with 15 µg/mL bleomycin (Fresenius Kabi Cat #10361), with DMSO diluted at 1:100,000 in medium (Merck Millipore Sigme, Sigma-Aldrich, Cat #D2438), or with 0.1 µM doxorubicin hydrochloride (Merck MilliporeSigma, Sigma-Aldrich, Cat #D1515-10MG), originally dissolved in DMSO at 10 mM. After 168 h (day 6), the supernatants were collected and frozen at −80 °C for future multiplex immunoassay analysis by Luminex™. The PCLS were snap-frozen in liquid nitrogen for future protein extraction and snRNA-seq using the 10x Genomics™ platform. In addition, PCLS were fixed for 30 min in 4% formaldehyde diluted in PBS (Life Technologies Cat #28908), washed in PBS, and embedded in paraffin. Separate

PCLS were fixed for 30 min in the fixative solution from the β-galactosidase staining kit (Cell Signaling Technology Cat #9860) and then washed in PBS.

## PCLS culture and senescence induction (Irradiation)

Peritumor control tissue from three non-chronic lung diseases (N-CLD) patients and one COPD patient were obtained from the CPC-M bioArchive at the Comprehensive Pneumology Center (CPC Munich, Germany). Patients were two males and two females and had a mean age of 72.5 years old. Human lung tissue was filled with 3% of low gelling temperature agarose in DMEM/F-12 (Thermo Scientific, USA) with phenol red supplemented with 0.1% FCS, 1% P/S, and 1% amphotericin B and kept at 4 °C for at least 1 h. In total, 500 μm PCLS were generated using either a vibratome HyraxV50 (Zeiss, Germany) or 7000smz-2 Vibratome (Campden Instruments, England). The day after slicing, fresh medium was added, and PCLS were exposed to ionizing radiation using the RS225 X-ray cabinet (Xstrahl, Camberley, UK). Dose was calculated according to exposure time (30 Gray (Gy) = 12 min 24 sec) at 195 kV and 15 mA. Then, PCLS were kept in culture for up to 5 days at 37 °C, 5% $CO_2$, and medium was changed every 2–3 days.

## Ethic statement

All studies involving human participants were conducted in accordance with the ethical standards of the institutional research committee and with the 1964 Helsinki Declaration and its later amendments. Informed consent was obtained from all participants prior to inclusion in the study. The study also adhered to the principles outlined in the U.S. Department of Health and Human Services Belmont Report.

The study was approved by the local ethics committee of the Ludwig-Maximilians University of Munich, Germany (Ethic vote 19-630). Written informed consent was obtained for all study participants.

## Histology and immunohistostaining

Paraffin-embedded PCLS were sliced at a thickness of 4 μm. One slide was subjected to H&E staining, and another slide was used for p21 and Ki67 immunostaining. The slides were rehydrated through a series of baths in xylene, followed by 100%, 95%, 85% ethanol, and finally water. The slides were then incubated at 105 °C for 20 min in 1× DAKO high pH buffer (Agilent Technologies, Dako Target Retrieval Solution pH 9 10×, Cat #S236784-2). Then the slides were washed in buffer A from Duolink® In Situ Wash Buffers (MilliporeSigma, Sigma-Aldrich Cat #DUO82049-20L), followed by incubation in 300 mM glycine for 30 min, and then in PBS containing 0.1% Tween and 0.5% Triton for 15 min.

The slides were then incubated with a blocking solution consisting of 2% BSA, 0.1% Triton, and 0.1% Tween at 37 °C for 45 min, followed by an overnight incubation at 4 °C with the primary antibody diluted in blocking solution (anti-p21 antibody [EPR362] Abcam, Cat #ab109520, 1:500 and anti-Ki67 antibody (8D5), Cell Signaling, Cat #9449, 1:600). After washing in buffer A, the slides were incubated for 1 h at room temperature with a 1:1000 dilution of the secondary antibody, anti-rabbit IgG 647; Biotium, Cat #20047 or anti-mouse IgG 750; Biotum, Cat #20463. Following

further washes in buffer B from Duolink® In Situ Wash Buffers, the nuclei were stained with DAPI. The slides were then washed in buffer B and mounted with Fluoroshield Mounting Medium (Abcam, Cat #ab104135). Images were captured using an IX83 Olympus microscope, acquiring the entire PCLS area at 20x magnification. Images were analyzed using Olympus OlyVIA 3.1 (Olympus Soft Imaging Solutions GmbH).

Quantification of p21 and Ki67-positive cells was performed using Fiji (version 2.9.0). Automatic quantification of positive cell number identified by a nuclear p21 or Ki67 signal overlapping with the DAPI signal and normalized to total cell count by image based on the DAPI signal.

No blinding was performed during the experiments or outcome assessment.

## Protein extraction and western blotting

Four PCLS were sonicated three times at 35% amplitude for 10 sec in 200 μL of buffer (TPER buffer, Thermo Scientific, Cat #78510, supplemented with Halt™ Protease and Phosphatase Inhibitor Cocktail, EDTA-free (100X), Thermo Scientific, Cat #1861281, to a final concentration of 1×) and kept on ice. The samples were then homogenized for 15 sec using a Thermo Fisher Scientific homogenizer and centrifuged at $300 \times g$ for 5 min at 4 °C. The supernatants were transferred to new 1.5-mL tubes and centrifuged at 10,000 rpm for 10 min at 4 °C. The supernatants were gently collected, and protein concentrations were quantified in triplicate using the Pierce detergent Compatible Bradford Assay kit (Thermo Scientific, Cat #23246), with 150 μL of reagent and 5 μL of sample per well. A standard curve was generated using the Prediluted Protein Assay Standards BSA Set (Thermo Scientific, Cat #23208). Absorbance was measured at 595 nm using SpectraMax ABS Plus Molecular Devices with SoftMaxPro 7.2.

Protein extracts were denatured in 1× Protein Loading Buffer (Li-COR, Cat #928-40004) containing 0.1 M dithiothreitol (DTT, Sigma-Aldrich, Cat #43816) for 10 min at 95 °C. A total of 10 μg of denatured protein was loaded onto Criterion TGX long shelf-life Precast Gels (4-15%, Bio-Rad, Cat #5671083) using 1× Tris/Glycine/SDS buffer (Bio-Rad, Cat #1610772). The proteins were then transferred onto an Odyssey Nitrocellulose Membrane (Li-COR, Cat #926-31092).

The membrane was then blocked for 1 h at room temperature in a 1:1 mixture of 1× TBS and Intercept Blocking Buffer (Li-COR, Cat #927-70001). The blocked membrane was incubated overnight at 4 °C with p21 antibody (Abcam, Cat #ab109520) diluted 1:1000 or GAPDH antibody (Abcam, Cat #ab9485) diluted 1:2500 in a 1:1 mixture of 1× TBST and Intercept Blocking Buffer.

The membrane was then washed three times for 10 min each with TBST and incubated with secondary antibodies: anti-rabbit red (IRDye® 680RD Donkey anti-Rabbit IgG, Li-COR, Cat #926-68073), diluted 1:20,000 in a 1:1 mixture of 1X TBST and Intercept Blocking Buffer for 1 h at room temperature.

The images were captured using Odyssey CLx Imager LICORbio and analyzed with Image Studio version 6.0 software.

## SASP assessment by multiplex immunoassay

PCLS supernatants were harvested after 6 days in culture and stored at −80 °C. The multiplex immunoassay was performed using

the Luminex™ platform, following the manufacturer's instructions (Bio-Techne).

## Single-nucleus RNA sequencing by 10x Genomics™

PCLS were snap-frozen in cryotubes using liquid nitrogen after 6 days of culture and stored in liquid nitrogen. Nuclei isolation from frozen PCLS follows the protocol at https://doi.org/10.17504/protocols.io.36wgqndb5gk5/v1. Four PCLS per experimental condition were then used for snRNA-seq, following the manufacturer's instructions (10x Genomics™). Some minor individual differences in total-counts-per-cell were found across conditions (Fig. 6A).

## Identifying senescent cells in aged individuals via PU learning under covariate shift

Given the lack of labeled data, supervised algorithms such as SenCID (Tao et al, 2024) cannot be applied to classify SnCs in the HLCA dataset. This limitation necessitates the use of learning paradigms that do not rely on fully labeled training sets. One such approach is positive-unlabeled (PU) learning.

PU learning is a variant of binary classification where the goal is to distinguish between positive and negative samples, with the restriction that only positive samples are seen during the training phase (Bekker and Davis, 2020). More concretely, the real data distribution is a mixture given by

$$p(x) = \alpha p_+(x) + (1 - \alpha)p_-(x)$$

where $\alpha$ is the mixture proportion, and $p_+, p_-$ are the probability density functions of the positive and negative samples, respectively. In traditional binary classification, we are given training data $\mathcal{D}_{tr} \sim p(x)$ and test data $\mathcal{D}_{te} \sim p(x)$. However, in PU learning, $\mathcal{D}_{tr} \sim p_+(x)$ which makes it more challenging. Many methods have been proposed, which typically assume some form of smoothness or separability of the classes or treat the negative samples as noise (Scott, 2015; Garg et al, 2021; Kiryo et al, 2017).

PU learning is a natural fit for our task. Because SnCs cannot yet be labeled with certainty from single-cell transcriptomes, we treat the population of SnCs as negative samples within an unlabeled set. The objective is therefore to train a PU classifier that separates SnCs (negative) from healthy cells (positive). Subsequently, differential expression (DE) analysis between these two groups can be used to identify marker genes.

To build the labeled training set and the unlabeled test set, we rely on the age of the donors, where $\mathcal{D}_{tr}$ models young individuals, and $\mathcal{D}_{te}$ models older ones. To accommodate the PU learning setup, we require the following assumptions to hold: (a) the young group contains few to no SnCs, (b) the negative (non-healthy) class in the senior group consists mostly of SnCs, and (c) healthy cells from older donors follow the same distribution as those from young donors. The last assumption is often violated, as older individuals suffer from other aging hallmarks not related to senescence, such as inflammation, epigenetic alterations, or mitochondrial dysfunction (Li et al, 2023a; Wang et al, 2022a; Amorim et al, 2022). Stated differently:

$$p_+^{young}(x) \neq p_+^{senior}(x).$$

Therefore, we face a covariate shift (Dharani. et al, 2019). In covariate shift, the dependence of the response variable $y$ (i.e., senescence status) on gene expression $x$ is the same for both training and test sets; however, the input distributions may not be:

$$p_{tr}(y|x) = p_{te}(y|x)$$
$$p_{tr}(x) \neq p_{te}(x).$$

To address this, the PU learner needs to accommodate a covariate shift. Here, we rely on the formulation of Sakai & Shimizu which propose an unbiased risk estimator for covariate shift adaptation on PU learning, termed PUc (Sakai and Shimizu, 2019).

PUc assumes we are given three sets of samples: labeled training data $\{x_i^{Ptr}\}_i \sim p_{tr}(x|y = 1)$, unlabeled training data $\{x_j^{Utr}\}_j \sim p_{tr}(x)$, and unlabeled test data $\{x_k^{Ute}\}_k \sim p_{te}(x)$. When covariate shift occurs, the PU risk on the test distribution $p_{te}$ differs from the PU risk on the train distribution $p_{tr}$. PUc addresses this by importance-weighting, where the ratio between test and train densities $w(x) := p_{te}(x)/p_{tr}(x)$ is used to weight each sample during the computation of the risk (Sugiyama et al, 2007, 2012). In this case, the PUc risk becomes

$$R_{PUc}(g) := \alpha \mathbb{E}_{x \sim p_{tr}(x|y=1)} \left[ \widetilde{\ell}(g(x))w(x) \right] + \mathbb{E}_{x \sim p_{tr}(x)}[\ell(-g(x))w(x)],$$

where $g$ is a classifier and $\ell$ is a loss function with $\widetilde{\ell}(x) := \ell(x) - \ell(-x)$. The PUc risk on training data can be shown to be an unbiased estimator of the PU risk on test data. The mixture proportion $\alpha$ is estimated from prior knowledge. Similar to the original work, we employ a linear-in-parameter classifier with a Gaussian kernel basis function. Concretely, we represent the decision function

$$g(x) = \sum_{\ell=1}^{b} \alpha_\ell \phi_\ell(x) \text{ with } \phi_\ell(x) = \exp(-||x - c_\ell||^2/(2\sigma^2)),$$

where the $c_\ell$ are up to $b$ centers randomly sampled from the unlabeled test data and $\sigma$ is selected by cross-validation. For more details on PUc, please refer to Sakai and Shimizu (2019).

## Deriving SenSet from the Human Lung Cell Atlas

During the gene set generation step, we kept only individuals without a smoking history from the HLCA (~300,000 cells) to minimize potential confounding effects on the results. As described in the previous section, the PUc estimator requires three sets of samples: positive training samples, and unlabeled training and test samples. We estimated positive samples from individuals under the age of 30, assuming that the prevalence of SnCs in this group is minimal. The unlabeled training samples were estimated from individuals aged 30–50. Finally, the test samples were obtained from older individuals aged 50 and above, which is when covariate shift occurs. We call these three age groups $Y$, $M$, and $A$, respectively (Fig. 2A,E). Cell types with fewer than 50 cells in any age group were excluded from the analysis.

In PUc learning, inclusion of a middle-aged group containing both senescent and healthy cells is essential, as covariate shift between young and old samples makes it difficult to distinguish senescent from non-senescent cells in aged tissue using only these two extremes. The middle-aged group provides an intermediate anchor between training

**Table 3. Number of cells and genes in the PCLS data after filtering.**

| Sample | Bleomycin | | | | Doxorubicin | | | |
|---|---|---|---|---|---|---|---|---|
| | Control | | Treated | | Control | | Treated | |
| | Cells | Genes | Cells | Genes | Cells | Genes | Cells | Genes |
| LTC 120 | 15,277 | 26,678 | 15,388 | 26,678 | N/A | N/A | N/A | N/A |
| LTC 124 | 10,498 | 25,570 | 12,674 | 25,570 | 11,631 | 23,028 | 11,085 | 23,028 |
| LTC 113 | 9285 | 23,989 | 9760 | 23,989 | 7753 | 21,854 | 7173 | 21,854 |
| LTC 117 | 11,974 | 24,837 | 9720 | 24,837 | 10,825 | 22,296 | 11,772 | 22,296 |

and test distributions, improving calibration and preventing conflation of gradual aging effects with true senescence.

To prepare the data for PUc learning, we first applied principal component analysis (PCA) independently for each cell type. We restricted the gene set used for PCA to the union **U** of all four existing senescence gene sets. By incorporating all known senescence-associated genes, we aim to achieve a "weak" separation of healthy cells and SnCs, which can be leveraged by the PUc learner. The top 10 components were used as training data for the PUc classifier. For all experiments, we set the mixture proportion $\alpha$ to 0.9, based on the prior assumption that approximately 10% of the cells are senescent. However, the estimator was robust to this value and returned percentages in the range 0–40%.

Given the inherent randomness of the algorithms used, some variability across runs is expected. Nonetheless, we observed that ~80-90% of SenSet genes were consistently selected across runs with different random seeds.

## Statistical analysis

Differential expression analysis is performed exclusively on the oldest age group, comparing healthy cells with SnCs. This approach is in contrast with methods that compare old and young individuals, where other aging signatures could introduce confounding factors. By directly comparing these two cell populations within the oldest age group, the analysis is specifically focused on senescence. A two-sided Wilcoxon rank-sum test was used to determine differentially expressed (DE) genes (FDR = 0.05). We tested only genes that belong to **U**. FDR-adjusted $P$ values were obtained using the Benjamini and Hochberg procedure (Benjamini and Hochberg, 1995). Cell types with fewer than 20 SnCs were not considered due to the limited sample size. We selected DE genes that were enriched in at least six cell types (either up- or downregulated), resulting in a set of 106 genes that constitute SenSet. A detailed table of all cell-specific SenSet genes and their test statistics is provided in the Appendix.

GSEA (Subramanian et al, 2005) was performed using the GSEApy package (Fang et al, 2023). To compute the Wasserstein distances in Fig. 8A, we sampled at random 5000 cells for cell types with too many cells to speed up computation. Scanpy was used in some parts of the analysis, including clustering and dimensionality reduction (Wolf et al, 2018).

## Integrating PCLS data and validating SenSet

For the overall PCLS sample comparison presented in Fig. 6B,C, we performed basic cell and gene filtering for all 11 samples. Cells with fewer than 500 total counts and fewer than 400 expressed genes

**Table 4. Number of cells and genes in the irradiation data after filtering.**

| Sample | Control | | Treated | |
|---|---|---|---|---|
| | Cells | Genes | Cells | Genes |
| E170 | 9822 | 23,110 | 8092 | 23,110 |
| E185 | 5651 | 19,924 | 3793 | 19,924 |
| E187 | 2524 | 18,969 | 2284 | 18,969 |
| E196 | 3656 | 19,855 | 1900 | 19,855 |

were excluded. Genes with fewer than 50 total counts were also removed. Dimensions of each sample post-processing are shown in Tables 3, 4. Gene counts between treatment and control samples were compared using a two-sided Wilcoxon rank-sum test (FDR = 0.05) for each gene set.

For the cell-specific analysis, we first integrated the data using scVI (Lopez et al, 2018), focusing on the top 2000 variable genes. We used 2 hidden layers with 1000 nodes each. The dimensionality of the latent space was set to 30. Since we used raw counts for scVI, the gene likelihood was modeled as a zero-inflated negative binomial distribution. Nearest neighbors were computed in the scVI latent space, and clusters were identified via Leiden clustering (Traag et al, 2019). Clusters were manually annotated based on canonical markers of lung cell types (Travaglini et al, 2020). Clusters where no marker was significantly expressed were excluded from the cell-specific analysis. Wilcoxon rank-sum tests were used to assess whether SenSet genes were enriched in treatment samples compared to controls. For this analysis, bleomycin, doxorubicin, and irradiation cells were combined into one group.

For each gene set, we obtain a list of $P$ values for each gene based on the DE test. In order to perform a meta-analysis, we combine these $P$ values for each set using Pearson's method, which emphasizes larger $P$ values. I.e., given a set of $P$ values $\{p_i\}_{i=1}^n$, Pearson's method computes the statistic

$$P := -2\sum_{i=1}^{n} \log(1 - p_i).$$

Under the null hypothesis $H_0: p_i \sim U[0,1], i = [n]$, the test statistic $P$ follows a $\chi^2$ distribution with $2n$ degrees of freedom (Pearson, 1933). The combined $P$ value provides an overall assessment of the enrichment of a gene set within a given cell type.

## Enrichment analysis of mesenchymal stem cells

The scRNA-seq dataset of mesenchymal stem cells (MSCs) undergoing replicative senescence was downloaded from GEO

(GSE200157). We used the two available replicas for this data (rep #2 and rep #3) at time points 1 (day 49) and time point 2 (day 86). For each sample, we filter out genes expressed in less than 50 cells and filter out cells with fewer than 100 expressed genes using Scanpy (Wolf et al, 2018). For each replica, only genes overlapping between the two time points are kept.

To rank genes for GSEA prerank, we compute the log1p fold change of average counts. I.e., if we denote the raw count vectors for gene $g$ by $\mathbf{x}^{(T1)} \in \mathbb{R}^{c_1}$ and $\mathbf{x}^{(T2)} \in \mathbb{R}^{c_2}$, we compute the score for $g$ as

$$g_{\text{score}} := \frac{1}{c_2} \sum_{i=1}^{c_2} \log\left(\mathbf{x}_i^{(T2)} + 1\right) - \frac{1}{c_1} \sum_{i=1}^{c_1} \log\left(\mathbf{x}_i^{(T1)} + 1\right).$$

GSEApy was then applied to all five gene sets to obtain an FDR $q$ value.

### Enrichment analysis of the IPF atlas

The scRNA-seq IPF atlas was downloaded from GEO (GSE136831). Similar to the MSC data, we filter out genes expressed in fewer than 50 cells and cells with fewer than 100 expressed genes. This reduces the data to 147,169 IPF cells and 96,303 control cells, while leaving 33,435 genes. Also, like before, we compute gene scores for preranked GSEA based on the difference of average log1p counts. We perform preranked GSEA individually for each cell type, dropping types with fewer than 30 cells in either the healthy or fibrotic portion.

### Enrichment analysis of the liver cirrhosis dataset

For the liver cirrhosis dataset, we also follow identical preprocessing steps to the IPF dataset. When performing preranked GSEA, we gather all cirrhotic samples and compare them against all control samples jointly.

## Data availability

The snRNA-seq data from this publication have been deposited to the SenNet portal at https://sennetconsortium.org/ and assigned the digital object identifier https://doi.org/10.60586/SNT354.RMQR.958. The code used for the experiments in this paper is available at https://github.com/euxhenh/SenSet. The code for PUc learning is included with permission from the author Tomoya Sakai and obtained from https://github.com/ZaydH/arbitrary_pu.

The source data of this paper are collected in the following database record: biostudies:S-SCDT-10_1038-S44318-026-00762-8.

## Peer review information

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

## Acknowledgements

We gratefully acknowledge the provision of human biomaterial and metadata from the CPC-M bioArchive and its partners at the Asklepios Biobank Gauting, the LMU Hospital, and the Ludwig-Maximilians University Munich. In addition, we thank the Center for Organ Recovery & Education (CORE). We are grateful to all Donors and their families for their support. This research was supported by grants from the National Institutes of Health, including NIHU54AG075931 and U24CA268108. We further acknowledge funding from the Hevolution Network on Senescence in Aging. This research was supported in part by the University of Pittsburgh Center for Research Computing and Data, RRID:SCR_022735, through the resources provided. Specifically, this work used the HTC cluster, which is supported by NIH award number S10OD028483.

## Author contributions

**Euxhen Hasanaj:** Conceptualization; Resources; Data curation; Software; Formal analysis; Investigation; Visualization; Methodology; Writing—original draft; Writing—review and editing. **Delphine Beaulieu:** Conceptualization; Resources; Data curation; Formal analysis; Investigation; Visualization; Methodology; Writing—original draft; Writing—review and editing. **Cankun Wang:** Resources; Data curation; Formal analysis; Investigation; Writing—review and editing. **Qianjiang Hu:** Resources; Data curation; Formal analysis; Investigation; Writing—review and editing. **Lorena Rosas:** Resources; Data curation; Formal analysis; Investigation; Writing—review and editing. **Marta Bueno:** Resources; Investigation; Writing—review and editing. **John C Sembrat:** Resources; Investigation; Writing—review and editing. **Ricardo H Pineda:** Resources; Investigation; Writing—review and editing. **Maria Camila Melo-Narvaez:** Resources; Investigation; Writing—review and editing. **Nayra Cardenes:** Resources; Investigation; Methodology; Writing—review and editing. **Zhao Yanwu:** Resources; Formal analysis; Investigation; Writing—review and editing. **Zhang Yingze:** Resources; Formal analysis; Investigation; Writing—review and editing. **Robert Lafyatis:** Resources; Data curation; Investigation; Writing—review and editing. **Alison Morris:** Resources; Investigation; Writing—review and editing. **Ana Mora:** Resources; Funding acquisition; Investigation; Methodology; Writing—review and editing. **Mauricio Rojas:** Resources; Investigation; Methodology; Writing—review and editing. **Dongmei Li:** Resources; Investigation; Methodology; Writing—review and editing. **Irfan Rahman:** Resources; Investigation; Writing—review and editing. **Gloria S Pryhuber:** Resources; Validation; Investigation; Writing—review and editing. **Mareike Lehmann:** Resources; Validation; Investigation; Writing—review and editing. **Jonathan Alder:** Resources; Validation; Investigation; Writing—review and editing. **Aditi Gurkar:** Resources; Validation; Investigation; Writing—review and editing. **Toren Finkel:** Conceptualization; Resources; Funding acquisition; Investigation; Project administration; Writing—review and editing. **Qin Ma:** Resources; Validation; Investigation; Methodology; Writing—review and editing. **Jose Lugo-Martinez:** Resources; Validation; Investigation; Methodology; Writing—review and editing. **Barnabás Póczos:** Conceptualization; Resources; Investigation; Methodology; Writing—review and editing. **Ziv Bar-Joseph:** Conceptualization; Resources; Supervision; Funding acquisition; Investigation; Visualization; Methodology; Writing—original draft; Project administration; Writing—review and editing. **Oliver Eickelberg:** Conceptualization; Resources; Supervision; Funding acquisition; Investigation; Visualization; Writing—original draft; Writing—review and editing. **Melanie Königshoff:** Conceptualization; Resources; Supervision; Funding acquisition; Investigation; Visualization; Methodology; Writing—original draft; Project administration; Writing—review and editing.

Source data underlying figure panels in this paper may have individual authorship assigned. Where available, figure panel/source data authorship is listed in the following database record: biostudies:S-SCDT-10_1038-S44318-026-00762-8.

## Disclosure and competing interests statement

The authors declare no competing interests.

# Expanded View Figures

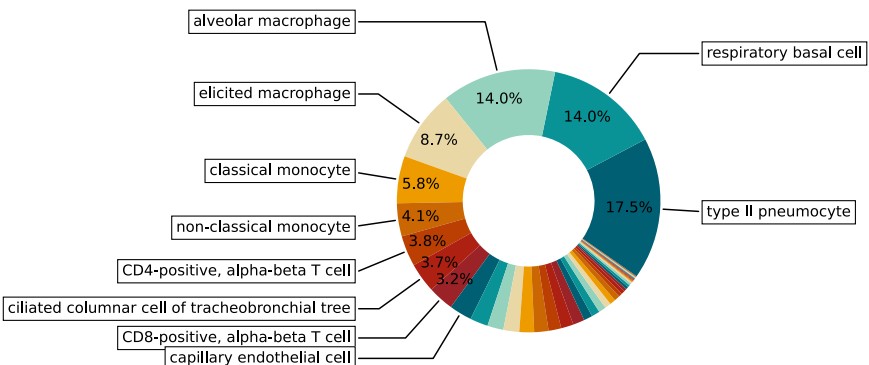

**Figure EV1. Cell-type composition in active smokers.**

Pie chart showing the relative contribution of each lung cell type in active smokers. Labels indicate the exact percentage contribution of each cell type.

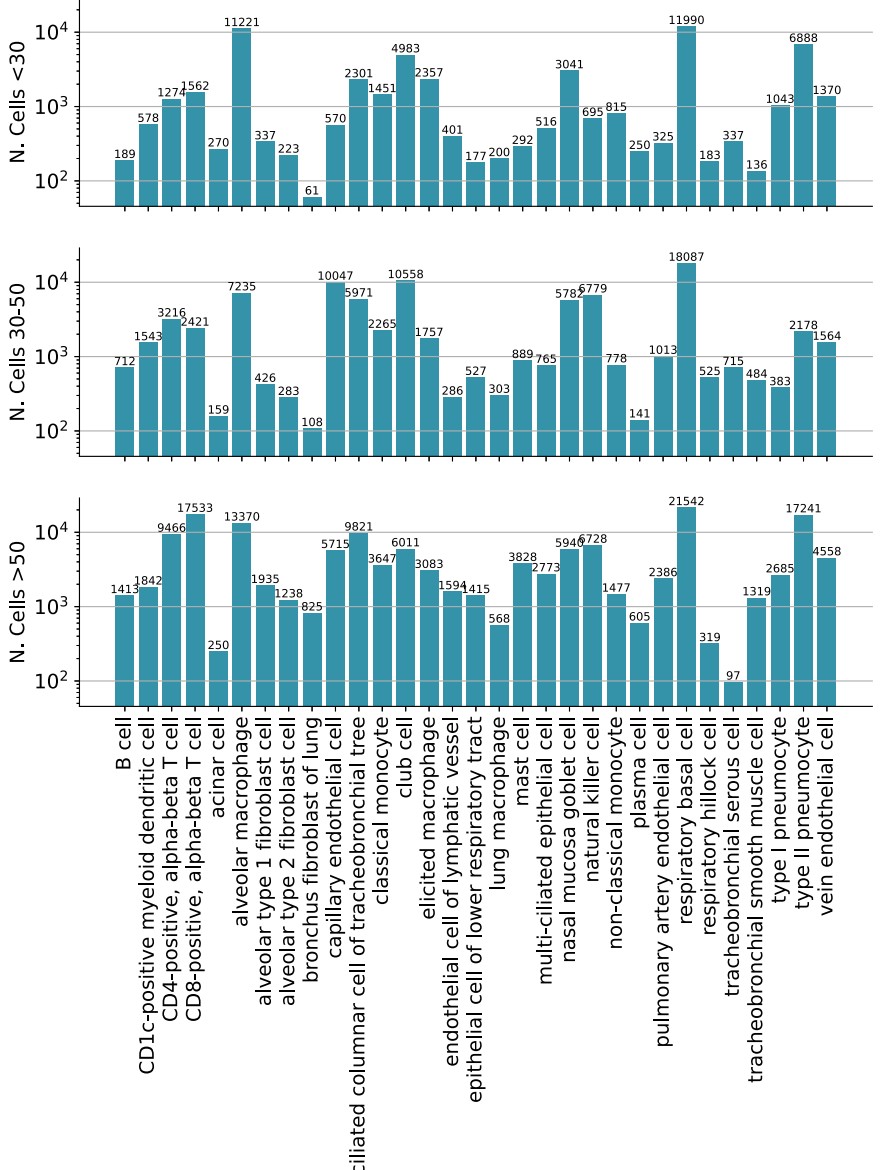

**Figure EV2. Cell counts per cell type and age group in the HLCA (non-smokers only).**

Bar charts showing the number of cells per cell type stratified by donor age group. Separate panels correspond to young (top), middle-aged (middle), and older (bottom) donors.

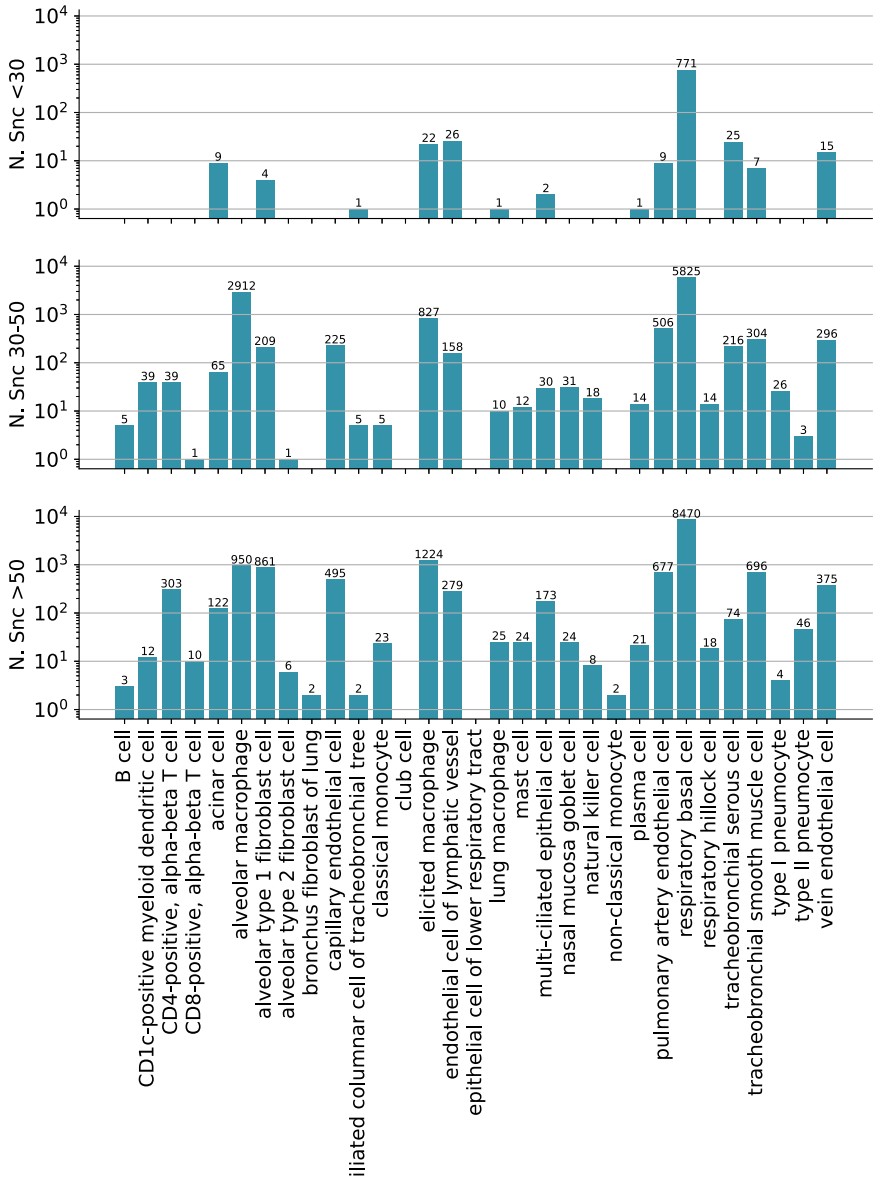

**Figure EV3. PUc-predicted senescent cells by age group in the HLCA.**

Bar charts showing the number of PUc-predicted senescent cells per cell type across age groups. Separate panels correspond to young (top), middle-aged (middle), and older (bottom) donors.

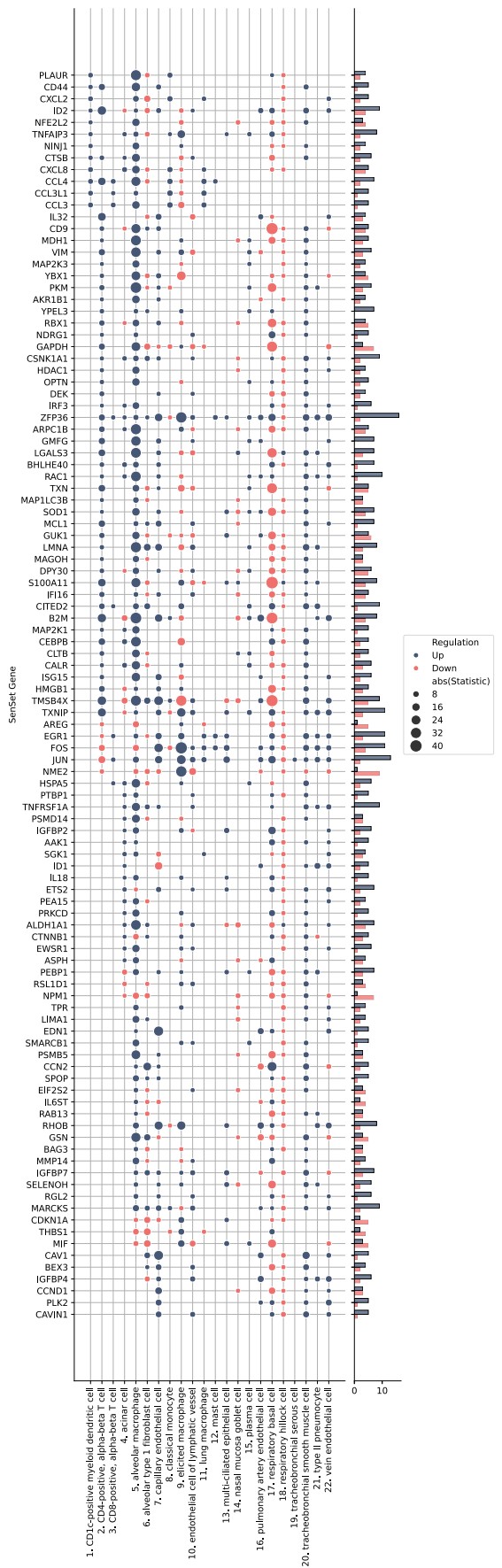

**Figure EV4.  Global enrichment of SenSet genes across lung cell types.**

Heatmap showing Wilcoxon rank-sum test statistics (absolute values) for each SenSet gene (rows) across 22 lung cell types in the HLCA (columns). Color indicates direction of regulation: upregulated (blue) or downregulated (red). Source data are available online for this figure.

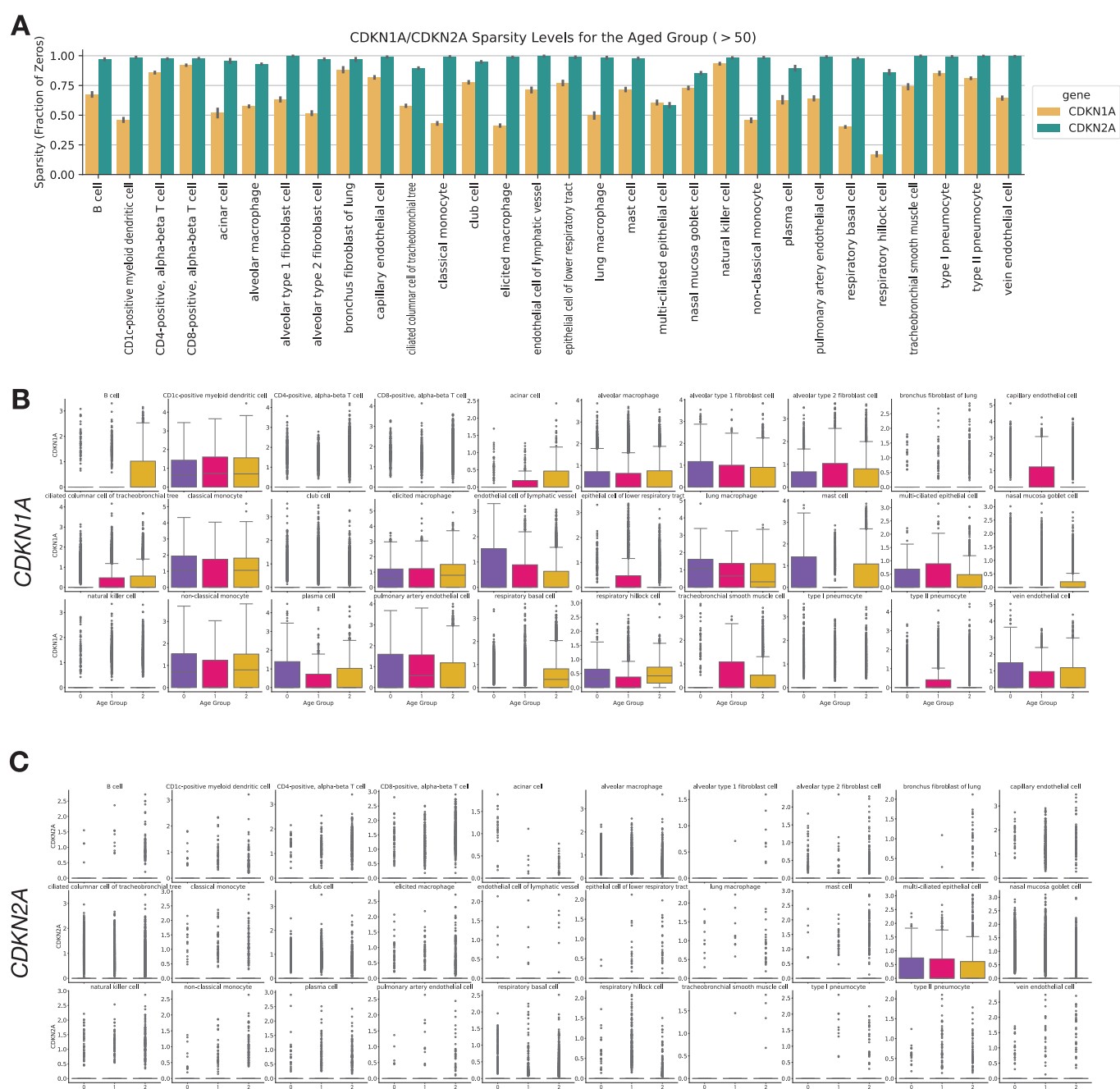

**Figure EV5. CDKN1A and CDKN2A expression across cell types and age groups in the HLCA.**

(A) Sparsity profiles showing the fraction of cells with zero expression of CDKN1A and CDKN2A across lung cell types in the oldest donor cohort. (B) Boxplots of normalized CDKN1A expression by cell type, stratified by young, middle-aged, and older donors. (C) Boxplots of normalized CDKN2A expression by cell type, stratified by the same age groups. Values of *n* shown in Figs. EV2 and 3. Boxplots show median, interquartile range, and whiskers at 1.5× IQR.

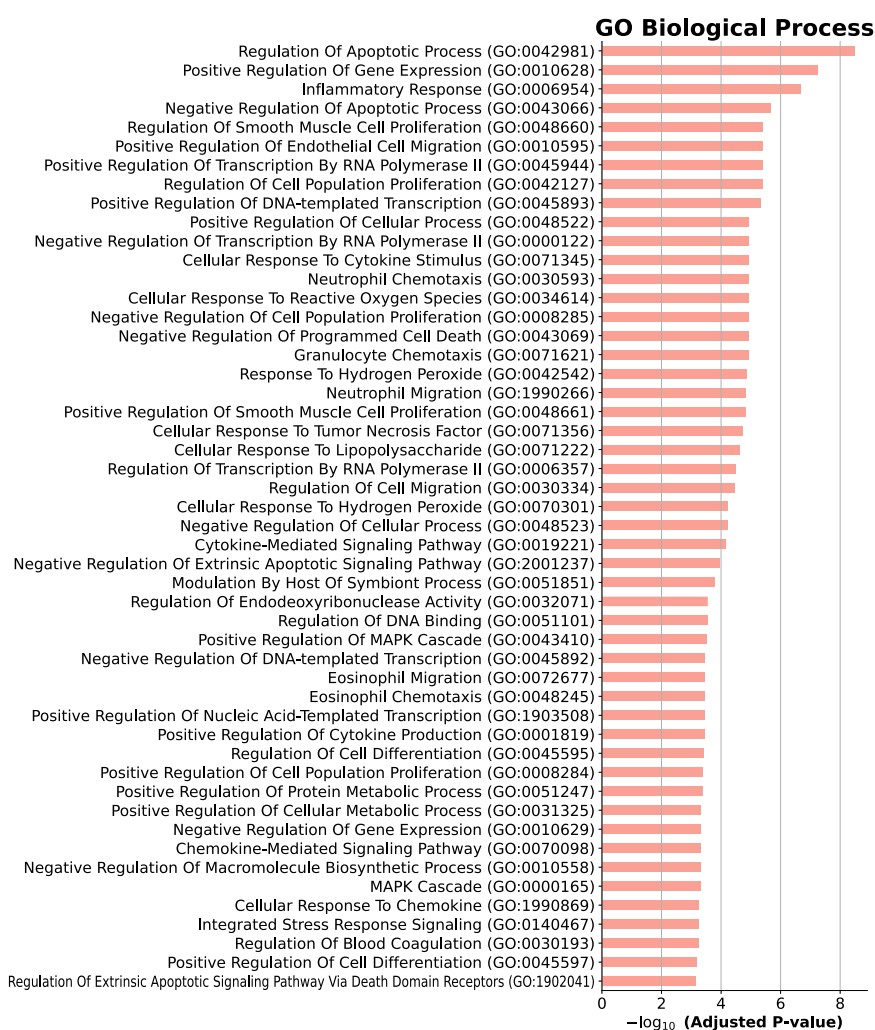

**GO Biological Process**

**Figure EV6. Functional enrichment of SenSet genes.**

Top 50 Gene Ontology (GO) terms enriched among SenSet genes. Adjusted *P* values were obtained using a hypergeometric test as implemented in the GSEApy enrichment module. Source data are available online for this figure.

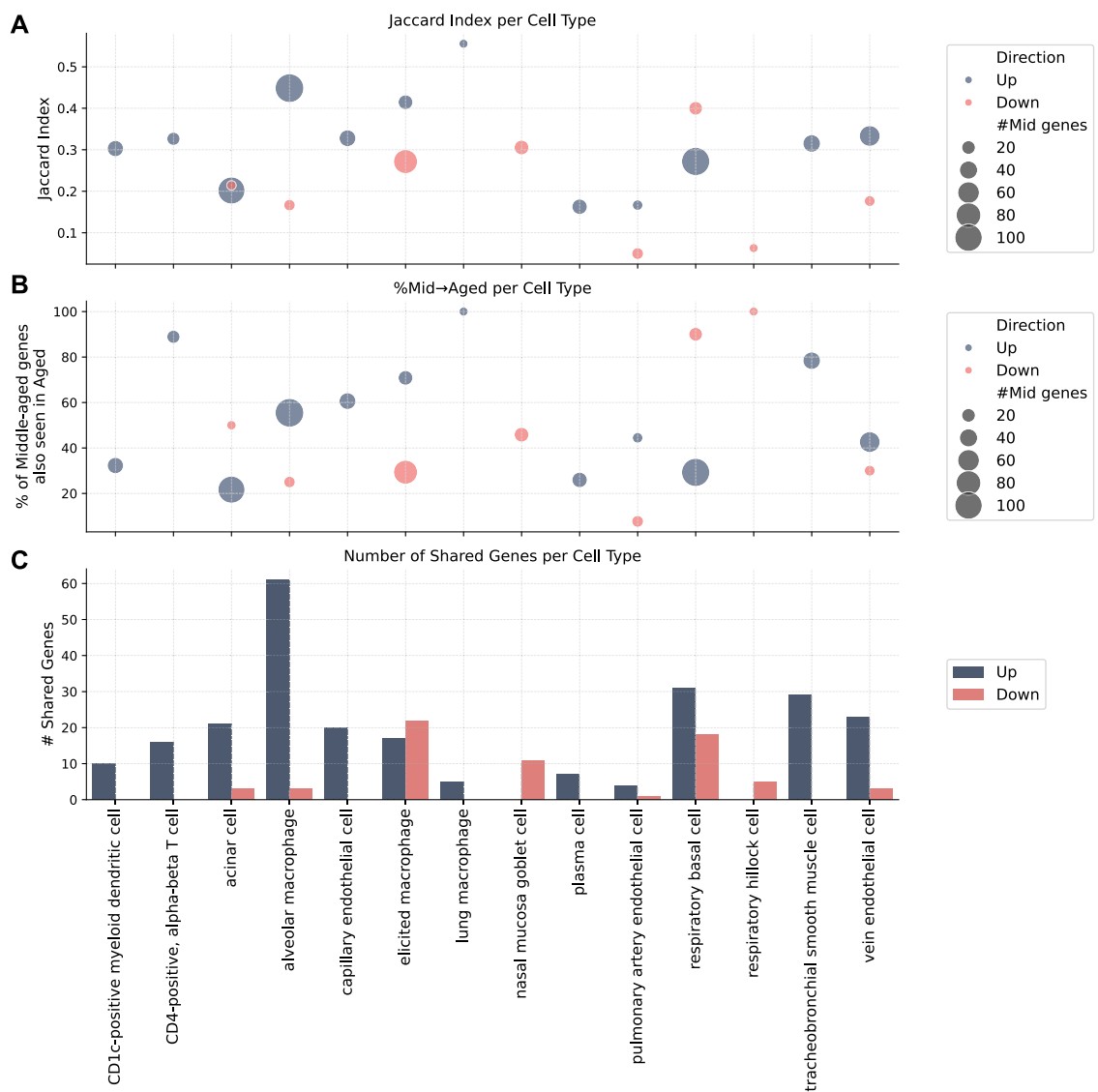

**Figure EV7. Consistency of marker genes between middle-aged and older donors.**

(A) Jaccard index per cell type comparing marker gene sets from middle-aged and older donor cohorts. (B) Fraction of middle-aged cohort markers also present in the older cohort. (C) Absolute number of shared marker genes between the two age groups. Source data are available online for this figure.

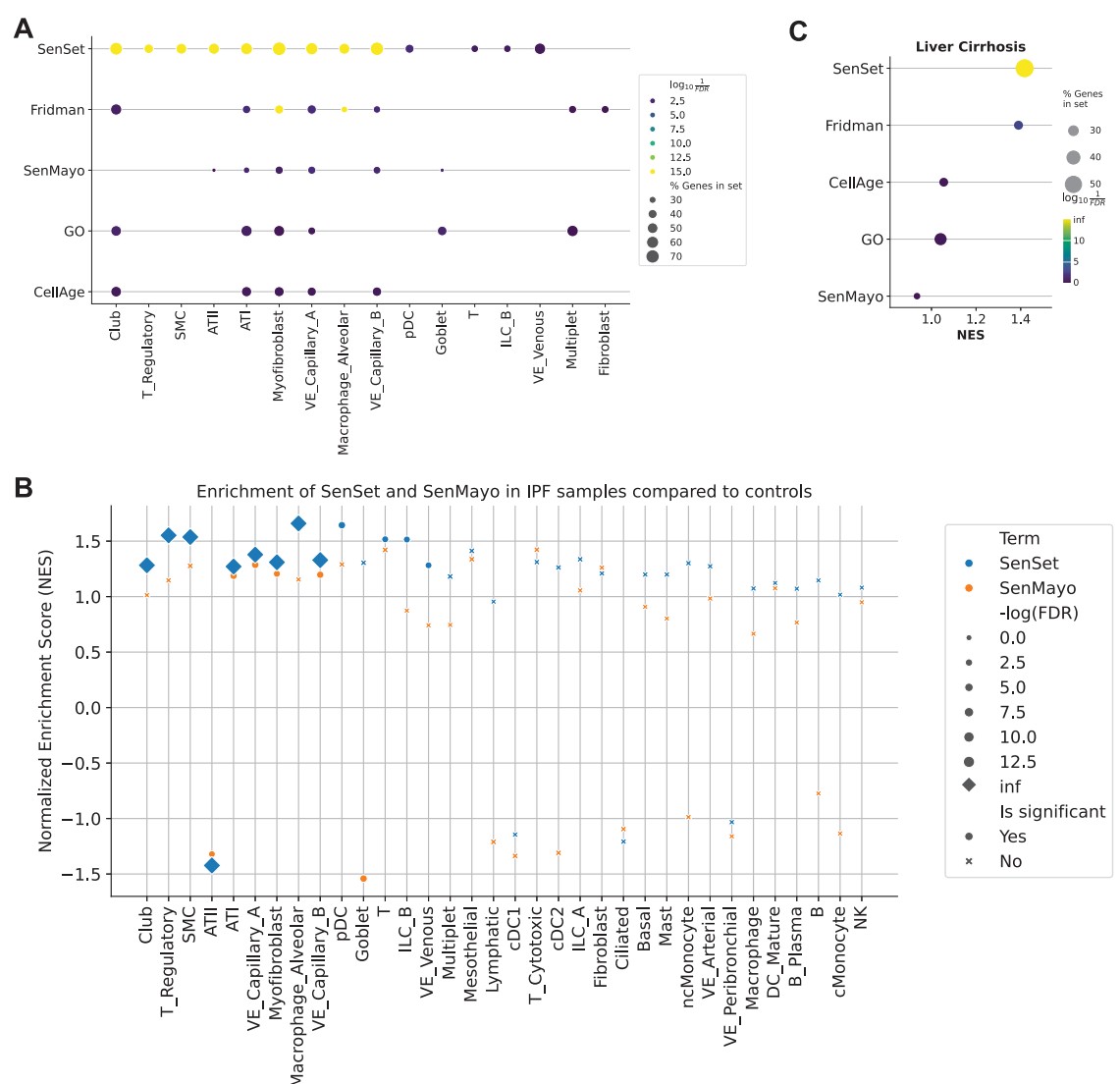

**Figure EV8. Preranked GSEA in IPF and liver cirrhosis cohorts.**

(A) Dot plot showing preranked enrichment results for five senescence gene lists in the idiopathic pulmonary fibrosis (IPF) dataset. (B) Comparative dot plot of normalized enrichment scores (NES) for SenSet versus SenMayo in IPF; diamonds indicate FDR = 0. (C) Preranked enrichment results for the liver cirrhosis dataset, highlighting SenSet enrichment. Source data are available online for this figure.

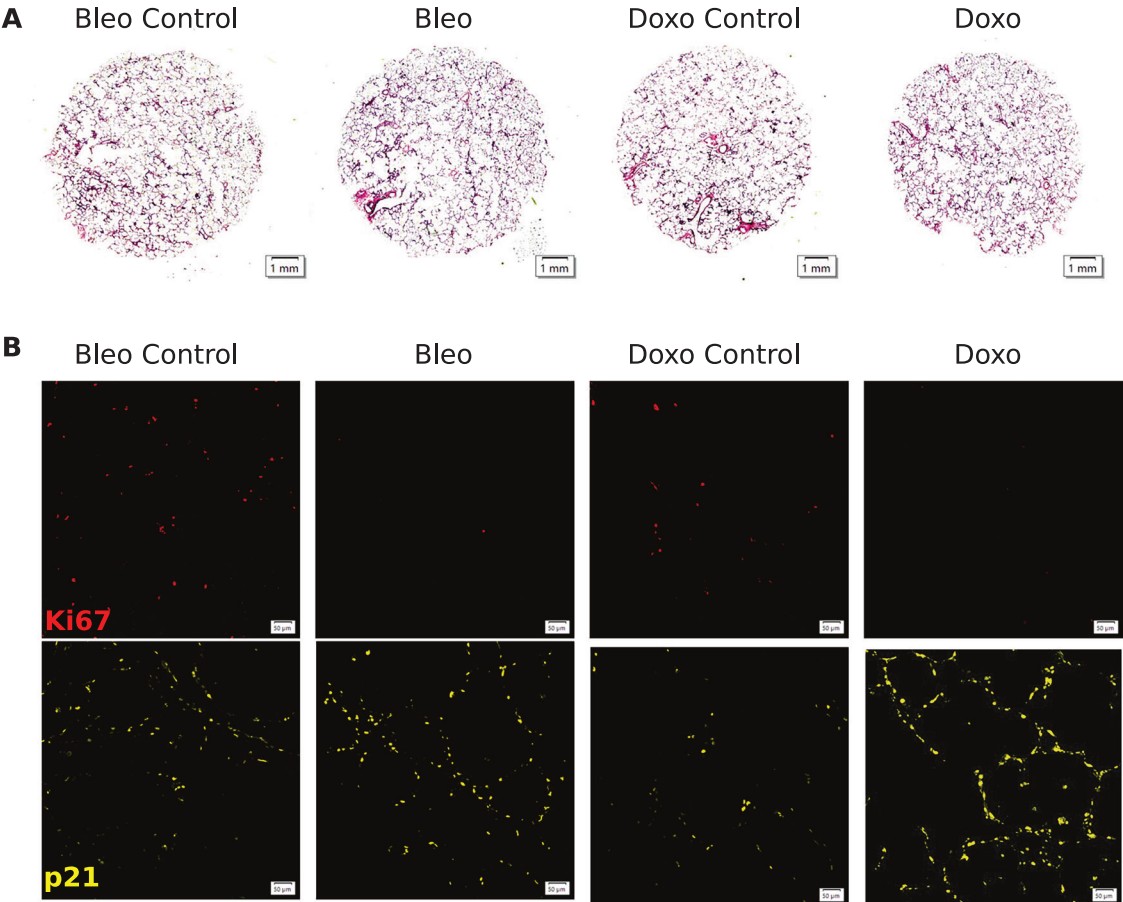

**Figure EV9.   Images from hPCLS treated with bleomycin and doxorubicin for 6 days.**

(A) Hematoxylin and eosin staining of 4 μm formalin-fixed paraffin-embedded human precision-cut lung slices (PCLS) at day 6. (B) Immunohistofluorescence staining for p21 (yellow) and Ki67 (red) on 4-μm PCLS sections at day 6. Source data are available online for this figure.

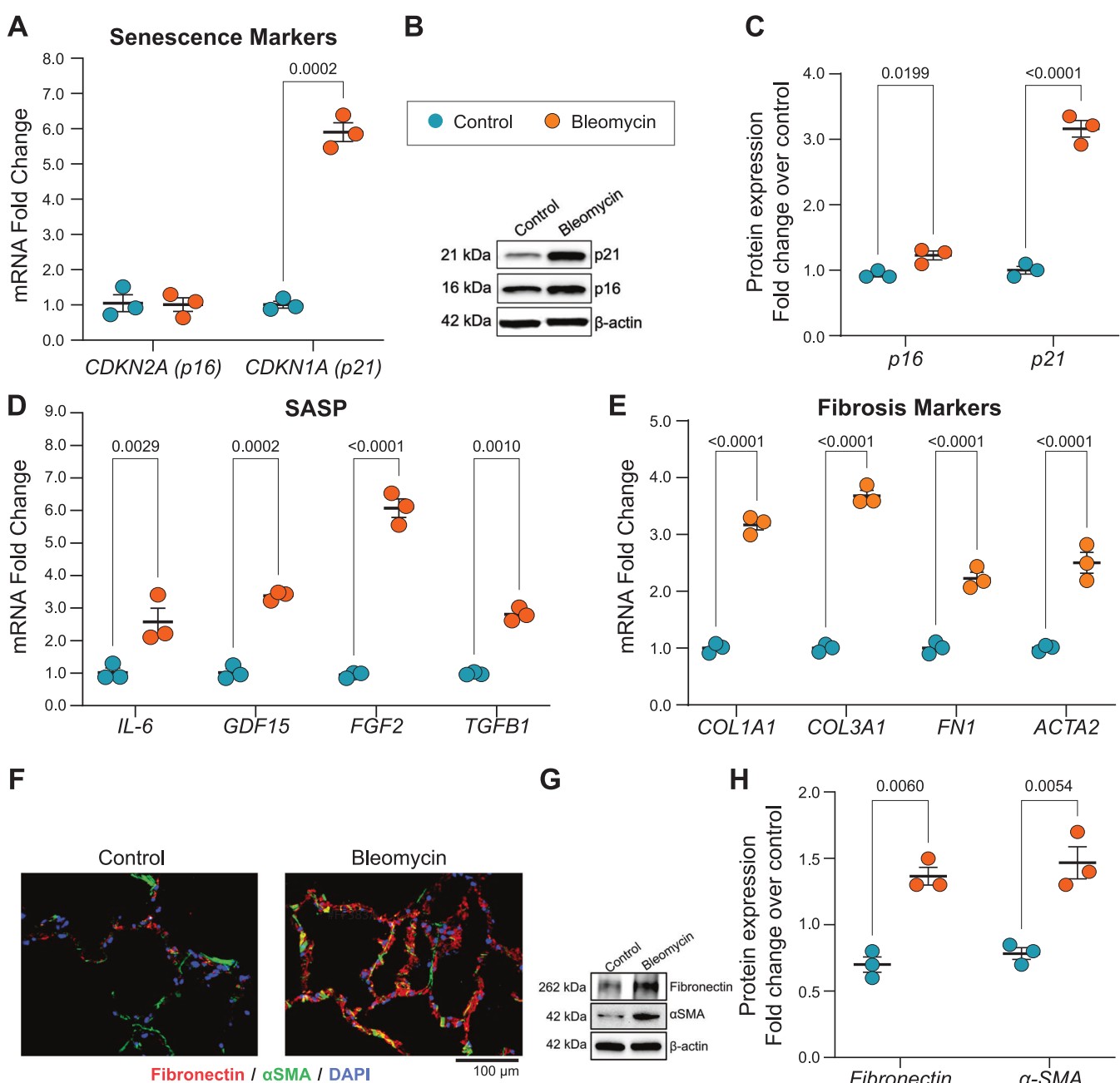

**Figure EV10.** **Validation of senescence and fibrotic responses in bleomycin-treated samples.**

(A) mRNA fold change of CDKN1A and CDKN2A relative to untreated controls ($n = 3$). (B) Representative Western blot for p21 and p16 in control versus bleomycin-treated samples. (C) Quantification of p16 and p21 protein levels normalized to control ($n = 3$). (D) mRNA fold change of key SASP factors following bleomycin exposure ($n = 3$). (E) mRNA fold change of fibrosis-associated genes relative to control ($n = 3$). (F) Immunofluorescence images comparing fibronectin (red) and α-SMA (green) in control versus bleomycin-treated cells. (G) Western blot analysis of fibronectin and α-SMA protein expression in control and bleomycin conditions. (H) Quantification of fibronectin and α-SMA protein fold change normalized to control ($n = 3$). All P values were obtained using a two-way ANOVA followed by Tukey's post hoc test ($P < 0.005$). All error bars represent the standard error of the mean (SEM). Source data are available online for this figure.

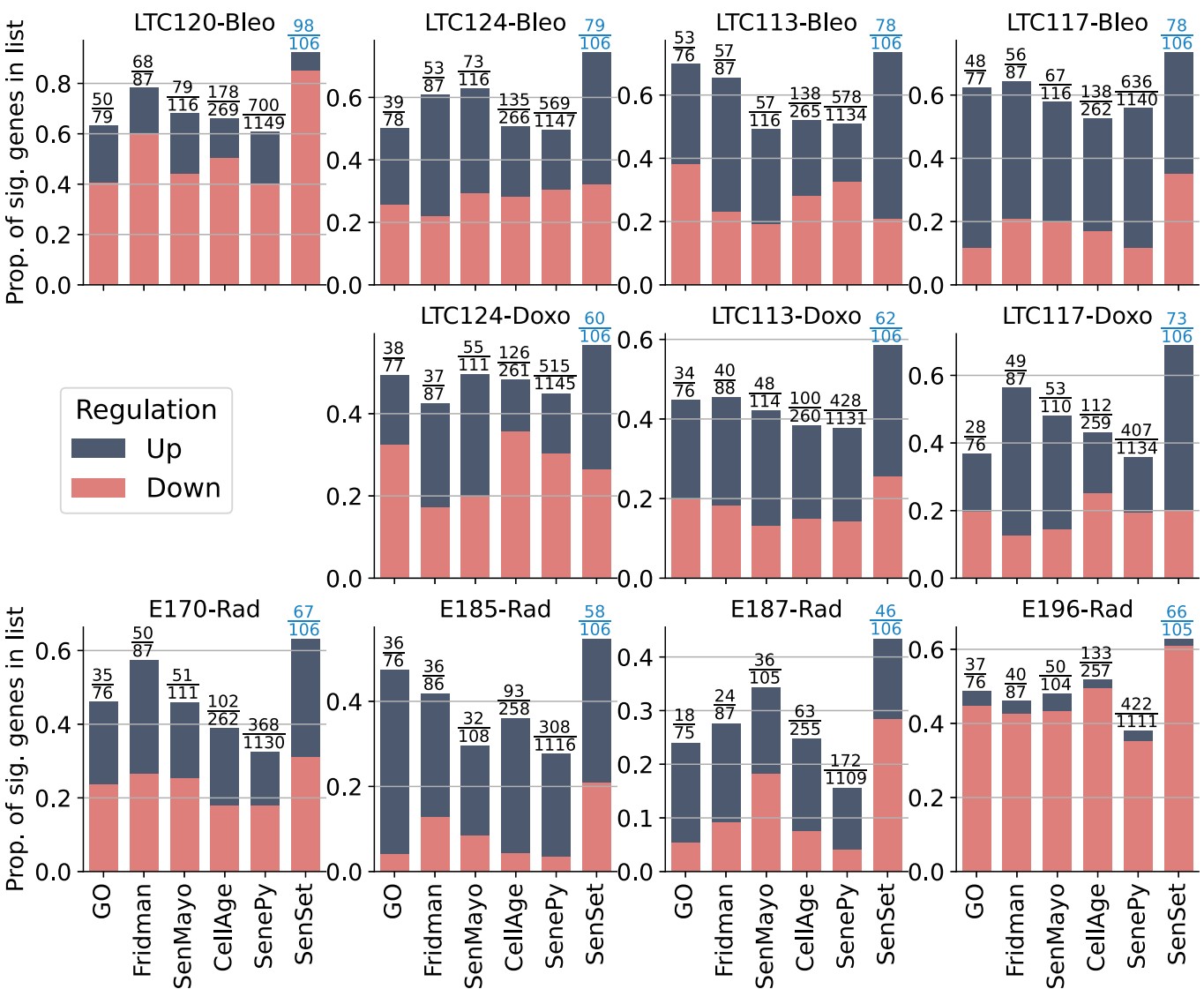

**Figure EV11. SenePy comparison of senescence signatures.**

Per-donor regulation fractions across senescence gene signatures. The *y* axis indicates the fraction of genes in each signature with significant expression changes. Colored segments denote upregulated (blue) and downregulated (red) genes following treatment. Source data are available online for this figure.

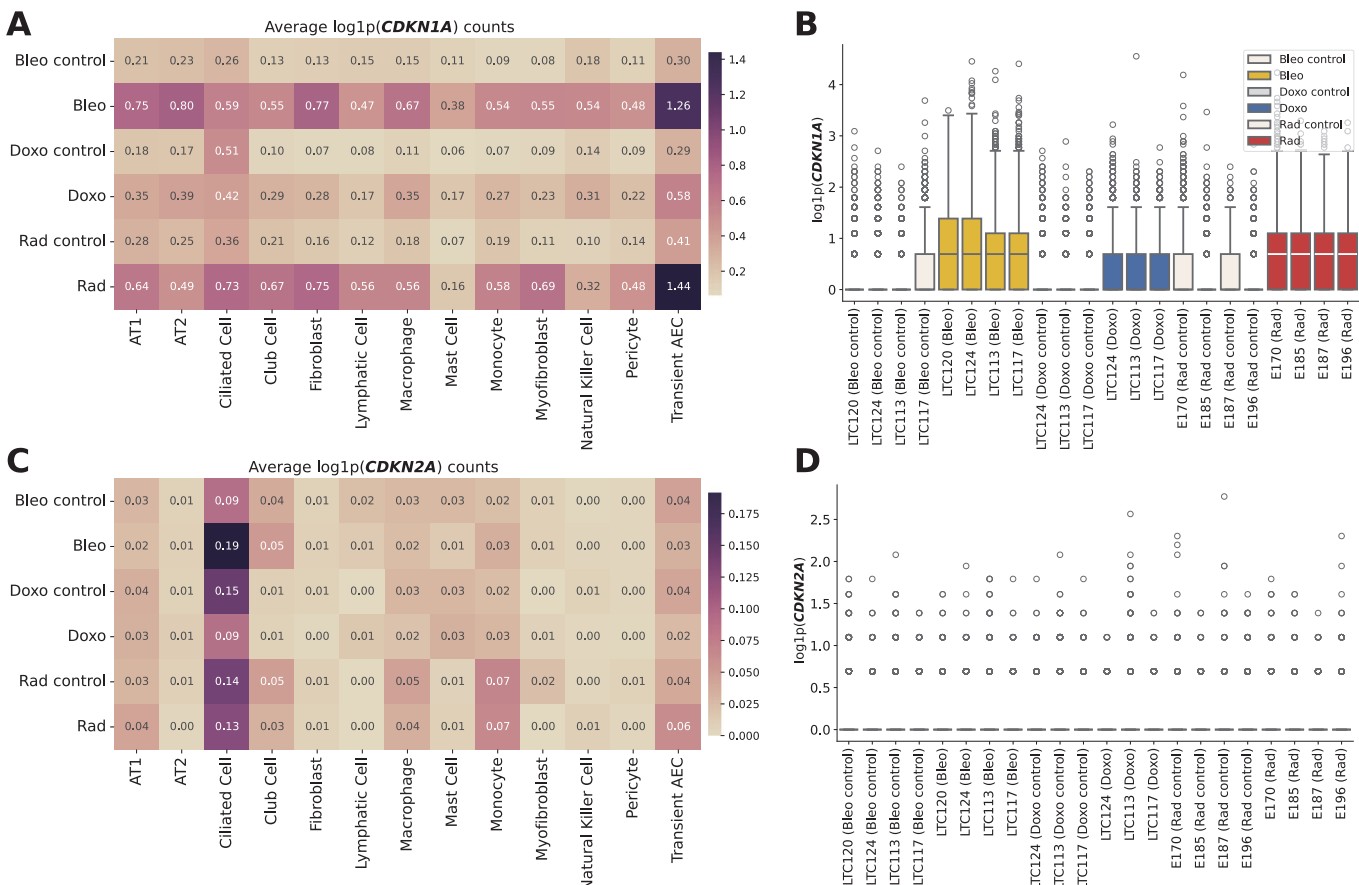

**Figure EV12. CDKN1A and CDKN2A expression in PCLS by cell type and sample.**

(A) Heatmap of mean log1p-normalized CDKN1A expression across experimental conditions and cell types. (B) Boxplots showing individual CDKN1A expression values per subject and condition. Values of *n* shown in Tables 3 and 4. Boxplots show median, interquartile range, and whiskers at 1.5× IQR. (C) Heatmap of mean log1p-normalized CDKN2A expression across the same conditions and cell types. (D) Boxplots showing individual CDKN2A expression values, matching the layout in (B).

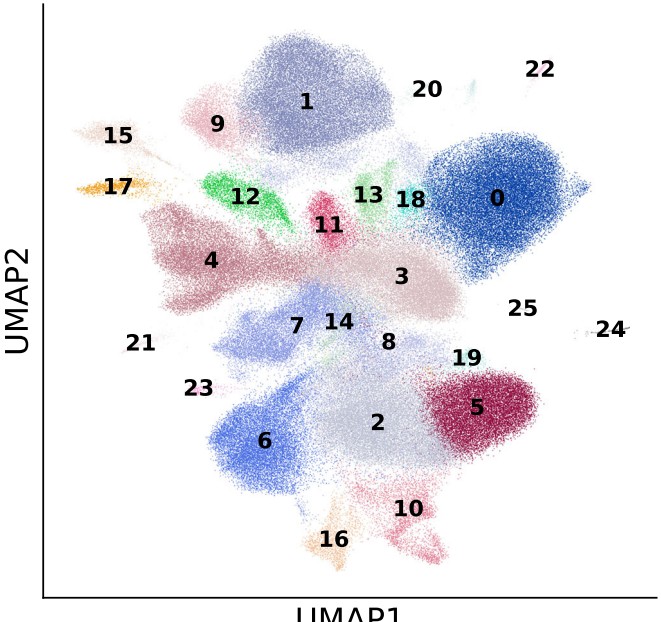

**Figure EV13. Clustering of integrated PCLS data.**

UMAP visualization showing all clusters identified using the Leiden clustering method in Scanpy.

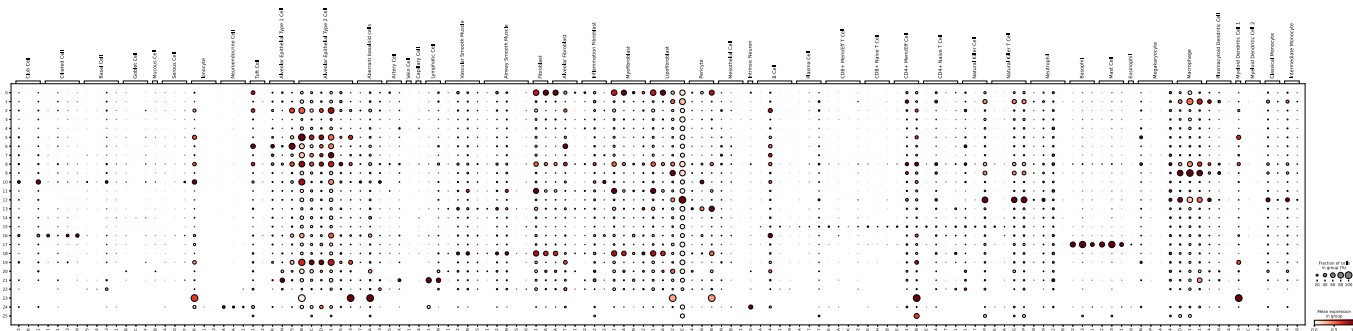

**Figure EV14. Canonical markers used to identify PCLS cell types.**

Dot plot showing canonical marker genes used for cell type annotation in the PCLS dataset. Markers were derived from Travaglini et al and supplemented with additional markers for transient alveolar epithelial cells. Source data are available online for this figure.

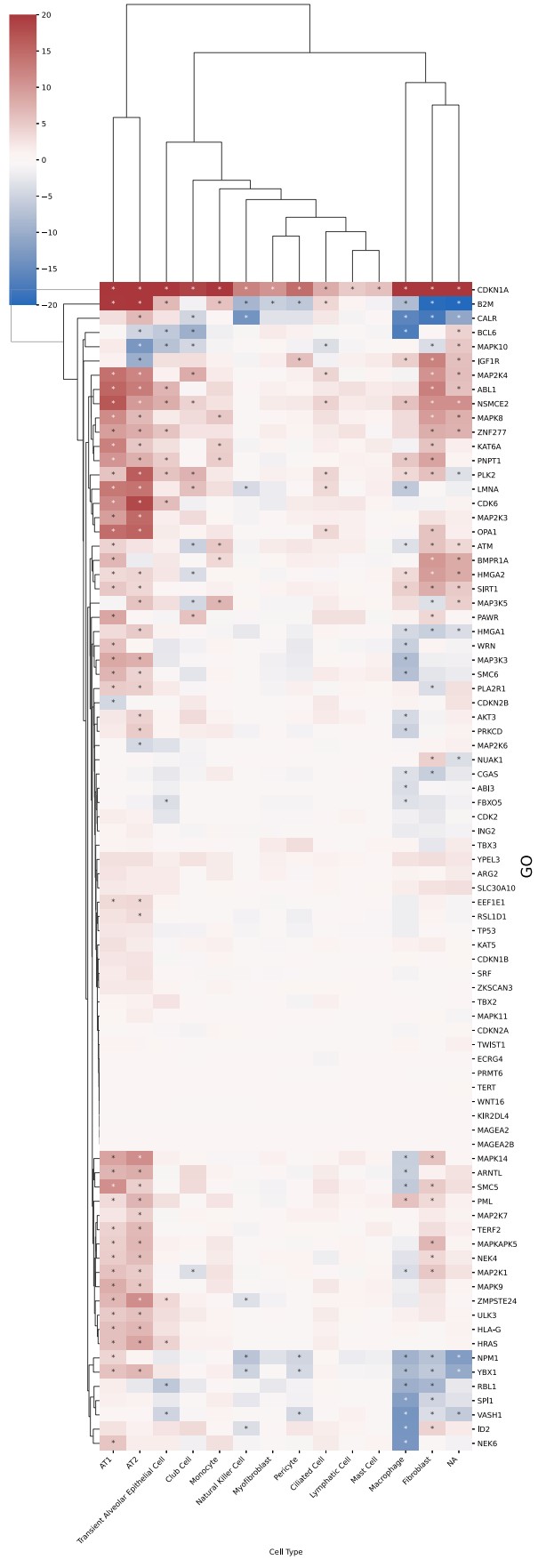

◀ **Figure EV15. Regulation of GO marker genes in the PCLS model.**

Heatmap showing regulation of Gene Ontology-derived marker genes across cell types in treated versus control PCLS samples. Source data are available online for this figure.

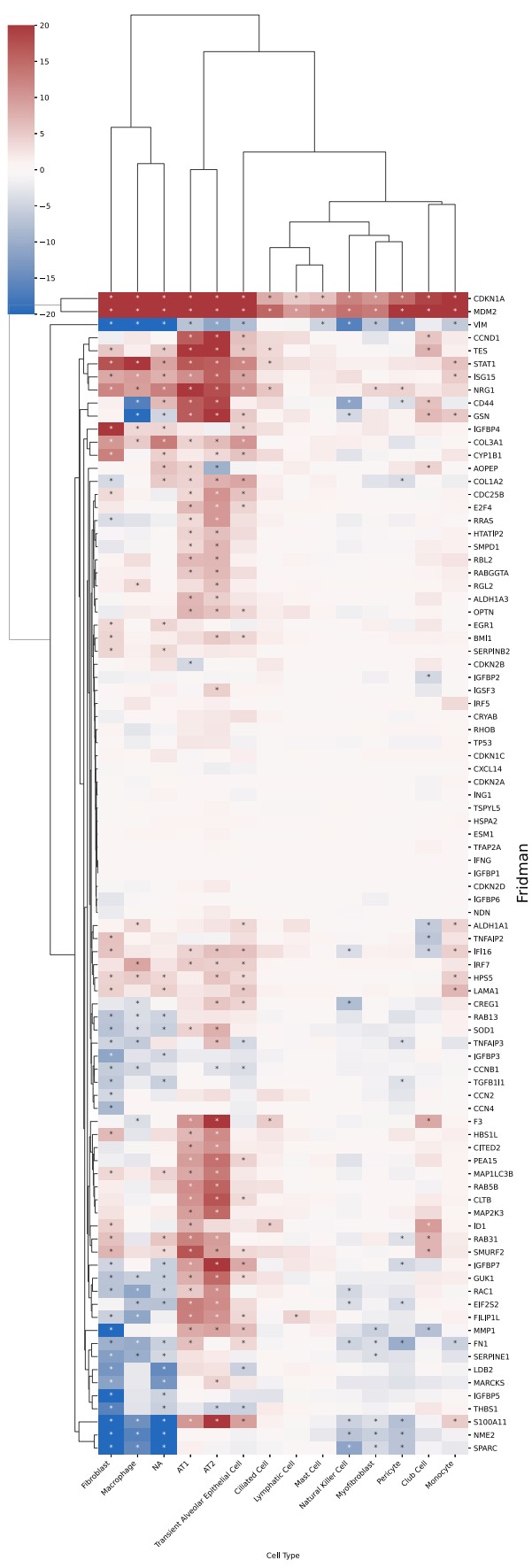

◄ **Figure EV16. Regulation of Fridman marker genes in the PCLS model.**

Heatmap showing regulation of Fridman senescence-associated marker genes across cell types in treated versus control PCLS samples. Source data are available online for this figure.

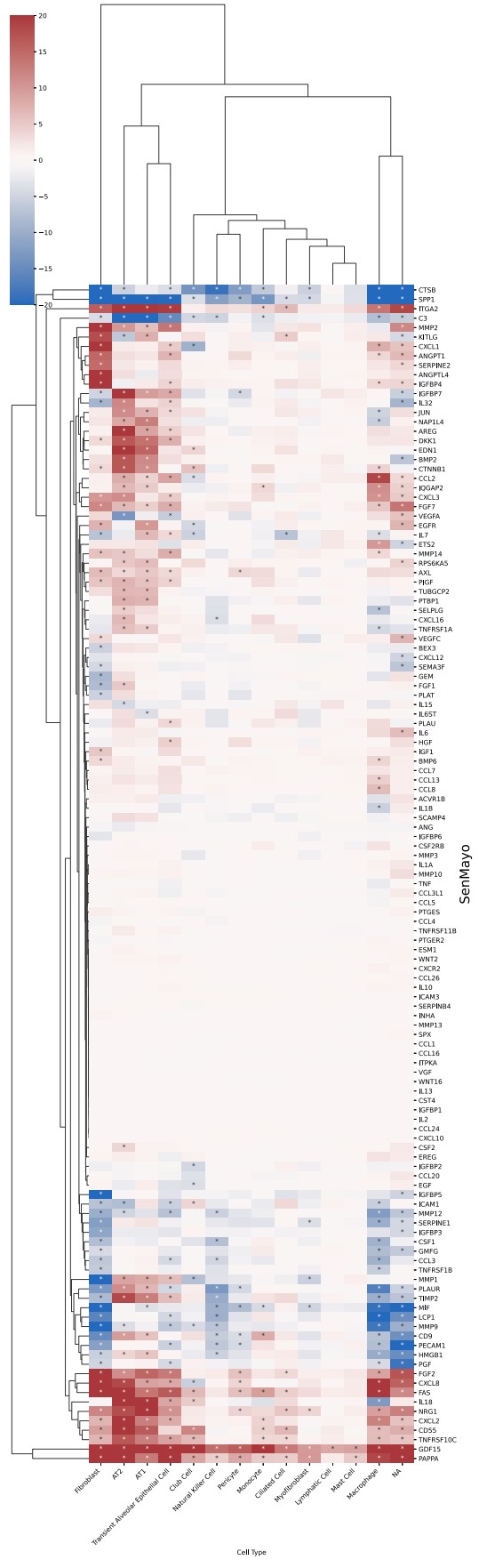

◄ **Figure EV17. Regulation of SenMayo marker genes in the PCLS model.**

Heatmap showing regulation of SenMayo marker genes across cell types in treated versus control PCLS samples. Source data are available online for this figure.

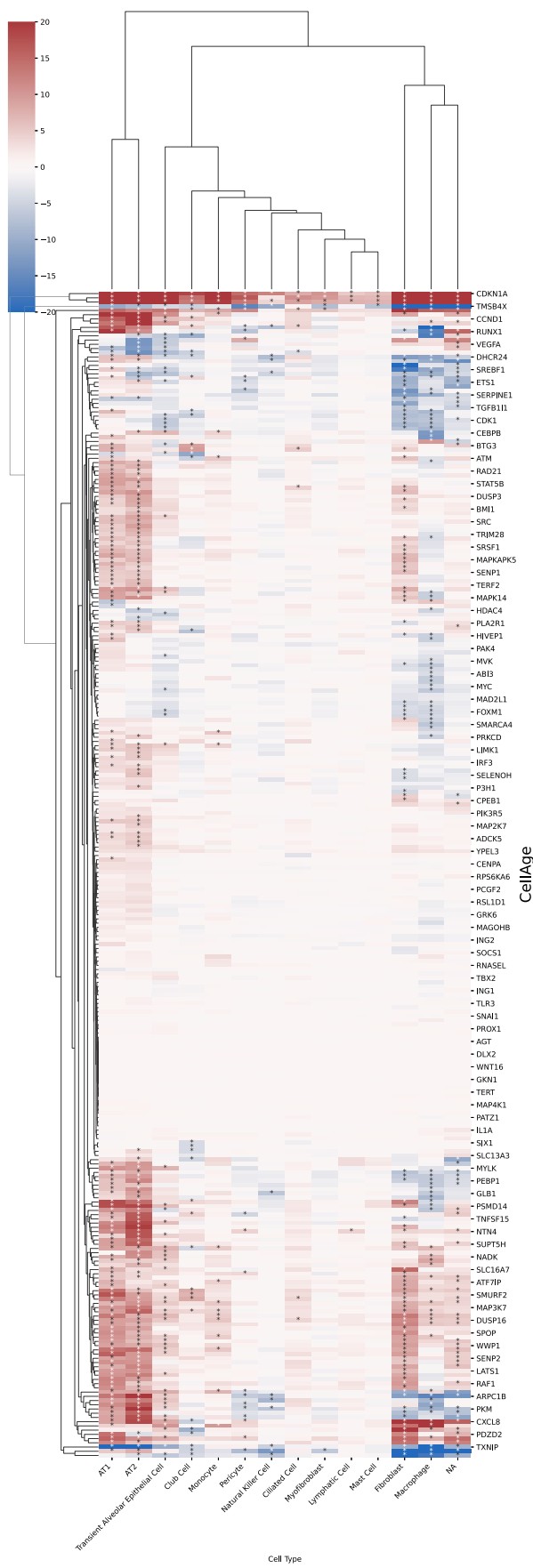

◀ **Figure EV18.  Regulation of CellAge marker genes in the PCLS model.**

Heatmap showing regulation of CellAge marker genes across cell types in treated versus control PCLS samples. Source data are available online for this figure.

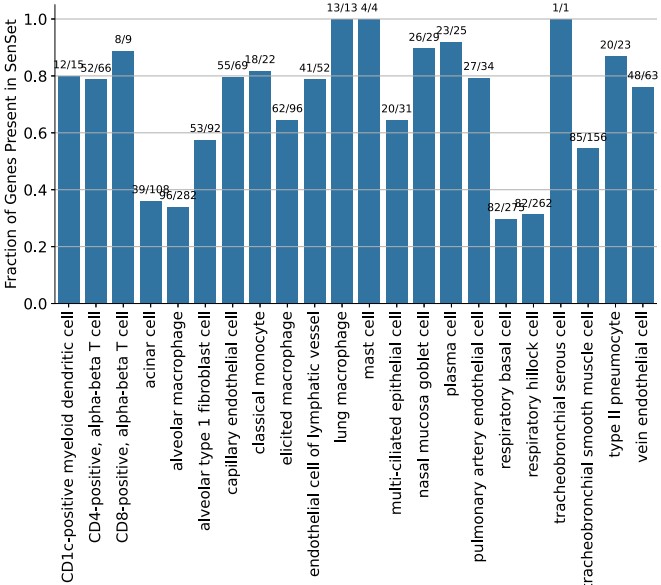

**Figure EV19. Fraction of cell-type-derived markers retained in SenSet.**

Bar plot showing the proportion of cell-type-specific marker genes that were ultimately included in the SenSet signature. Source data are available online for this figure.

