## [Peer Review File · The EMBO Journal]

SenSet defines cell-type specific senescence signatures in the aged human lung

Euxhen Hasanaj, Delphine Beaulieu, Cankun Wang, Qianjiang Hu, Lorena Rosas, Marta Bueno, John Sembrat, Ricardo Pineda, Maria Camila Melo-Narvaez, Nayra Cardenes, Zhao Yanwu, Zhang Yingze, Robert Lafyatis, Alison Morris, Ana Mora, Mauricio Rojas, Dongmei Li, Irfan Rahman, Gloria Pryhuber, Mareike Lehmann, Jonathan Alder, Aditi Gurkar, Toren Finkel, Qin Ma, Jose Lugo-Martinez, Barnabás Póczos, Ziv Bar-Joseph, Oliver Eickelberg, and Melanie Koenigshoff

Corresponding authors: Melanie Koenigshoff (koenigshoffm@upmc.edu) , Ziv Bar-Joseph (zivbj@cs.cmu.edu), Oliver Eickelberg (eickelbergo@upmc.edu)

Review Timeline:

Submission Date:	23rd Oct 25
Editor's Correspondence:	2nd Dec 25
Authors' Correspondence:	4th Dec 25
Editor's Correspondence:	23rd Dec 25
Authors' Correspondence:	19th Jan 26
Editorial Decision:	23rd Jan 26
Editor's Correspondence:	28th Jan 26
Revision Received:	22nd Feb 26
Accepted:	11th Mar 26

Editor: Daniel Klimmeck

Transaction Report:

Please note that the manuscript was previously reviewed at another journal. As EMBO Press has a transfer agreement with that journal, revision was invited based on the reports from that previous external submission.

Reviewer #1 (Remarks to the Author):

Hasanaj et al. presents a gene list termed SenSet to characterize senescence signatures in distinct lung cell populations upon aging and environmental exposures. The study is intriguing, as it aims to identify cell-type-specific senescence markers and validate them in an ex vivo human 3D lung tissue model. However, I have several concerns regarding the accuracy, depth, and experimental support of the study, which I outline below.

Q1. The manuscript lacks a clear rationale for classifying cells from young individuals as non-senescent. Without definitive markers or functional validation, this assumption may introduce bias in the interpretation of the results. The authors should clarify the criteria used for defining non-senescent cells and provide supporting evidence for their classification.

R1: We thank the reviewer for this important point. Our main assumption is that younger lungs have significantly fewer senescent cells than older ones. This is the only assumption we make for PU learning. Since it's a probabilistic framework, we do not need to specifically make the assumption that there are no senescent cells in younger lungs, only that the probability to encounter such cells is much lower. This point is now clarified in the **Discussion**.

Q2. The study reports an absence of *cdkn2a* positive cells, which contradicts findings from previous studies using single-cell RNA sequencing (e.g., DOI: <https://www.biorxiv.org/content/10.1101/2023.04.19.536722v1.full.pdf>). Given the well-established role of *cdkn2a* as a senescence marker, its absence raises questions about the sensitivity and validity of the approach used. The authors should discuss this discrepancy and, if possible, reanalyze their dataset to detect *cdkn2a* expression.

R2: The reviewer highlights an important point in detecting increases in *cdkn1a* and *cdkn2a* mRNA associated with senescent cell burden in complex tissues. The study cited above by De Man et al. only shows an increase in *cdkn1a* mRNA, not *cdkn2a* mRNA (Fig. 4c), like the data we present. While *cdkn2a* mRNA is robustly increased with senescence in cell culture models, its detection in complex tissues seems more challenging, with varying reports in the literature. We *do* see a robust increase of p16 protein levels in PCLS exposed to bleomycin (**see new Supp. Fig. 10C, also below**), highlighting the notion above that its induction is either more robust in ex vivo models of senescence, or that it reflects a different senescence program, as suggested in the publication mentioned. We have highlighted this in the **Discussion** and referenced the citation accordingly.

Figure for reviewers removed

Q3. The study assumes that young tissues predominantly contain healthy, non-senescent cells, which may not accurately reflect biological reality. Not all senescent cells are associated with pathology, and conversely, aged cells do not necessarily equate to senescence. The authors should acknowledge and address this limitation by discussing the heterogeneity of senescent cells in both young and aged tissues.

R3: We would like to emphasize that we do not assume that young tissues do not include **any** senescence cells and that all older tissues are **mainly** senescent cells. The premise of our approach is that older tissues have *significantly more* senescence cells than younger ones (even if, as a percentage of total cells, older tissues only have a few senescent cells based on organ size). It is well established that senescence “is a normal developmental mechanism during embryogenesis” (PMID: 24238961; 24238962), but not as much during early life, and then increases in multiple organs with age (PMID: 33981041; 37486065; 25568097). We now clarify this in detail in the Discussion.

Q4. It is unclear whether the identified SenSet gene list reflects true senescence-induced growth arrest or merely captures a general stress response. Senescence is defined by a stable cell cycle arrest, yet the study does not establish whether the genes in SenSet are associated with this key feature. The authors should provide functional validation, such as cell cycle profiling or proliferation assays, to determine whether SenSet genes are specific to senescence rather than transient cellular stress.

R4: We have addressed this specific point in several ways: First, we investigated proliferation in our ex vivo PCLS model upon Bleomycin- and Doxorubicin-driven senescence induction. To this end, we performed additional analysis to determine the functional GO categories enriched in our SenSet gene list. Gene categories enriched in SenSet include “**Regulation of Cell Population Proliferation (GO:0042127)**” (19 genes, adj. $p=1e-6$), “**Regulation of Cell Differentiation (GO:0045595)**” (8 genes, adj. $p=1e-4$) and “**Negative Regulation of Programmed Cell Death**”

(GO:0043069)” (13 genes, adj. $p=1e-5$). Accordingly, we have included a comprehensive list of GO enrichments in the supplement to this submission (New Supp. Fig. 6).

To further demonstrate growth arrest in senescent cells identified in this submission, we present novel data demonstrating that CDKN1A-positive cells (identified as senescent) are negative for the proliferation marker Ki67, after treatment with bleomycin or doxorubicin (Updated Figure 5F, New Supp. Fig. 9, also below).

Moreover, we demonstrate that SenSet gene list enrichment is also detected in independent models of replicative senescence. We performed SenSet enrichment analysis in cells undergoing **telomere dysfunction**-induced cell cycle arrest and senescence. We use the recently published and well-described human alveolar epithelial cells (A549) with shortened telomeres induced by TRF2 deletion (PMID: 33634147) (new Figure 4G,H). This analysis shows significant enrichment of SenSet genes *in vitro* (A549 cells).

Moreover, we analyzed a publicly available scRNA-seq dataset of mesenchymal stem cells (MSCs) undergoing **replicative senescence** (PMID: 37314668). For this data, we compared cells at a late stage (Day 86) with cells at mid-stage (Day 49). We find that SenSet shows very strong enrichment for both the MSC replicates, achieving FDR=0 for one of them (Rep #3). The Fridman list also performed well, which is not surprising as this list is primarily focused on replicative senescence (new Figure 4A-F, also below). Together, these new data strongly support our finding that SenSet detects senescent cells in stable cell cycle arrest.

Figure for reviewers removed

Figure for reviewers removed

Q5: The validation of SenSet in an ex vivo human lung tissue model is not entirely convincing. The experimental conditions used to induce senescence appear to strongly upregulate p16, which is not detected in vivo (according to the authors). This raises concerns about whether the system accurately mimics aging-related senescence. Furthermore, the reported induction of only 3–5% p21 positive cells under high-stress conditions is unexpectedly low. The authors should explore whether these conditions truly recapitulate physiological senescence and clarify how stressor intensity might affect their findings. It would be valuable to understand whether differences exist between datasets derived from single-nuclei RNA sequencing versus single-cell RNA sequencing. The authors should address this point by comparing their findings with publicly available single-cell datasets and discussing any potential discrepancies.

R5: The *CDKN2A* expression in the PCLS model is indeed low compared with *CDKN1A* expression (snRNA seq dataset, Figure 6F), a finding very similar to what is shown *in vivo* in the HLCA dataset (Figure 2J) and other atlases (PMID: 37291214). Importantly, most (published) datasets detect robust expression and induction of *CDKN2A* in senescence, which we have further expanded on in our ex vivo model. We observe a robust expression of p21-positive cells. Indeed, in Figure 5 E, F and G, we have mislabeled the y-axis, which now correctly depicts fold change. We have included two new supplementary figures that show boxplots of expression levels of *CDKN1A* and *CDKN2A* for both the HLCA and PCLS (new Supp Fig 5, 11, also below). While outliers expressing *CDKN2A* exist, overall the *CDKN2A* signal is much lower compared to *CDKN1A* in both datasets.

Figure for reviewers removed

Figure for reviewers removed

Reviewer #2 (Remarks to the Author):

Major points:

Q1: One concern, at the innovation level, is the fact that the authors are based on published single-cell datasets (HLCA) and genesets (many of them coming from in vitro senescent ecosystems/cell lines unrelated to the lung or unrelated senescent triggers, Figure 6). It would be ideal that SenSet had been developed from specific signatures of human lung senescent cells coming first from author's datasets (like those from Figures 4-5 explants) and then correlated with both bona fide (lung) published dates. This would enhance specificity of human lung senescence when compared to other organs, triggers, and cellular states. The innovative aspect and specificity of the tool for human lung senescence should be addressed by the authors.

R1: We appreciate the reviewer's concern regarding the specificity and origin of the SenSet signature. We selected the Human Lung Cell Atlas (HLCA) as the foundation for SenSet because it represents the largest and most diverse single-cell dataset of healthy human lungs to date, with over 100 individuals across a broad age range (10–76 years) and providing high-resolution, cell type-specific data. We believe that our innovation is primarily driven by the novel machine learning approach, which is applied to one of the largest currently available human lung datasets and its subsequent validation in a living human tissue model. We agree that future studies investigating inducer-specific gene sets directly from our own ex vivo datasets will be of interest, however, these datasets are currently limited in both scale and donor diversity to serve as an initial exploratory cohort to apply and validate novel computational approaches. We are increasing our donor pool constantly and are planning to do this analysis in the future.

Q2: A critical point is that the use of PU learning algorithm requires labelled training and unlabelled test datasets to work. In this manuscript, the algorithm assumes that all young cells in lung tissues are non-senescent (labelled positive according to Fig. 1) and that the gene expression of old non-senescent cells (unlabelled positive according to Fig. 1) have similar expression patterns and distribution to those of young cells. Previous reports have shown that ageing indeed alters gene expression patterns in lung cells in ways that are directly related to senescence (<https://doi.org/10.1101/2023.06.16.545378>, <https://doi.org/10.1038/s42003-024-06111-x>). However, this assumption could introduce some bias the selection process, potentially leading to the identification of false positive cells whose age-related gene expression changes may not necessarily be linked to cellular senescence, even if some genes from SenSet are enriched in those cells. SenePy and SenCID are both published methods of senescent cell detection in single cell RNA-seq datasets and the authors should provide evidence that SenSet is an improved technique or comparable to these validated methods.

R2: We thank the reviewer for this important point and the opportunity to clarify our methodology. The PUC learning framework does not assume that the gene expression profiles of old non-senescent cells (> 50) are like those of young cells. In fact, our approach explicitly

accounts for age-related expression changes by incorporating covariate shift correction. We only assume that middle-aged (30 - 50) and young cells (< 30) are similar in distribution, which is biologically more plausible, although we agree it is still not absolute. Following this comment, we performed additional analysis to compare SenSet to **SenePy**. We used SenePy's full lung set (~1,000 genes) and performed a similar enrichment analysis using SenePy on our ex vivo datasets. We added this result to the **comparison figure** in the supplement (also below). We observed that the proportion of SenePy genes significantly enriched in treated samples was generally lower than the other sets, showing that SenSet has higher specificity.

Figure for reviewers removed

In addition, we looked at the overlap between cells determined to be senescent cells using either SenSet or SenePy. As evident from the figure below, even though only few overlapping genes exist between SenSet and SenePy (8 genes), we found good agreement between cells labeled as senescent by either method. Nonetheless, SenePy's gene set lacks several key senescence markers, such as CDKN1A or CXCL8. We have now fully included this comparative analysis as **Supp. Fig. 12**.

Figure for reviewers removed

As for **SenCID**, we could not include a comparison with SenCID, since it's a supervised method that requires known senescence labels for a subset of cells; this is not available for our data. Additionally, SenCID was trained on a relatively small cohort of approximately 600 cells, which may limit its generalizability when applied to large datasets such as the HLCA. We now **reference** this work accordingly in our manuscript.

Q3: Another important aspect that should be clarified is the precise definition of “truly” senescent cell populations within the dataset. For example, tracheobronchial serous cells appear to be overrepresented among the senescent cells detected by SenSet (76% of all cells in this cluster), raising the possibility that some may be false positives. Additionally, incorporating more explicit thresholding criteria for identifying “true” senescent cells could help refine the method and reduce potential misclassification. Exploring “bona fide” senescence and non-senescent cell signatures (e.g., perhaps from the controlled experiment in the snRNA-seq data included in the paper) in human lung cells could further strengthen the validation of the gene set and the PU learning pipeline.

R3: We agree with the reviewer that the proportion of senescent cells appears very high in some cell types. Tracheobronchial cells were excluded from downstream analyses, as the high fraction is likely due to noise or a violation of PUC learning assumptions. In our framework we set the mixing parameter to 90%, reflecting the assumption that most cells in healthy lung tissue are non-senescent. Despite this assumption, the PUC learner identified <1% senescent cells in many cell types, indicating that the model is robust to this parameter. Moreover, to generate SenSet, we selected genes that were differentially expressed in *most* cell types. So even if a few cell types showed a high proportion of false positive senescent cells, these had only a limited impact on the final gene set. We expanded our **discussion** highlighting these limitations and considerations.

Q4: To strengthen the selection of “true” senescent cells, the authors may benefit from using a snRNA-seq dataset of ex vivo samples completely untreated. This is important since in Fig. 5D and 5E the UMAP plots show a clear pattern of CDKN1A and CDKN2A expression in cell clusters from the control treatments (e.g., AT2 cells). The use of dotplots or heatmaps could also help the readers to understand in which clusters these markers are mostly expressed and their percentages in the datasets across cell types.

R4: We thank the reviewer for the suggestion. We have expanded our supplement with figures showing the average expression of CDKN1A and CDKN2A for PCLS (new Supp. Fig. 11, also below). As can be seen below, there is a presence of outlier cells expressing CDKN2A, however, the distribution is very sparse when compared to CDKN1A. Some cell types (primarily ciliated cells) show an upregulation of CDKN2A across all conditions.

Figure for reviewers removed

Q5: It is unclear whether the data showed in Figure 4 is related to the age of the lung samples (donor information from Table 2). Since SenSet heavily relied on comparisons between young and old lung cells, it is relevant to show how responses to bleomycin, doxorubicin, or irradiation differ across age groups in the human lung. This clarification is also needed for Figure 5. Authors should break down the graphs and snRNA-seq data exploration into the three age groups available in their dataset and assess the senescent cell signatures and cell types in an age-group specific manner.

R5: We thank the reviewer for this comment. While it is true that the PU learning aims at identifying senescent cells in old people, once they are identified, SenSet selects genes by comparing **senescent** and **non-senescent in old individuals only**. Thus, age is unlikely to be a major confounding factor. We nevertheless provide below, as requested, p21 induction by Bleomycin and doxorubicin, separated by age. While p21 might be increased on baseline in older donors, given the very small size of this cohort, we believe that it is hard to derive specific conclusions about differences in response for cells from young and old individuals. As for treatment, it is also not clear what the assumption should be, since ex vivo induction may have similar impact regardless of age.

Figure for reviewers removed

Legend: Age dependent p21 quantification from IHF on 4 μ m sliced formalin-fixed paraffin embedded human PCLS at day 6.

Q6: In Figure 4, there is a lack of validation showing whether bleomycin, doxorubicin, or irradiation induce damage in the ex vivo lung cultures. It would be helpful to include data on markers of proliferation, apoptosis, fibrosis, or inflammation to better validate the results. Additionally, the expression of p16 could provide valuable insights into whether the observed changes in the human lung in response to these treatments are exclusively dependent on p21. In Figure 4, the authors should include other markers of the SASP to help better characterize the effects of these treatments on ex vivo lung cultures (e.g., using snRNA seq obtained from the experiment), as higher GDF-15 expression is significantly associated with aged tissues.

R6: We thank the reviewer for this question. We have accordingly expanded upon our characterization of the ex vivo model. In this revised version of our manuscript, we have added

new data on cell proliferation (new Supp. Fig. 9), SASP factors (see new Supp. Fig. 10) secreted into the supernatants, and extent of fibrosis (Supp. Fig. 10). In short, we observe a robust and strong reduction of Ki67 staining concomitant with increased expression of several SASP transcripts in ex vivo lung cultures. We further show that fibrotic markers are indeed induced in ex vivo lung cultures, which is in line with our analysis of SenSet in lungs obtained from patients with pulmonary fibrosis (see comments to Q8).

Q7: How specific is SenSet for senescent cells within the human lung? Could it also be used to detect senescent in single-cell transcriptomic data from other tissues, species, or diseases, whether lung-associated or not? Including this information in the discussion section would improve the quality of the manuscript.

R7: We thank the reviewer for this suggestion, which we have performed in this revised version with added data. We performed enrichment analysis of SenSet and the other lists in publicly available datasets from liver cirrhosis (PMID:31597160) and found strong enrichment of SenSet in fibrotic livers compared to controls, substantiating the extent of senescence in these samples. We have further expanded our Results section to include this important point. This figure is now a new Supp. Fig. 8C.

Figure for reviewers removed

Q8: The authors state that the findings have implications for several lung-related diseases (lines 52-54). However, no data are presented to support the use of SenSet in single-cell datasets from diseased human lungs. I would recommend toning down this claim or provide further supporting evidence.

R8: In this revised version, we have now performed enrichment analysis of SenSet and SenMayo in idiopathic pulmonary fibrosis (IPF), a devastating lung disease known to exhibit significant amounts of senescent cells. We utilized the IPF cell atlas (www.ipfcellatlas.com) and performed a prerank enrichment analysis, which in addition to a p-value also returns an enrichment score. As expected, we find that SenSet showed positive NES in many cell types in

the IPF atlas, outperforming all other lists (new Supp. Fig. 8A). We then zoomed into the performance of SenSet and SenMayo for individual cell types, and found that the enrichment scores for SenMayo were generally lower, with SenSet achieving an FDR=0 for 9 types while SenMayo for none. We have added these data below and in the supplement in Fig. 8B.

Figure for reviewers removed

Figure legend: Enrichment of senescence gene sets in IPF lung samples. A diamond signifies a p-value of 0; only significant results (FDR < 0.05) are dotted. Original data from the IPF cell atlas are published in www.science.org/doi/10.1126/sciadv.aba1983

Q9: Following with the innovation level, the manuscript claims to be the first to comprehensively analyse senescent cells in the healthy aging lung. However, the works mentioned in the previous comment (SenePy and SenCID) have also characterized the niche of senescent cells in human lungs. Additionally, similar approaches have been applied to murine datasets in other studies. It would be helpful for the authors to clarify what makes their analysis distinct from these previous studies, whether in methodology, resolution, or specific insights gained.

R9: We appreciate this suggestion by the reviewer and have carefully revised and modified our claims. In addition, we have performed further analysis to compare with previously published work. While other methods have also used healthy lung data, those analyses relied entirely on supervised approaches. Our approach offers a key advantage to those: By leveraging PU learning under covariate shift, we were able to identify senescent cells directly within the aged population, without relying on direct comparisons to young individuals. This is particularly important, because most existing senescence detection strategies depend on comparing old versus young samples. While informative, such comparisons can conflate general aging-related

transcriptional changes with true cellular senescence. We have now added comparisons with SenPy that show how such approaches can lead to very different results (see comment to Q2).

Q10: The discussion section primarily focuses on evaluating the relevance and functions of the different genes included in SenSet. Notably, CDKN2A is not included in SenSet, and the authors should further discuss the rationale behind this choice. Does this imply that p16 is not relevant for senescence induction in the human lung? Additionally, how cell type-specific is CDKN1A-versus CDKN2A-dependent senescence in this context? Moreover, the discussion lacks a proper comparison of SenSet with other existing tools that have already been published. Given the need for further benchmarking of SenSet, it would be valuable for the authors to include a discussion on how SenSet compared to methods such as SenPy and SenCID and any others, to better position its potential advantages and limitations.

R10: We have substantially revised our manuscript by performing new experiments and improving the clarity of the discussion section, based on those suggestions by the reviewer. First, while our list (as well as others, such as SenMayo) does not include p16, we don't imply that p16 is not important for senescence. It is well known that p16 transcript detection in single cell and single nuclei data is low, often giving less robust results for approaches identifying gene signatures. We do, however, see robust p16 protein induction in our ex vivo PCLS model (see new Supp. Fig. 10C), suggesting that p16 is part of a DNA damage-induced senescence program in the lung. Second, we have performed additional experiments comparing SenSet with SenPy (see comment to Q2 and new Supp. Fig. 12) and further discussed other approaches with advantages and limitations in the revised manuscript.

Q11: This manuscript does not explore the presence of senescent cells in middle-aged human lungs (30 to 50 years). This would be clinically relevant, as lung cancer incidence rises dramatically from the age of 50, suggesting that senescent cells in younger age groups (30 to 50 years) may play a role in disease progression upon exposure to damage (e.g., smoking). Including an analysis of this age group or discussing its potential relevance in the context of SenSet would enhance the quality of the manuscript.

R11: We have added a new supplementary figure (new Supp. Fig. 7) analyzing senescence in the middle-aged group. Using our PU learning framework, we identified cells classified as "senescent" within this group and performed a differential expression analysis analogous to that applied to the aged group. This resulted in 139 marker genes, of which 90 overlap with our original SenSet (106 genes), demonstrating strong agreement between the two signatures. The new figure provides a cell type breakdown of this overlap, highlighting consistent senescence-associated transcriptional patterns between middle-aged and aged individuals.

Figure for reviewers removed

Q12: Cell clusters in Figure 2D are defined quite broadly, with many small clusters included under the definition of cell identity of larger clusters. However, the UMAP plot clearly shows a separation between these clusters, suggesting that the annotation of cell types might benefit from further refinement (e.g., cluster 9 of myeloid cells in yellow). Refining these annotations could better delineate the identity of senescent cells across the figures and improve the consistency throughout the manuscript.

R12: We agree with the reviewer, since this is what we did. The figure is just for illustration purposes. In Figure 2D, we only used level 2 annotation of the HLCA. For our main analysis and results, however, we use level 5 annotation, which provides annotations for more than 50 cell

types/populations. For more detailed annotations, please see Figure 3E (cluster numbers match those in panel A).

Q13: Furthermore, the authors claim to identify cell-specific senescence mechanisms in the healthy human lung during ageing, but they do not include any piece of conclusive mechanistic data. It would be important for authors to include mechanistic data to enhance the quality of the manuscript.

R13: In this revised version, we have now performed several new experiments clarifying the above. We refer to the new Figure 4G,H, which comprises SenSet analysis of a recently published senescent cell culture model. We performed SenSet enrichment analysis in cells undergoing **telomere dysfunction**-induced cell cycle arrest and senescence. We use the recently published and well-described human alveolar epithelial cells (A549) with shortened telomeres induced by TRF2 deletion (PMID: 33634147). This analysis shows significant enrichment of SenSet genes *in vitro* (A549 cells).

Minor points:

Q14: Please indicate sample size in all age groups and conditions in the corresponding figure legends.

R14: We have added sample sizes in all figure legends accordingly. Additionally, we have added new tables under Methods with the number of cells and genes analyzed for each dataset used in this study.

Q15: Figure 1 is missing panel indications, which would help clarify the legend and make the figure easier to follow. Please include this for better clarity.

R15: We have now added panel indications to Figure 1 and updated the caption accordingly.

Q16: Cell cluster colours in Figure 3E do not correlate with those in Figure 2D, and the UMAP plots are somewhat difficult to follow. Also, the names of these 22 new clusters are not provided. To maintain consistency and avoid confusion, I would recommend using the same cell identity scheme throughout the manuscript. Additionally, it would be helpful to name the cell clusters instead of using numbers.

R16: We clarified the discrepancy between Figure 3E and 2D in a previous comment above. Cell numbers in 3E are the same as those depicted in Figure 3A. We understand this was difficult to follow and have accordingly clarified the legends.

Q17: The authors have cited figure panels out of order in the text, which could cause some confusion. This issue is present in several panels of Figures 3 and 5. It would be helpful to ensure that the figure panels are references in the correct order to improve clarity. Additionally,

Figure 5A is not cited in the main text, only in the methods section. Please, include a citation for Figure 5A in the appropriate place, or alternatively, move the figure to the supplementary section.

R17: We have addressed these comments in our revised manuscript to ensure all figures are referenced appropriately.

Q18: Figure 4B-4D. Could the authors clarify why only the results with bleomycin are shown? Do they have similar data for doxorubicin? Additionally, the images in panel 4B are quite small and difficult to see. It would be helpful to provide larger images for better visibility.

R18: We have now added new and larger images for Figure 5B. Due to limited space, we focused largely on Bleomycin results in the main figure and added supplemental figures and new data for doxorubicin. See new Supp. Fig. 9A.

Q19: In the legend of Figure 4, the authors mention using 15 mg/mL of bleomycin, but in the results and materials section, they state 15 μ g/mL. Please review and correct accordingly.

R19: We thank the reviewer for catching this. It has been corrected to 15 μ g/mL.

Q20: Figure 4F. Could the authors clarify why there are apparent differences in p21 WB levels between the doxorubicin and bleomycin controls?

R20: We performed experiments with appropriate hence different controls based on solvents used for doxorubicin and bleomycin (DMSO is used as neg control for the Doxorubicin versus PBS for Bleomycin).

Q21: Please review some nuclear expressions, such as those in line 162 (“in signaling in transcriptional response”); title in line 191; line 197 (“living, diseased human tissue”); or Figure 5C legend (“Up(down) regulated in most samples”).

R21: We thank the reviewer for noticing these and revised the text accordingly.

Q22: Pie charts should include percentage values to better convey the results (e.g., Supp Fig 1 and Fig 2G). Adding these values would improve clarity.

R22: We revised the figures accordingly.

Q23: The authors mention that the expression space of young smokers is closer in distribution to that of old non-smokers than to that of young non-smokers. Could they please elaborate on this observation and provide further discussion on its potential implications?

R23: These data could indicate that smoking accelerates aging signatures, which has been discussed in the literature. Smoking is a major risk factor for many age-related lung diseases. Smoking causes elevated levels of reactive oxygen species (ROS), which lead to ROS-induced cellular damage and oxidative stress resulting in inflammation and senescence. Ultimately, this is thought to lead to abnormal lung injury responses, impaired tissue repair, and increased susceptibility to diseases such as COPD and pulmonary fibrosis (PMID: 40222750, PMID: 34735706). We have revised our **discussion** to include this point.

Q24: Please ensure that all references are reviewed and formatted according to the journal's guidelines before publication. This will help maintain consistency and meet the journal's requirements.

R24: We have revised references accordingly. Please note that a final round of reference corrections/formatting will be made during the proofs stage.

Reviewer #2 (Remarks on code availability):

n/a

Reviewer #3 (Remarks to the Author):

Major:

Q1. In the bulk of your analysis section, you are very careful to avoid using your gene list to explicitly predict proportions of senescent cell in single-cell data. In your discussion, you state that “fibroblasts and basal cells exhibit a high proportion of SnCs in aged lungs, accounting for 44% and 39% of all cells of that type in the oldest age group”. How you arrive at this prediction isn’t covered in your results or methodology, could you elaborate?

R1: We thank the reviewer for bringing this point to our attention and apologize for the confusion. We have now clarified this in the methods section with details on the specific decision function used. In short, for our PU learning setup, we set a mixing prior of 90% which reflects an assumption that the vast majority of cells are non-senescent. However, this is only a prior and the posterior is computed by the method. For the classification of the cells, we rely on a linear-in-parameter Gaussian model, like the one used by the authors of the PUC estimator.

Q2: The above proportion is likely quite a bit higher than previous estimates of SnC counts in vivo. How could you explain this other than your method is predicting more than just the SnC burden?

R2: To our knowledge this is the first paper to describe cell-type specific SnC burden in human aged lungs. While these % might seem high, we would like to point out that the cell types themselves (fibroblast and basal cells) are only a small fraction of all cell types detected, so that the SNC burden of the whole lung over age is still in the low percentage range. Data in this area are sparse, which was one of the main reasons to perform these studies (and to create the Sennet Consortium).

Q3: In your validation scRNA-seq data, how well do your numbers of predicted senescence cells using SenSet overlap with a prediction you could derive from your imaging and fluorescence staining?

R3: Thank you for this important point. We performed a comparison analysis between p21-positive cells as measured by IHF and CDKN1A-positive count cells in our single-nuclei data for each subject (both Doxo and Bleo treatments). We find that, across the majority of samples, the fold-change in % positive cells upon treatment is well within experimental variation in both modalities (one exception is sample LTC 124 under bleomycin) (see figure below). The correlation between a SenSet-derived senescence score and p21-positive cells is more complicated. As we show, neither of these proteins alone is sufficient to capture the full spectrum of senescence-associated transcriptional changes, hence the need for a new list. Furthermore, one assay measures protein accumulation while the other relies on mRNA expression, which makes this correlation for multiple genes less interpretable.

Figure for reviewers removed

Minor:

Q4: Could the authors elaborate on why they think the senescent cell burden is higher in some cell-types than others? Is this linked to their methodology surrounding their gene set or to the underlying biology?

R4: Our methodology is agnostic to specific cell types; however, we would like to highlight that we only considered genes to be included into SenSet that were found in at least 6 different cell types. So, in theory, if senescence in single/few cell types is driven by a completely different gene set than others, this might not be detected leading to larger variations among cell types. Independent of our methodology, our data support the hypothesis that the biology driving cell type origins, phenotypes, and functions contribute to differences in susceptibility and extent of SnC burden. We have expanded our **discussion** to cover this important topic.

Q5: The authors have primarily used models of cytotoxic stress and DNA damage to generate their SenSet list. How well does it perform on other senescence models, such as oncogenic and replicative senescence?

R5: Following this and similar comments from Reviewer 1 we analyzed additional datasets. Specifically, we looked at a scRNA-seq dataset of mesenchymal stem cells (MSCs) undergoing replicative senescence (PMID: 37314668). For this data, we compared cells at a late stage (Day 86) with cells at mid-stage (Day 49). We find that SenSet shows very strong enrichment for both the MSC replicates, achieving FDR=0 for one of them (Rep #3). The Fridman list also performed

well, which is not surprising as this list is primarily focused on replicative senescence (new Figure 4A-F).

We also performed SenSet enrichment analysis in cells undergoing telomere dysfunction-induced cell cycle arrest and senescence. We use the recently published and well-described human alveolar epithelial cells (A549) with shortened telomeres induced by TRF2 deletion (PMID: 33634147) (new Figure 4G,H). This analysis shows significant enrichment of SenSet genes *in vitro* (A549 cells). Together, these new data strongly support our finding that SenSet detects senescent cells in stable cell cycle arrest.

Figure for reviewers removed

Q6: The authors should include links to data in publicly available repository such as NCBI GEO prior to publication. This should include raw data and processed matrices. As data is mainly common primary cell lines, no reason for any exclusion.

R6: We have added accession numbers for all our LTC samples deposited at the SenNet data portal <https://data.sennetconsortium.org>. Additionally, the integrated data is also available for download:

https://drive.google.com/file/d/1-700I1a-JtiAJh7LUit0HH9LkNreI_iY/view?usp=sharing

Reviewer #1 (Remarks to the Author):

I thank the authors for addressing some of the concerns. However, most of the responses do not fulfill the level of depth required.

Q1. While probabilistic assumptions are valid in principle, the lack of experimental or computational evidence supporting the baseline classification remains problematic. Even if senescent cells are less frequent in young tissues, their presence can bias the training set and affect the sensitivity and specificity of SenSet. Clarifying assumptions in the discussion does not solve the core issue of potential mislabeling. At minimum, the authors should provide sensitivity analyses testing how different fractions of senescent cells in the “non-senescent” population would impact the performance of SenSet.

R1. The main assumption that Reviewer 1 questions is our assumption that senescent cells are more prevalent in aged tissues vs. tissues from young individuals. While this is indeed an important assumption, it is strongly supported by several previous studies. For example, this is systematically reviewed in *Ageing Res Rev.* 2021 Jul;68:101334 and *Nature Reviews Molecular Cell Biology* volume 22, pages 75–95 (2021). In these and in several other studies researchers show that it is indeed the case that senescent cells are more prevalent in older people.

In addition, based on the suggestion of the reviewer we conducted a synthetic simulation study. We simulated a scenario where the reference set of healthy cells in the young donors was progressively contaminated with senescent cells (from 0 to 30% of total cells). We found that even when the healthy set was contaminated with up to 30% senescent cells, the F1-score decreased by around 8% on average. This modest decline demonstrates that our approach is robust to a presence of senescent cells in the young.

Figure for reviewers removed

Q2. The difficulty in detecting *cdkn2a* mRNA does not eliminate its well-established role as a senescence marker in multiple in vivo studies, including those using scRNA-seq in lung tissue. The authors do not address whether technical limitations (low read depth, dropout effects, etc.) or biological differences explain the discrepancy. Simply citing variability in the literature without systematic reanalysis leaves unresolved whether SenSet genuinely captures canonical senescence programs or reflects dataset-specific artifacts. At minimum, comparisons with multiple independent single-cell datasets should be provided to contextualize this absence.

R2. The concerns about *cdkn2a* mRNA detection by single cell analysis are well-known and described within the HCLA dataset, which has been built by combining 49 independent studies (Figure 2K and Supplementary Fig. 5). These challenges are largely due to the gene's complex alternative splicing, the low sensitivity and high technical noise of single-cell RNA sequencing.

Q3. The reply restates their assumption but does not address the biological heterogeneity of senescence across cell types and ages. Senescent phenotypes differ by trigger, tissue type, and microenvironment, and the assumption of higher abundance in older tissues does not guarantee that SenSet captures the diverse functional states of senescence. This risks oversimplifying conclusions and overinterpreting associations as causal.

R3. Please see our response to Q1 related to the assumption of high senescence burden in older tissues, which is well documented. We agree with the reviewer that this does not guarantee that SenSet captures all diverse functions, however, the main rationale for our study was to provide an insight into the potential biological heterogeneity and - using SenSet - highlight cell-specific differences. We fully agree that we don't demonstrate mechanistic studies that show causality for specific pathways, but are excited to decipher these in future follow up studies.

Q4. While these additions are valuable, they remain indirect measures. GO term enrichment and Ki67 negativity correlate with, but do not prove, stable senescence-associated cell cycle arrest. Many stress responses can transiently reduce proliferation and upregulate p21 without establishing canonical senescence. Direct functional assays or single-cell trajectory analyses would be needed to disentangle senescence from quiescence or reversible stress responses.

R4. Several citations demonstrate that Bleomycin/Doxorubicin induces senescence, and this is an established method in the field to test various molecular aspects of senescence (e.g. PMID: 14516132, PMID: 37751045, PMID: 19801496, PMID: 33823141.)

Q5. High-stress induction models (bleomycin/doxorubicin) may not mimic gradual, age-associated senescence, limiting physiological relevance. The low frequency of p21+ cells (3–5%) even under strong stress raises questions about detection sensitivity or the appropriateness of the model system. Differences between single-cell vs. single-nucleus RNA-seq remain insufficiently addressed; these platforms differ in sensitivity for low-abundance transcripts like *cdkn2a*. Comparative analyses across independent aging lung datasets using

both technologies are necessary to confirm whether SenSet captures authentic in vivo senescence signals rather than artifacts of the chosen model system.

R5. The reviewer incorrectly states that we only have 3-5% p21-positive cells, whereas we actually have up to 40%. The reviewer was referring to the original submission but not our response, in which we made clear that we corrected all labels. This is clearly presented in Figure 5E.

Reviewer #2 (Remarks to the Author):

The revised manuscript has been improved, and the authors have addressed some of the questions and comments raised during the first revision period. I acknowledge their efforts to provide an enriched piece of work. However, I still have two specific concerns and believe further clarification and experimental outputs are needed on the following points:

Q1. A major concern is the novelty of the work, as SenSet gene set largely derives from previously published gene sets. The discussion section is somehow a list of genes previously linked to senescence in other contexts, with limited conceptual and critical analysis of the current data. It is not convincing that SenSet performs better than other existing gene sets (e.g. SenMayo) in replicative senescence, IPF or liver cirrhosis models, as new Figure 4 shows similar confirmation by other sets. These points should be thoroughly debated and incorporated to the Discussion, and the limitations of the study fully recognised.

R1. The novelty of our work should indeed be better explained in Discussion as the Reviewer suggests. Briefly, our work is fundamentally novel in that it constitutes the first comprehensive generation of a cell type specific senescence-associated gene lists which is confirmed and validated not only via analysis of existing databases, but also by a complex experimental setup using living lung tissue. In fact, the Reviewer acknowledges the novelty in asking us in the next comment to validate some of the new cell type specific lists we provide. And again, the Reviewer incorrectly states that our results are not much better than prior work. However, as we show in Supplementary Figure 8, our work is much better than SenMayo and other tools for the new datasets we analyzed in the revision, especially when using datasets from complex tissue versus cell line data (IPF and liver cirrhosis data).

- Another major and still pending concern is the lack of mechanistic explanation of why different cell types have different senescence-associated gene signatures. The paper claims that SenSet identifies cell-specific senescence mechanisms (including in the Title) but does not offer any data to support this claim. To address this point, the authors could analyse more in depth one cell type of interest to test their predictions relative to SenMayo/other tools. For instance, approaches such as multiplexed IF against senescent cell makers and cell type markers, or in vitro depletion of senescent cells in lung explants, could provide mechanistic insights and prediction validation. Nonetheless, this point should at least be addressed more extensively in the Discussion.

We agree that mechanistic studies are of high interest but believe these are out of scope of the current manuscript, in which we described these cell-specific signatures for the first time. We added a paragraph in the discussion in which we discuss future studies reading as follows:

“Finally, while we identify cell type–specific senescence-associated gene signatures, we do not experimentally dissect the underlying mechanisms driving these differences. Future work should focus on mechanistic studies to validate and functionally characterize these cell type–specific pathways.”

Other points

- The results of SenSet vs SenMayo or other tools may not be related to the nature and actual biological role of the particular genes, but rather to the number of genes.

R1. This was the motivation behind us choosing approximately 100 genes (107 exactly), in order to obtain a similar number of genes as the other lists (GO, Fridman, and SenMayo have 83, 90, 125 genes, respectively; Fig. 3B). Furthermore, the statistical tests we ran factor in the size of the set when deriving a p-value.

- Line 415. “A comparative analysis of smoker and non-smoker groups within the HLCA revealed that smoking may accelerate aging in the lung”: not novel and not shown in this paper.
- Analysis of smokers in the HLCA: still missing the conclusion here and not clear the relevance.
- Consider the following paper for the Discussion:
<https://pmc.ncbi.nlm.nih.gov/articles/PMC7093180/>. Could this be a confounding factor?

R2. All three comments about the analysis of smokers can be addressed by modifying the text and discussion. We do not claim to show a novel link, but rather to confirm existing literature that links smoking with aging and senescence.

Reviewer #2 (Remarks on code availability):

n/a

Reviewer #3 (Remarks to the Author):

I am satisfied with the changes the authors made.

Dear Dr Koenigshoff,

Thank you again for the transfer of your revised manuscript (EMBOJ-2025-122817) to The EMBO Journal, as well as for the patience with our feedback which got protracted due to delayed expert input. We have carefully assessed your manuscript and the related point-by-point response provided to the referee concerns that were raised during review at a different journal. In addition, and as mentioned before, we decided to involve two arbitrating advisors - one of the previous referees, and one additional expert - to evaluate the revised version of your work, with respect to technical robustness, conceptual advance and overall suitability of your work for publication in The EMBO Journal.

As you will see from the arbitrating comments enclosed below, advisor #1 (former referee #2) states that the study has been further improved and s/he is now in favour of publication of the work. In contrast, arbitrating advisor #2, while acknowledging the potential of the developed signature and related findings, points to important remaining caveats with the analysis, related to interpretation and annotation of potential false annotations, potentially confounding experimental design and insufficient benchmarking by reference datasets, which need to be addressed in his/her view to meet the standards required for our venue.

We carefully went through these issues raised and feel that as they relate to robustness of the analysis and core aspects, we would need them to be conclusively addressed in order to proceed with this work for our journal.

I am hence reaching out to learn about your views on these critiques, in favour of a fair and informed decision. I would appreciate if you could go through the arbitrators' reports and let us know about your views at your earliest convenience. I would also be available to talk directly on the matter should this be helpful.

Thank you in advance for your kind cooperation.

I look forward to hearing back from you.

Best regards,

Daniel Klimmeck

Daniel Klimmeck PhD
Senior Editor

EMBOJ-2025-122817

Arbitrating advisor #1 (former referee #2)'s comment:

I believe the answers to my questions and those of the other reviewer have been clarified. Limitations of the study have been acknowledged. I consider this tool will be of significant support for the scientific community focused on lung diseases and the broader community working in senescence. The study is appropriate for EMBO Journal and its scope.

Arbitrating advisor #2's comment:

The manuscript presents a computational framework for refining the identification of senescence markers and applies this approach to derive a more sensitive senescence signature (SenSet) for lung tissue, based on previously established senescence marker lists and a series of cross-validation analyses. While the study meets the general requirements for a resource-type article, if the following issues are not adequately addressed, it would still fall short of the standards typically expected for publication in a high-impact journal such as The EMBO Journal. The main concerns are as follows:

1. Misclassification of tracheobronchial serous cells

The authors did not provide a sufficient analysis of why the PUC model mislabeled most tracheobronchial serous cells as senescent (SnCs). It seems overly speculative to attribute this misclassification solely to the limited overlap of differentially expressed genes with other cell types. An alternative explanation is that a subset of SnCs within this population exhibits distinct, cell type-specific senescence features. If these cells are indeed misclassified, the authors should clarify the underlying causes, outline how such false positives can be recognized in practice, and specify measures to minimize them. Addressing these points is essential for assessing whether the proposed method can be reliably generalized across different biological contexts.

2. Potential confounding in model construction

The authors performed differential expression analysis for the middle-aged group and identified 139 senescence marker genes, suggesting a substantial presence of SnCs in this group. However, in constructing the model of healthy cells, data from both young and middle-aged samples were included, which may have introduced confounding effects and potentially compromised the accuracy of the modeling.

3. Conceptual and methodological concerns in constructing the SenSet list

The rationale for including differentially expressed (DE) genes shared by at least six cell types in the SenSet gene list is questionable. Given that senescence signatures are inherently heterogeneous across cell types, a more suitable approach would be to establish distinct senescence marker lists for each cell type-or at least for clusters of closely related ones. Conversely, if the aim is to identify conserved markers across diverse lung cell types, a more stringent selection pipeline should be applied to ensure the inclusion of robust, universally expressed senescence markers. However, this strategy would contradict the paper's stated focus on a cell type-specific signature. In the section "Cell type-specific enrichment of SenSet genes," the observed enrichment pattern seems largely driven by how the SenSet list was constructed rather than reflecting genuine biological specificity. Furthermore, during validation with human lung tissue snRNA-seq datasets, the authors found that SenSet markers exhibited predominant upregulation in AT1 and AT2 cells, whereas they were largely downregulated in fibroblasts and macrophages. It remains unclear whether this pattern reflects genuine biological differences or systematic bias introduced during the construction of the SenSet list. The authors should therefore refine the conceptual framework and methodological design of SenSet to ensure that it captures biologically meaningful variation rather than analytical artifacts.

4. Inappropriate validation metrics and interpretation

In validation analyses, the authors demonstrated that SenSet outperformed previous senescence marker lists. Nevertheless, in the validation using human lung snRNA-seq data, the strategy of evaluating different senescence lists based solely on the number of significantly enriched cell types is not entirely appropriate. A larger number of enriched cell types does not necessarily indicate a better-performing list. Under various stress conditions or during natural aging, senescent cells tend to arise in specific cell types across tissues and organs. Therefore, a robust senescence list should show selective enrichment in truly senescent cell types. To further evaluate the sensitivity and specificity of SenSet relative to other lists by experiments, the authors could leverage senescence-

related genetic mouse models-such as the p16reporter line-to test their enrichment in reporter⁺ versus reporter⁻ populations.

Dear Dr. Klimmeck,

Thank you very much for the assessment and guidance on how to move forward. We have discussed and addressed the comments from the arbitration report below, outlining our views and responses to further strengthen the robustness and validity of our findings.

Thank you for your cooperation and guidance!

Melanie

Response to Arbitration Report

1. Misclassification of tracheobronchial serous cells

The authors did not provide a sufficient analysis of why the PUC model mislabeled most tracheobronchial serous cells as senescent (SnCs). It seems overly speculative to attribute this misclassification solely to the limited overlap of differentially expressed genes with other cell types. An alternative explanation is that a subset of SnCs within this population exhibits distinct, cell type-specific senescence features. If these cells are indeed misclassified, the authors should clarify the underlying causes, outline how such false positives can be recognized in practice, and specify measures to minimize them. Addressing these points is essential for assessing whether the proposed method can be reliably generalized across different biological contexts.

The proposed alternative explanation by the reviewer is valid and we are happy to include this in our discussion. We also can further clarify potential causes for misclassification: We analyze roughly 30 cell types, and this is the only one where we likely extensively misclassified cells. We identify 74 / 97 cells (76%) of tracheobronchial serous cells as senescent in the oldest age group (>50). We observed that in the HLCA, only 67 of these were labeled as "SMG serous" in the original datasets and 27 were labeled as "SMG duct" (recall that the HLCA consists of refined annotations). It is quite possible that this refining step might have been inaccurate for this rare cell type, and seeing as the number of SnCs is close to those that were originally marked as serous cells, the classifier might have decided to split on these two subtypes rather than true senescence. We are happy to extensively discuss this important limitation.

2. Potential confounding in model construction

The authors performed differential expression analysis for the middle-aged group and identified 139 senescence marker genes, suggesting a substantial presence of SnCs in this group. However, in constructing the model of healthy cells, data from both young and middle-aged samples were included, which may have introduced confounding effects and potentially compromised the accuracy of the modeling.

We believe this might be a misunderstanding. We assume that the middle-aged group consists of BOTH healthy and senescent cells. The only difference this group has with the old one is the presence of covariate shift in aged donors. We ONLY assume that the young population has no senescent cells. Furthermore, the number of senescence markers is due to the number of cell-types threshold, so it could be less if we are more stringent with this number (increasing it led to 80 marker genes). At the same time, more marker genes do not equal more senescence, it just means there's more genes explaining senescence whenever it exists.

3. Conceptual and methodological concerns in constructing the SenSet list

The rationale for including differentially expressed (DE) genes shared by at least six cell types in the SenSet gene list is questionable. Given that senescence signatures are inherently heterogeneous across cell types, a more suitable approach would be to establish distinct senescence marker lists for each cell type-or at least for clusters of closely related ones. Conversely, if the aim is to identify conserved markers across diverse lung cell types, a more stringent selection pipeline should be applied to ensure the inclusion of robust, universally expressed senescence markers. However, this strategy would contradict the paper's stated focus on a cell type-specific signature. In the section "Cell type-specific enrichment of SenSet genes," the observed enrichment pattern seems largely driven by how the SenSet list was constructed rather than reflecting genuine biological specificity. Furthermore, during validation with human lung tissue snRNA-seq datasets, the authors found that SenSet markers exhibited predominant upregulation in AT1 and AT2 cells, whereas they were largely downregulated in fibroblasts and macrophages. It remains unclear whether this pattern reflects genuine biological differences or systematic bias introduced during the construction of the SenSet list. The authors should therefore refine the conceptual framework and methodological design of SenSet to ensure that it captures biologically meaningful variation rather than analytical artifacts.

The number six was chosen to obtain a list of approximately the same length as the previous lists, otherwise, we run into issues with comparisons (even though we use a hypergeometric test to correct for the size of the set, very drastic differences in size across gene sets would still lead to statistical flukes). The reason we don't use the initial cell-type specific marker lists is because many genes there could be due to noise. Choosing markers based on at least six types provides some guarantee on their expression in senescence. That being said, we are not opposed to other approaches using the cell-type specific lists and check for their enrichment in our annotated cell types in the PCLS.

Overall, we fully agree that the biological meaning and consequences in specific cell types will need further follow up studies, however, we believe that these are out of scope for this resource manuscript.

4. Inappropriate validation metrics and interpretation

In validation analyses, the authors demonstrated that SenSet outperformed previous senescence marker lists. Nevertheless, in the validation using human lung snRNA-seq data, the strategy of evaluating different senescence lists based solely on the number of significantly enriched cell types is not entirely appropriate. A larger number of enriched cell types does not necessarily indicate a better-performing list. Under various stress conditions or during natural aging, senescent cells tend to arise in specific cell types across tissues and organs. Therefore, a robust senescence list should show selective enrichment in truly senescent cell types. To further evaluate the sensitivity and specificity of SenSet relative to other lists by experiments, the authors could leverage senescence-related genetic mouse models-such as the p16reporter line-to test their enrichment in reporter⁺ versus reporter⁻ populations.

For the validation work, we do not simply rely on the number of enriched cell types, but also on 1) the number of markers expressed (Fig. 5B), 2) enrichment in replicative senescence and IPF (Fig. 4A-B, Supp Fig. 8). IPF and replicative senescence studies were meant to address the stress conditions that the reviewer mentions. We respectfully disagree that a mouse model would be appropriate to validate a human tissue-based signature.

To arbitrate the overall reviewer concern about validation and robustness, we are happy to share analysis of human precision cut lung slices treated with Nintedanib, which is an established anti-fibrotic which also has been shown to have senolytic activity in lung tissue (PMID: 36055997). As shown below, we observed that SenSet showed the strongest negative enrichment score (NES) with 24/104 genes contributing to this downregulation. These data are part of a different manuscript. While we cannot include these data here, we are happy to share the data with editor and reviewer to help the arbitration process.

Figure for reviewers removed

Dear Melanie,

We have now received the advisor's additional comments, which I informally enclose below FYI.

Best regards,
Daniel

Daniel Klimmeck, PhD
d.klimmeck@embojournal.org

EMBOJ-2025-122817-T, arbitrating advisor #2's additional comments:

We thank the authors for making a clear effort to address each of the concerns. Nevertheless, several responses remain insufficiently detailed and lack quantitative support, and thus do not fully resolve the core issues raised. Further strengthening the responses through additional analyses or clearer and more rigorous methodological explanations would substantially enhance the manuscript's clarity and overall rigor.

1. Misclassification of tracheobronchial serous cells

We appreciate that the authors acknowledge the alternative explanation and express willingness to incorporate it into the revised discussion. They propose a plausible hypothesis that the observed misclassification may arise from annotation refinement issues within the HLCA dataset and appropriately recognize this as an important limitation. However, the response remains largely descriptive rather than analytical. In particular, it remains unclear whether SMG serous cells are more enriched in the oldest age group compared with SMG duct cells, and whether these SMG serous cells genuinely exhibit senescence-associated transcriptional features. Furthermore, after excluding SMG duct cells, it would be important to assess whether SMG serous cells can be further subdivided into senescent and non-senescent states, and whether SMG serous-specific senescence-associated genes can be identified. Overall, additional statistical and transcriptomic analyses would be necessary to adequately address this issue.

2. Potential confounding in model construction

We appreciate that the authors clearly restate their assumption that middle-aged samples comprise a mixture of healthy and senescent populations. However, this clarification does not directly address the concern regarding potential confounding introduced by including middle-aged samples—which may contain senescent cells—in the construction of the healthy cell model. The central issue is whether it is methodologically justified to include both young and middle-aged samples when defining the healthy reference population. Alternatively, the authors should provide evidence or justification demonstrating

why incorporating middle-aged samples improves model performance relative to training the model exclusively on young samples.

3. Conceptual and methodological concerns regarding SenSet construction

We understand the authors' rationale for requiring enrichment across six cell types in order to maintain comparable gene-set sizes. However, if SenSet is constructed in this manner, the title "**SenSet, a cell-type-specific senescence signature in the lung**" may constitute an overstatement or may not accurately reflect the underlying methodology, and revision of this claim should be considered. Moreover, during SenSet construction, if the number of differentially expressed genes between senescent and non-senescent cells varies substantially across cell types, then cell types contributing more genes will disproportionately influence the composition of SenSet. Consequently, downstream analyses may become biased toward those cell types with greater gene-set enrichment. However, the number of differentially expressed genes should not itself determine senescence classification, as for cell types with fewer differential genes, even a small number of relevant markers may be biologically meaningful and indicative of a senescent state. As a result, concerns regarding potential analytical artifacts remain unresolved. If the authors are currently unable to more directly evaluate whether the SenSet construction pipeline introduces systematic bias, these potential limitations and application risks should be clearly and explicitly discussed.

4. Validation metrics and interpretation

We appreciate that the authors clarify their use of multiple validation metrics beyond simple enrichment counts and that they present SenSet's performance across replicative senescence, telomere dysfunction, fibrosis, and precision-cut lung slice (PCLS) models. The inclusion of additional Nintedanib-treated PCLS data further strengthens the validation. One remaining issue that should be addressed is the quantitative relationship between SenSet-enriched cell types identified in single-cell datasets and senescence-associated marker-positive cells in tissue contexts. Specifically, it would be informative to quantify the proportion of SenSet-enriched cell types in senescence markers (such as p16, p21, or SA- β -gal)-positive cells on sections, as well as the fraction of senescence-marker-positive cells within each of these cell types. Such analyses would provide a more direct and biologically grounded assessment of SenSet specificity.

Dear Daniel,

Please find attached our responses to the last comments from the arbitrating advisor. We also took your comments during our call into account. You will see that we have performed additional analysis, provided extended clarifications and suggestions to changes to our manuscript.

We hope this clarifies the last open questions for consideration at EMBO Journal.

Looking forward to hearing from you and we remain with warm regards,

On behalf of all corresponding authors,

Melanie

EMBOJ-2025-122817-T, arbitrating advisor #2's additional comments:

We thank the authors for making a clear effort to address each of the concerns. Nevertheless, several responses remain insufficiently detailed and lack quantitative support, and thus do not fully resolve the core issues raised. Further strengthening the responses through additional analyses or clearer and more rigorous methodological explanations would substantially enhance the manuscript's clarity and overall rigor.

1. Misclassification of tracheobronchial serous cells

We appreciate that the authors acknowledge the alternative explanation and express willingness to incorporate it into the revised discussion. They propose a plausible hypothesis that the observed misclassification may arise from annotation refinement issues within the HLCA dataset and appropriately recognize this as an important limitation. However, the response remains largely descriptive rather than analytical. In particular, it remains unclear whether SMG serous cells are more enriched in the oldest age group compared with SMG duct cells, and whether these SMG serous cells genuinely exhibit senescence-associated transcriptional features. Furthermore, after excluding SMG duct cells, it would be important to assess whether SMG serous cells can be further subdivided into senescent and non-senescent states, and whether SMG serous-specific senescence-associated genes can be identified. Overall, additional statistical and transcriptomic analyses would be necessary to adequately address this issue.

2. Potential confounding in model construction

We appreciate that the authors clearly restate their assumption that middle-aged samples comprise a mixture of healthy and senescent populations. However, this clarification does not directly address the concern regarding potential confounding introduced by including middle-aged samples—which may contain senescent cells—in the construction of the healthy cell model. The central issue is whether it is methodologically justified to include both young and middle-aged samples when defining the healthy reference population. Alternatively, the authors should provide evidence or justification demonstrating why incorporating middle-aged samples improves model performance relative to training the model exclusively on young samples.

3. Conceptual and methodological concerns regarding SenSet construction

We understand the authors' rationale for requiring enrichment across six cell types in order to maintain comparable gene-set sizes. However, if SenSet is constructed in this manner, the title "**SenSet, a cell-type-specific senescence signature in the lung**" may constitute an overstatement or may not accurately reflect the underlying methodology, and revision of this claim should be considered. Moreover, during SenSet construction, if the number of differentially expressed genes between senescent and non-senescent cells varies substantially across cell types, then cell types contributing more genes will disproportionately influence the composition of SenSet. Consequently, downstream analyses may become biased toward those cell types with greater gene-set enrichment. However, the number of differentially expressed genes should not itself determine senescence classification, as for cell types with fewer differential genes, even a small number of relevant markers may be biologically meaningful and indicative of a senescent state. As a result, concerns regarding potential analytical artifacts remain unresolved. If the authors are currently unable to more directly evaluate whether the SenSet construction pipeline introduces systematic bias, these potential limitations and application risks should be clearly and explicitly discussed.

4. Validation metrics and interpretation

We appreciate that the authors clarify their use of multiple validation metrics beyond simple enrichment counts and that they present SenSet's performance across replicative senescence, telomere dysfunction, fibrosis, and precision-cut lung slice (PCLS) models. The inclusion of additional Nintedanib-treated PCLS data further strengthens the validation. One remaining issue that should be addressed is the quantitative relationship between SenSet-enriched cell types identified in single-cell datasets and senescence-associated marker-positive cells in tissue contexts. Specifically, it would be informative to quantify the proportion of SenSet-enriched cell types in senescence markers (such as p16, p21, or SA- β -gal)-positive cells on sections, as well as the fraction of senescence-marker-positive cells within each of these cell types. Such analyses would provide a more direct and biologically grounded assessment of SenSet specificity.

1. Misclassification of tracheobronchial serous cells

We appreciate that the authors acknowledge the alternative explanation and express willingness to incorporate it into the revised discussion. They propose a plausible hypothesis that the observed misclassification may arise from annotation refinement issues within the HLCA dataset and appropriately recognize this as an important limitation. However, the response remains largely descriptive rather than analytical. In particular, it remains unclear whether SMG serous cells are more enriched in the oldest age group compared with SMG duct cells, and whether these SMG serous cells genuinely exhibit senescence-associated transcriptional features. Furthermore, after excluding SMG duct cells, it would be important to assess whether SMG serous cells can be further subdivided into senescent and non-senescent states, and whether SMG serous-specific senescence-associated genes can be identified. Overall, additional statistical and transcriptomic analyses would be necessary to adequately address this issue.

We appreciate that the reviewer agrees that we "propose a plausible hypothesis that the observed misclassification may arise from annotation refinement issues within the HLCA dataset" and that we "appropriately recognize this as an important limitation". In response to the reviewer's comment regarding potential misclassification of serous cells, we re-examined the HLCA atlas cell-type composition by age (Supplementary Fig. 2 and below). Serous cells were the least represented population in old tissue ($n = 97$), followed by acinar cells ($n = 250$) and hillock cells ($n = 319$). Our method classified a high fraction of cells in two groups as senescent (76% in serous; 49% in acinar). We thus suspect that the high senescence rates in these populations may also be driven in part by small-sample effects, in addition to the potential labeling issues discussed above.

Based on these comments and observation we will add the following to Discussion 'We observe good performance for the cell type specific analysis. However, for cell types with very few cells (Supplementary Fig. 2) the method can lead to overestimate of the senescence proportions. Specifically, Serous cells were the least represented population in old tissue ($n = 97$), followed by acinar cells ($n = 250$). Our method classified a high fraction of these cells as senescent (76% in serous; 49% in acinar). We thus recommend using SenSet set for larger number of cells when performing cell type specific analysis'.

Figure for reviewers removed

2. Potential confounding in model construction

We appreciate that the authors clearly restate their assumption that middle-aged samples comprise a mixture of healthy and senescent populations. However, this clarification does not directly address the concern regarding potential confounding introduced by including middle-aged samples—which may contain senescent cells—in the construction of the healthy cell model. The central issue is whether it is methodologically justified to include both young and middle-aged samples when defining the healthy reference population. Alternatively, the authors should provide evidence or justification demonstrating why incorporating middle-aged samples improves model performance relative to training the model exclusively on young samples.

We think we now understand the concern the reviewer is raising and hope to address the issue more directly. The inclusion of a mixed, middle-aged group (containing both senescent and healthy cells) is necessary for PUC learning. Because old samples may differ substantially from young samples due to covariate shift (age-related changes), it is **statistically impossible** to distinguish senescent from non-senescent cells in old tissue using only young and old groups. In this setting, young cells could be arbitrarily mapped to any subgroup of cells in the old (the covariate shift could be arbitrary). Therefore, the middle-aged group serves as an "anchor" or important intermediate point, by providing overlap between the young training distribution (the healthy cells) and the aged test distribution (via the relative expression of healthy and senescent cells in middle-aged). The equivalent biological interpretation is that, since aging is a continuous process, a model trained exclusively on young and old cells is poorly calibrated and risks conflating gradual aging effects with true senescence. We hope this clarifies our methodology.

3. Conceptual and methodological concerns regarding SenSet construction

We understand the authors' rationale for requiring enrichment across six cell types in order to maintain comparable gene-set sizes. However, if SenSet is constructed in this manner, the title "**SenSet, a cell-type-specific senescence signature in the lung**" may constitute an overstatement or may not accurately reflect the underlying methodology, and revision of this claim should be considered. Moreover, during SenSet construction, if the number of differentially expressed genes between senescent and non-senescent cells varies substantially across cell types, then cell types contributing more genes will disproportionately influence the composition of SenSet. Consequently, downstream analyses may become biased toward those cell types with greater gene-set enrichment. However, the number of differentially expressed genes should not itself determine senescence classification, as for cell types with fewer differential genes, even a small number of relevant markers may be biologically meaningful and indicative of a senescent state. As a result, concerns regarding potential analytical artifacts remain unresolved. If the authors are currently unable to more directly evaluate whether the SenSet construction pipeline introduces systematic bias, these potential limitations and application risks should be clearly and explicitly discussed.

We agree with the reviewer that some section titles did not accurately reflect the scope of the analysis. We will therefore rename the cell-type analysis section to "Cell-type representation of SenSet genes in the atlas." To further address the comment on the construction of SenSet, we revisited the full set of cell type-specific marker genes identified during SenSet derivation and show in the figure below, for each cell type, the fraction of these markers that were ultimately retained in SenSet. For 14 of the 22 cell types, more than approximately ~80% of the derived markers were included, suggesting broadly similar inclusion rates across most types. The main exceptions were acinar, hillock, alveolar macrophage, and basal cells. For acinar and hillock cells, the underlying sample sizes were small (as discussed in our response to point 1), and we therefore suspect that the corresponding differential expression signatures, and thus excluded markers, may be noisy. For alveolar macrophages, we performed an additional check using our treated PCLS data: only about one third (68/185) of the

macrophage markers that were not included in SenSet were also differentially expressed in PCLS (second figure below), suggesting that many of the excluded macrophage markers are unlikely to be robust across datasets. For basal cells, we could not perform the same validation because basal cells were not identifiable in our PCLS samples; however, we will provide the complete basal-cell marker list as a supplementary table for transparency.

Figures for reviewers removed

4. Validation metrics and interpretation

We appreciate that the authors clarify their use of multiple validation metrics beyond simple enrichment counts and that they present SenSet's performance across replicative senescence, telomere dysfunction, fibrosis, and precision-cut lung slice (PCLS) models. The inclusion of additional Nintedanib-treated PCLS data further strengthens the validation. One remaining issue that should be addressed is the quantitative relationship between SenSet-enriched cell types identified in single-cell datasets and senescence-associated marker-positive cells in tissue contexts. Specifically, it would be informative to quantify the proportion of SenSet-enriched cell types in senescence markers (such as p16, p21, or SA- β -gal)-positive cells on sections, as well as the fraction of senescence-marker-positive cells within each of these cell types. Such analyses would provide a more direct and biologically grounded assessment of SenSet specificity.

All SenSet enriched cell types are p21 positive (p21 is part of the SenSet list) and we have plots showing the expression of p16, p21 in the supplement for both HLCA and PCLS in the manuscript. In all of our models/disease samples senescence markers, such as p16, p21, or SA- β -gal-positive cells are described either in the manuscript (Fig. 4, Fig. 5, Supplement Fig .5, among others) or well documented in the literature.

Dear Dr Koenigshoff,

Thank you again for the submission of your amended manuscript (EMBOJ-2025-122817-T) to The EMBO Journal. We have carefully assessed your additional point-by-point responses to the arbitration input, and found the remaining issues to be addressed satisfactorily.

We are thus pleased to inform you that we can offer to swiftly move forward towards acceptance of this work at The EMBO Journal as a data resource, pending minor revision as detailed below.

Please revise your manuscript accordingly by integrating additional data from your rebuttal response or textual adjustments and introducing caveats where appropriate.

Also, we now need you to take care of a number of minor issues related to formatting and data annotation, which I will share shortly in a separate message, together with additional changes and requests by our production team and information regarding Source Data provision.

Please submit a revised version of the manuscript at your earliest convenience using the link enclosed.

As you might have noticed on our web page, every paper at the EMBO Journal includes a 'Synopsis', displayed on the html and freely accessible to all readers. The synopsis includes a 'model' figure as well as 2-5 one-short-sentence bullet points that summarize the article. I would appreciate if you could provide this figure and the bullet points.

Thank you for giving us the chance to consider your manuscript for The EMBO Journal, I look forward to hearing from you and receiving your final revised version of the manuscript.

Best regards,

Daniel Klimmeck

Dear Dr Koenigshoff,

Further to below, I enclose the mentioned additional formatting requirements for your final manuscript.

Please consider them carefully and let us know any time of there are additional questions related.

Best regards,

Daniel Klimmeck

>> Please add up to five keywords to your study.

>> Provide the main manuscript text as .docx file.

>> Section order should be as follows: title page with complete author information, abstract, keywords, introduction, results, discussion, methods, data availability section, acknowledgements, disclosure and competing interests statement, references, main figure legends, tables, expanded figure legends.

>> Adjust the title of the 'Declaration of Interests' section to 'Disclosure and Competing Interests Statement'.

>> Provide a completed Author Checklist.

>> Figures in separate files: Please remove the figures from the manuscript and upload them as separate, high res figure files, and compile the legends at the end of the manuscript text.

>> References: please adjust reference format to EMBO Journal format, 10 authors et al.

>> Figure callouts: please recheck callouts for Fig 5H, Fig 6F; Suppl. Fig 13; Suppl. Fig. 12 is called out before Suppl. Fig 11, Suppl. Figure 5 is called out after Suppl. Fig 19. There is a callout for a Suppl. Table 1, which is missing. Please add this to the appendix as "Appendix Table S1", or upload it as a separate file and name it "Table EV1".

>> Please add a Reagents and Tools table to the Methods section, as a separate file using the existing template in the Guide For Authors, listing key reagents, experimental models, software and relevant equipment.

>> Add a separate 'Statistical Analysis' section to the Methods part, detailing the algorithms and statistical tests applied.

>> Please provide editable versions of Table 1 -5 and place them after the figure legends.

>> Please provide source data for the study as to the separate request e-mail. Source data should be uploaded as one (zipped) file per figure.

>>Appendix File with ToC: Please correct the name of the file with suppl. figures to "Appendix", add a table of contents with page numbers, and correct the figure nomenclature to "Appendix Figure S1" - "Appendix Figure S19".

>> Data availability section: merge the Code Availability with the Data Availability section and move this to the end of the Methods section. Deposit newly generated sequencing data in a public repository and provide a URL for the dataset, ensuring privacy is released and the data is public.

>> Please recheck references for the bioRxiv entry De Man et al. (2023) and update the citation if in the meantime published as regular article.

>> Consider additional changes and comments from our production team as indicated below:

- DAS:

Please note that the specific URLs for SNT657.STXR.948, SNT536.TGBM.294, SNT829.BMNR.662, SNT983.FMRG.925, SNT493.RSKL.732, SNT792.KVKQ.344, SNT339.DQDB.424, SNT432.BQVB.669, SNT493.TKJD.678, SNT397.SPGW.436, SNT757.BGFN.798, SNT372.NMWW.264, SNT627.QHCP.732, SNT835.GVZP.228. datasets are not provided in the data availability statement.

- Figure legends:

1. Please define the annotated p values ****/****/**/* as well as provide the exact p-values for the same in the legend of figure 2J as appropriate.
2. Please note that the exact p values are not provided in the legends of figures 5E, F, G, I; EV10 C-E
3. Please indicate the statistical test used for data analysis in the legends of figures 2I, 3C, EV6, EV10 A, C, D, E, H
4. Please note that the box plots need to be defined in terms of minima, maxima, centre, bounds of box and whiskers, and percentile in the legends of figures 2J, K; 3C, 4C-F; EV5 B, C; EV11 B
5. Please note that information related to n is missing in the legends of figures 2J, K; 3C, EV5 A-C; EV11 B
6. Please note that the error bars are not defined in the legends of figures 5E, F, G, I; EV5 A, EV10 A, C, D, E, H

Further information is available in our Guide For Authors: <https://link.springer.com/journal/44318/submission-guidelines>

Dear Dr Koenigshoff,

Thank you again for the submission of your amended manuscript (EMBOJ-2025-122817-T) to The EMBO Journal. We have carefully assessed your additional point-by-point responses to the arbitration input, and found the remaining issues to be addressed satisfactorily.

We are thus pleased to inform you that we can offer to swiftly move forward towards acceptance of this work at The EMBO Journal as a data resource, pending minor revision as detailed below.

Please revise your manuscript accordingly by integrating additional data from your rebuttal response or textual adjustments and introducing caveats where appropriate.

Also, we now need you to take care of a number of minor issues related to formatting and data annotation, which I will share shortly in a separate message, together with additional changes and requests by our production team and information regarding Source Data provision.

Please submit a revised version of the manuscript at your earliest convenience using the link enclosed.

As you might have noticed on our web page, every paper at the EMBO Journal includes a 'Synopsis', displayed on the html and freely accessible to all readers. The synopsis includes a 'model' figure as well as 2-5 one-short-sentence bullet points that summarize the article. I would appreciate if you could provide this figure and the bullet points.

Thank you for giving us the chance to consider your manuscript for The EMBO Journal, I look forward to hearing from you and receiving your final revised version of the manuscript.

Best regards,

Daniel Klimmeck

The authors addressed the remaining editorial issues..

Dear Dr Koenigshoff,

Thank you for submitting the revised version of your manuscript. I have now evaluated your amended manuscript and concluded that the remaining minor concerns have been sufficiently addressed.

I am thus pleased to inform you that your manuscript has been accepted for publication in the EMBO Journal.

Best regards,

Daniel Klimmeck

Daniel Klimmeck, PhD
Senior Editor
The EMBO Journal
EMBO
Postfach 1022-40
Meyerhofstrasse 1
D-69117 Heidelberg
contact@embojournal.org